# Lateral facet growth of ice and snow I: observations and applications to secondary habits

Jon Nelson[1], Brian Swanson[2,3]

[1]Redmond Physical Sciences, Redmond, 98052, USA
[2]Earth and Space Sciences, University of Washington, Seattle, 98195, USA
[3]Laucks Foundation Research, Salt Spring Island BC, V8K2E5, Canada

*Correspondence to*: Jon Nelson (jontne@gmail.com)

**Abstract.** Often overlooked in studies of ice growth is how the crystal facets increase in area, that is, grow laterally. This paper reports on observations and applications of such lateral facet growth for vapor-grown ice in air. Using a new crystal-growth chamber, we observed air pockets forming at crystal corners when a sublimated crystal is regrown. This observation indicates that the lateral spreading of a face can, under some conditions, extend as a thin overhang over the adjoining region. We argue that this extension is driven by a flux of surface-mobile molecules across the face to the lateral-growth front. Following the pioneering work on this topic by A. Yamashita, we call this flux "adjoining surface transport" (AST) and the extension overgrowth "protruding growth". Further experiments revealed other types of pockets that are difficult to explain without invoking AST and protruding growth. We develop a simple model for lateral facet growth on a tabular crystal in air, finding that AST is required to explain observations of facet spreading. Applying the AST concept to observed ice and snow crystals, we argue that AST promotes facet spreading, causes protruding growth, and alters layer nucleation rates. In particular, depending on the conditions, combinations of lateral- and normal-growth processes can help explain presently inexplicable secondary features and habits such as air pockets, small circular centers in dendrites, hollow structure, multiple-capped columns, scrolls, sheath clusters, and trigonals. For dendrites and sheaths, AST may increase their maximum dimensions and round their tips. Although these applications presently lack quantitative detail, the overall body of evidence here demonstrates that any complete model of ice growth from the vapor should include such lateral growth processes.

## 1 Introduction

Snow crystals, or ice crystals precipitated to ground, are known for their wide variety, a notion perhaps first popularized via the photomicrographs of Bentley (1901). Although his classic collection (Bentley and Humphreys, 1962) indeed shows an immense variety of crystal forms, it still contains only a fraction of the 121 general categories now recognized (Kikuchi et al., 2013). Beyond the aesthetics, numerous atmospheric processes are affected by in-cloud ice-crystal size and shape. These sizes (maximum dimensions) generally range from about 10–1000 μm (e.g., Um et al., 2015), with ratios of axial length to maximum diameter that can vary from less than 0.02 for thin prisms (Shimizu, 1963) to over 100 for dendrites (Takahashi et al., 1991). In between, the more equidimensional crystals that form near −10 °C fall further in a given length of time, and thus tend to collect more rime (drops that freeze on impact), growing into large, blobby graupel precipitation (Fukuta and Takahashi, 1999). In contrast, the thin tabular forms such as the dendrites instead tend to fall the slowest, despite growing the fastest from the vapor, thus lofting up higher in clouds (e.g., thunderstorm anvils; Um et al., 2015) before precipitating. The vapor-diffusional growth rate itself was found to influence the collisional ice-particle charging rate (Baker et al., 1987), a phenomenon consistent with several theories (e.g., Dash et al., 2001; Nelson and Baker, 2003), leading to it being well accepted as the main charging mechanism in thunderstorms. Concerning climate, clouds mainly containing ice crystals (e.g., cirrus) significantly affect the

Earth's radiation budget, but because the overall process is complex, a precise estimate of the ice-cloud impact on climate remains elusive (e.g., IPCC, 2013). Indeed, some research suggests that even small surface features of cloud ice crystals can significantly affect this radiative transfer (e.g., Smith et al., 2015; Järvinen et al., 2018).

Research on ice-crystal growth from the vapor usually focuses on the rates of growth normal to the basal and prism faces (e.g., Takahashi et al. 1991). The rates are often called the linear growth rates (e.g., Lamb and Scott, 1972), but to help distinguish face-normal growth from face-lateral (or areal) growth, we refer to them as the normal growth rates. For a given crystal, the normal rates on its basal and prism faces determine the crystal's maximum dimensions and aspect ratio, thus defining the primary habit. But ice and snow crystals usually have more complex shape features, such as hollows and branches, known as the secondary habit (e.g., Kikuchi et al., 2013).

Both primary and secondary habit depend on temperature and humidity, as first shown as the Nakaya habit diagram. This diagram has generally remained the same since Ukichiro Nakaya first proposed it (Nakaya, 1954), though some extensions and modifications have come from subsequent studies (e.g., Hallett and Mason, 1958; Takahashi et al., 1991; Bailey and Hallett, 2004; Takahashi, 2014). Concerning the mechanism for the primary habit, at liquid-water saturation this habit likely arises from the temperature dependence of the layer-nucleation rates (Nelson and Knight, 1998). At the lower supersaturations, defects likely control the primary habit (e.g., Bacon et al., 2003; Harrington et al., 2019), though this part of the habit diagram has not been studied as extensively, with results less consistent, as that near liquid-water saturation.

Secondary habit features have been observed for a long time, but have seen relatively little study. Wilson Bentley, known for his extensive photo-micrography work, paid much attention to the crystals' interior markings including various air enclosures (Bentley, 1901, 1924). For example, in his 1901 paper, he suggested that these markings and air pockets (enclosures) give clues about the crystal's trajectory, an idea no doubt true, yet both unexploited and unexplained. Later, Maeno and Kuroiwa (1966) examined the patterns of apparent air enclosures in snow crystals, verifying through sublimation and melting that they were indeed enclosed pockets of air and not surface features. More recently, Yamashita categorized 16 types of pockets in tabular crystals (2016, 2019). Several examples of air enclosures in small prisms can be seen in Fig. 1d–f. Studies of other interior markings include those of hollows (Mason et al., 1963) and of ridges and ribs on branch backsides (Nelson, 2005; Yamashita, 2013; Shimada and Ohtake, 2016, 2018).

Although the normal-growth mechanisms, including layer nucleation and defect-driven steps, provide a solid framework for understanding the primary habit and other crystal features, many secondary habits remain inexplicable. In addition to the air pockets, these other habits include i) the small spherical form at the center in many dendritic snow crystals, ii) the thin basal planes in capped and multiple-capped columns, iii) the abrupt bending of thin prism planes in scroll crystals, iv) the structure of sheath clusters, and v) trigonal crystals. Can inclusion of lateral facet growth processes help explain these forms? We argue here that they can, and thus should be included in any complete ice-growth model.

## 2 Background

The most widely used model for the growth of crystal faces from the vapor is the "BCF" model (Burton, Cabrera, and Frank, 1951; see Woodruff, 2015 for updates and history). This model supposes that a given molecule in the vapor above a faceted surface strikes the crystal surface and becomes temporarily trapped in a mobile state until either desorbing back to the vapor or migrating along the surface and reaching a more strongly bound state at a step edge. These individual steps are abrupt changes in surface height, generally just one or two crystal layers, a height much less than their usual separation, and thus the face appears flat.

As a source of step edges, BCF and later studies considered layer nucleation and defect-generated steps, most commonly spiral-step sources. The former has been argued to be the main source for ice-crystal growth from the vapor under most atmospheric conditions (e.g., Knight, 1972; Nelson and Knight, 1998), but not for many other crystals (Frank, 1982). Under relatively low supersaturations, defect-generated step sources usually dominate. Once a step is generated, the flow of molecules to the step edge causes it to sweep across the macroscopically flat facet (or face, the terms used interchangeably here). When one step sweeps past a given position, that point on the face advances normally by the step height, and thus the frequency of the sweeping steps gives the normal growth rate. Hence, normal growth rates are proportional to the step-generation rate. In contrast, non-flat surface regions are said to be "rough" and grow at the maximum rate allowed by the rate of impingement of vapor molecules. Such growth is called either rough growth or continuous growth, with individual steps close enough together that all impinging vapor molecules reach a step. In ice growth from the melt, continuous growth dominates for non-basal orientations, but for vapor growth, the leading fronts (i.e., outermost faces that define the maximum diameter and have the fastest normal growth) are usually facetted. Individual steps, and steps clumped into macrosteps, instead tend to have a rough edge as indicated by their curved perimeter (generally circular or spiral). Also, when the leading front is very thin, it may appear rounded.

The BCF model of surface diffusion assumes that the mobile surface molecules are sparse and non-interacting. For ice, this assumption is suspect over much of the atmospheric temperature range. Specifically, the ice–vapor interface is widely thought to contain significant disorder, a phenomenon also called the quasi-liquid layer QLL (e.g., Rosenberg, 2005). A recent study finds that this "layer" is limited to two ice bilayers (~0.74 nm) below −2 °C, and less than half that below −16 °C (Nagata et al., 2019). Despite this layer's thinness, such a surface still deviates greatly from the BCF assumption. Nevertheless, the BCF model is often used to interpret experimental results (e.g., Sei and Gonda, 1989; Asakawa et al., 2014). A key parameter in the model is the mean migration distance $x_s$ of a mobile molecule on the surface before desorbing, a distance that should differ between the basal (*b*) and prism (*p*) faces as well as depend on temperature. With interactions between these surface-mobile molecules (e.g., Myers-Beaghton and Vvedensky, 1990), $x_s$ should also depend on supersaturation. In addition, the migration of surface vacancies may also affect $x_s$ (Frank, 1993). Experiments reported in the 1960s indicated that $x_s$ on the basal face varied dramatically with temperature, changing by a factor of 5–7 between about −7 and −12 °C (Mason et al., 1963; Kobayashi, 1967). Although the exact values of $x_s$ may be disputed, both studies independently found the values to be largest in the tabular regime, smallest in the columnar. Corresponding values for the prism have not been determined. Later, Nelson and Knight (1998) found a similarly sharp behavior in basal-face critical supersaturation between these temperatures. A possible link between these two parameters is clustering of the mobile species responsible for growth: when the temperature is such that clustering is strong, the critical supersaturation is low and surface-mobile molecules would become temporarily trapped in sub-critical nuclei, giving them very low mobility. Thus, the critical supersaturation would be low when $x_s$ is low and vice-versa as found by experiments. The values of the measured critical supersaturations led Nelson and Knight to conclude that the surface was indeed disordered but " ...the view of the ice surface as a liquid layer is not a useful idealization for crystal growth processes." Hence, at least as a first approximation, it is still useful to compare observed behavior of ice to the BCF model and make use of measured $x_s$ values.

A second simplification of BCF is the assumption of a uniform vapor density. This condition should hold in a pure vapor, but not for ice growth from the vapor in an atmosphere of air. Gilmer et al. (1971) showed that an exact treatment predicts vapor-depleted air immediately adjacent to a step edge, slowing the crystal growth rate over that of BCF, but the exact calculation is difficult for a 3-D polyhedral crystal such as that for ice (Nelson, 1994). Instead, atmospheric crystal-growth models usually assume a locally uniform vapor density near the step source and allow the vapor density to monotonically decrease or increase across the surface. (Most cloud models use the more extreme simplifications of the "capacitance model", which includes no detail of surface structure and assumes local equilibrium over the entire surface. But the recent work by Harrington et al., 2019 is

a welcome exception.) As the crystal shape presents a greater modeling challenge, recent work has focused less on the exact surface model than on the modeling of more realistic crystal shapes (e.g., Wood et al., 2001).

Atmospheric ice crystals generally begin with the simplest of shapes—a solid ice sphere, also called a droxtal. The droxtal forms when a droplet freezes. That freezing is a crucial first step for atmospheric ice was greatly supported by the extensive

cloud studies of Hobbs and Rangno (e.g., Hobbs and Rangno, 1985). This two-step process of vapor-to-droplet then droplet-to-ice, instead of direct vapor nucleation to ice, is thought to prevail because the nucleation rate is exceedingly sensitive to the interfacial surface energy, with the surface energy for the liquid–vapor case being lower than that for the ice–vapor case (e.g., ten Wolde and Frenkel, 1999).

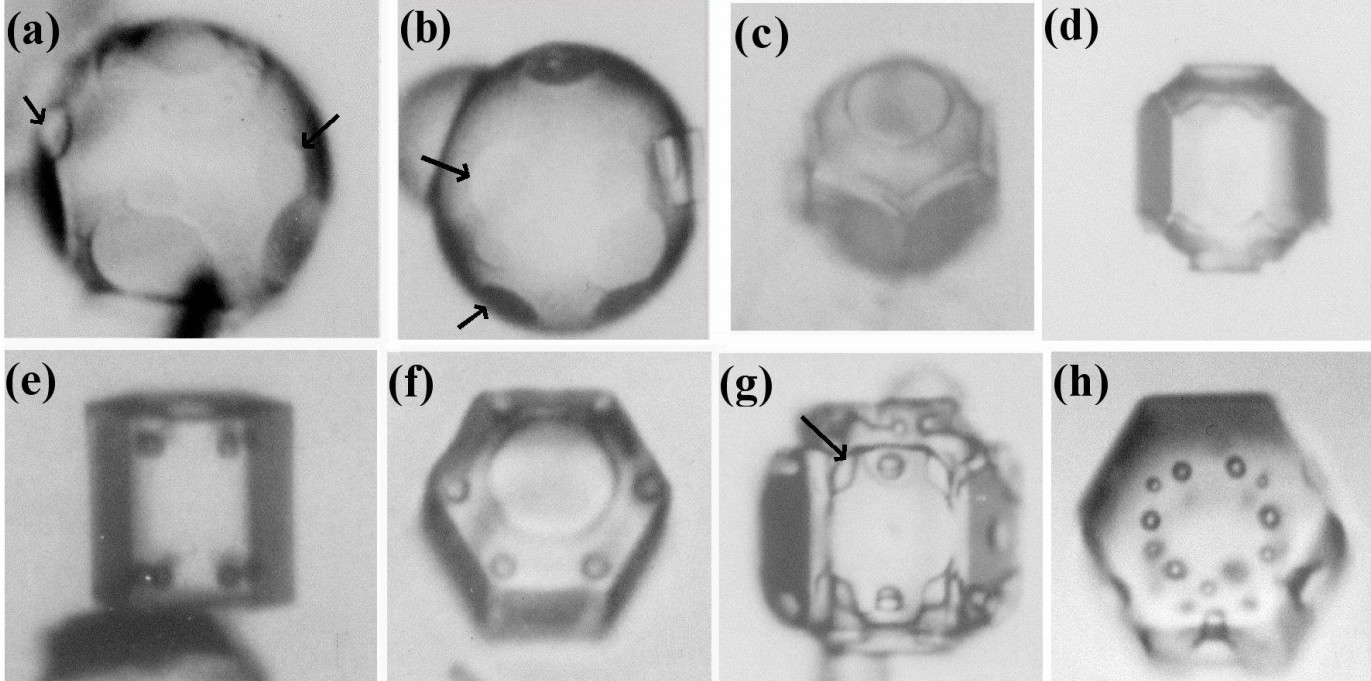

**Figure 1:** Crystals at different stages between large droxtals (just-frozen droplets) and prisms at temperatures between −6 and −12 °C and supersaturations near liquid-water saturation. Top row shows initial development of basal and prism faces, with some pyramidal faces (marked with arrows) in (a) and (b). Bottom row shows filled-out faces with corner pockets in (e) and (f). In (g)

and (h), pockets appear where pyramidal faces may have hollowed before being overtaken by basal and prism faces. Arrow in (g) marks an apparent protrusion. Diameters are within 45–90 μm. (From the cloud chamber, courtesy of A. Yamashita.)

After a droxtal forms, facets start to develop as shown by the examples in Fig. 1a,b. The facets spread in all directions parallel to the facet as in (c) and (d) until intersecting another facet. The crystals here are larger than typical droxtals from cloud

droplets, but Gonda and Yamazaki (1984) observed a similar transformation in smaller (~20-μm diameter) droxtals. This facet spreading, or "F-growth", is partly driven by adjoining surface transport (AST) in which mobile molecules on the facet migrate over the edge, adhering to the lateral-edge front, region e–c, as sketched in Fig. 2a–c. Figure 1a–d shows these edge fronts as rounded, indicating rough edge and hence an efficient collector of molecules. This facet spreading also occurs on larger crystals when a similarly rounded form changes to a flat, facetted form. The cases in Fig. 1a,b also show small pyramidal faces between

the basal and each prism that are not included in Fig. 2. These faces are usually engulfed by the basal and prism faces, which are growing faster laterally but slower normally.

After the *b* and *p* faces fill out, the crystal is more easily described by its normal growth rates $N_B$ and $N_P$, though the faces are also growing in area. This type of lateral growth is a standard aspect of polyhedral growth, so we refer to it as "S-type", but we focus on the F- and P-types (described next). During the facet-spreading phase Fig. 2b, normal growth may also be occurring, but surface-mobile molecules on the relatively small facet are already close to the molecular sink at the lateral-growth front **e–c**. Thus, the propagation rate (and nucleation of new layers in the absence of a permanent step source) of surface steps will be reduced until the facet radius **m–e** exceeds the surface migration distance $x_s$. Also, if $x_s$ and radius **m–e** both exceed the thickness **e–c**, then this AST flux may lead to a lateral growth rate F that is much greater than the normal rates N due to the relatively large molecular collection region on the facet.

Under some conditions, the facet spreading may produce an overhanging planar extension. Following Yamashita (2014), we call the growth of this planar overhang "protruding growth" or "P-type", marking it "P" in Fig. 2b'. Here, the lateral-edge front **e–i** extends over the inside corner **c**, becoming narrower than the case in Fig. 2b. The thickness of this edge-front should depend on how far surface-mobile molecules can migrate on the rough edge. If this length-scale is $l_r$, then the edge thickness should be of order $l_r$ and less than $x_s$ for P-growth. This length-scale has not been studied, so we will not analyze it further except to note that the high-density of growth sites on a rough edge compared to that on a facet would suggest that $l_r << x_s$. Such P-growth from two intersecting faces, such as *b* and *p* in Fig. 2c', produces a pocket. The examples in Fig. 1e,f show pockets in the corners, which likely formed from the intersection of three protruding faces. The pockets in Fig. 1g are less clear, but the pocket may be due to protruding growth from opposing directions on the pyramidal face. This image also seems to show thickened protrusions at the four corners of the front prism face. The case in Fig. 1h seems to show both sets of pockets (i.e., both cases f and g).

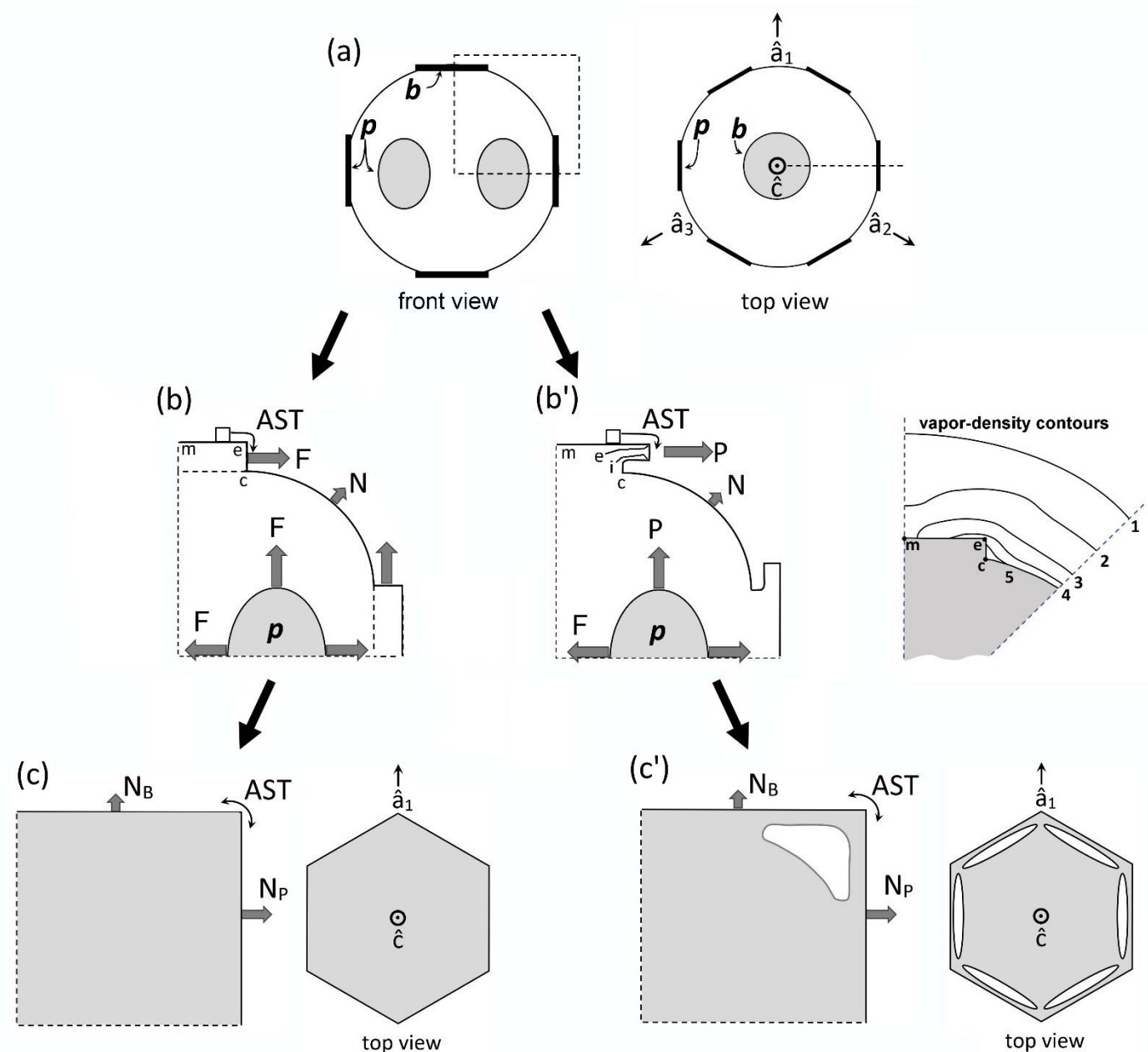

**Figure 2:** Concepts of lateral facet growth (types **F**, **P**, **S**) for the droxtal-to-prism transition, driven by AST. (a) Droxtal with small basal **b** and prism **p** faces. Front view at left shows four prism and both basal faces. The top view at right shows all six prisms and the crystallographic directions of the c-axis ĉ and the a-axes â. (b) Upper-right quadrant of (a) in dashed box, front view. Facet spreading **F** on top basal and two prism faces largely driven by adjoining surface transport AST. Normal growth **N** occurs on rough regions between faces (continuous growth). m is the middle of the facet. (c) Filled-out basal and prism faces. AST continues, likely with net amount to faster-growing face. Top view shows crystallographic directions. (b') Like (b), except having protruding growth **P** between basal and prism faces. e–i is the new lateral-growth front. (c') Like (c), except with air enclosure. After (c) and (c'), standard lateral growth (**S**-type) occurs due to normal growth of adjoining faces on a fully facetted crystal. Middle row, right shows qualitative features of five representative vapor-density contours near the lateral-growth front in (b).

Concerning the conditions needed for such droxtal pockets, the size of the droxtals may be important. In the cases shown here, the radii are all above 22 μm. In the figures of Gonda and Yamazaki (1978, 1984), the droxtals have radii of about 10 and 15 μm, yet do not reveal any pockets upon filling out. Given their small droxtal sizes and darkness of their images, one cannot rule out the existence of very small pockets, but their results show no indication of pockets of the scale seen in Fig. 1. Their study

examined droxtals at −7 and −15 °C with air present and supersaturations from 1–2% (1984 study) to water saturation (1978 study). Thus, the overhanging aspect of P-growth may require a larger-area rounded region as occurs on a larger droxtal. If $x_s$ depends on temperature as experiments suggest (Mason et al., 1963) or if $l_s$ depends on temperature, then droxtal pockets should also depend on temperature. Both quantities may also decrease with increasing supersaturation.

The F- and P- types of lateral growth here are driven by AST. Evidence for AST on ice is indirect, partly coming from early studies of spreading ice layers on covellite (Hallett, 1961; Mason et al., 1963, Kobayashi, 1967). In these studies, the rates of approaching micron-scale layers, also known as macrosteps (arising from clustering of smaller steps or contact between crystals of differing height) changed in a way consistent with a flux of molecules over the top edge of the layer. The AST concept has long been applied to the growth rates of metal whiskers (e.g., Sears, 1955; Avramov, 2007), but rarely applied to ice.  More
recent experiments on ice find evidence for the flux over the tops of much thinner layers (Asakawa et al. 2014). In both the macrostep cases (Kobayashi, 1967) and in many other observations of thinner layers (e.g., Gonda et al., 1990; 1994), the step-front is rough as determined by its rounded perimeter. The cause of this roughness may be the thermal roughening proposed by Frenkel (1945) and BCF (1951), but, as we consider later, the roughness may also involve other processes and apply to the growth-front of F- and P-growth.

For applications, earlier studies applied the concept of AST to the primary-habit change. Mason et al. (1963) considered it the main factor driving primary habit, but the specific mechanism they proposed has been criticized because it does not consider the role of critical supersaturation on the nucleation of new layers. Frank (1982) argued instead that AST should make the change of primary-habit with temperature more abrupt due to layer nucleation on one face hindering nucleation on the adjoining face. Yamashita (2015, 2016, 2019) has revived the general concept, expanding its applications to secondary features via lateral and
protruding growth.

Finally, to help clarify subsequent discussion, we use the following definitions:

- **Lateral facet growth:** areal growth on fully facetted faces, includes S-, F-, and P-types. At times, we shorten this to just "lateral growth", and the following four processes as collectively "lateral-growth processes".
- **Standard lateral growth (S-type):** areal growth of a facet bound by other facets, determined by their normal growth.
- **Facet spreading (F-type):** areal increase of a facet on a rounded surface, driven mainly by AST.
- **Protruding growth (P-type):** extending growth of thin, usually planar, face region that extends over adjoining regions, driven mainly by AST.
- **Adjoining surface transport (AST):** surface transport of mobile molecules from a face, over the edge of the face, to the adjoining region where the growth occurs.

This paper arose from two studies. In the first study, one of us (JN) had been examining images from earlier cloud-chamber experiments and images of precipitated snow with Prof. A. Yamashita of Osaka Kyoiku University, hereafter AY, exploring ideas about how AST may help explain some perplexing ice-crystal growth forms including pockets. Then, in a later study, both of us (JN and BS) began measuring normal growth rates in a newly developed chamber, but unexpectedly discovered corner pockets appearing on a thick plate after a brief sublimation period. Recognizing the connection to the first study, we ran similar
experiments, finding them to be reproducible and also revealing other types of pockets. We present our evidence and ideas here, with the goal of making a convincing case that such lateral-growth processes should be included in any complete ice-crystal growth model, particularly when modeling the more complex crystal features.

## 3 Methods

For this work, we used a new crystal-growth apparatus, hereafter CC2, that improves upon the first "capillary–chamber" method in Nelson and Knight (1996). Like that apparatus, the observed crystal hangs pendant on an ultra-thin glass capillary within an isothermal, stagnant atmosphere. But in CC2, the ambient supersaturation around the crystal is controlled by the surface temperature of one of two vapor-sources in its own adjoining chamber, the connection of which is controlled by a translatable valve stopper. Briefly, the vapor source (ice, pure melt, or solution) has a surface area vastly greater than that of the observed crystal on a capillary. Thus, except very near the observed crystal (when air is present), the vapor density throughout the system is the equilibrium value of the vapor source from which we calculate far-field supersaturation. With this system, we can grow and then sublimate a given crystal without changing the temperature surrounding the crystal. The temperatures of the vapor-source surfaces are controlled by a thermoelectric element below each vapor-source container. The block encasing all three chambers is made of gold-plated, high-conductivity Te-Cu of dimensions 3" x 5" x 7" and submerged in optically clear cooling fluid pumped with a Neslab ULT-80 circulating cooler. To start an experiment, we insert HPLC water into the vapor-source containers and the capillary. The source water and capillary are cooled to the desired temperature and frozen. We then monitor the crystal at the capillary tip using back illumination and a full-frame DLSR 24-megapixel tele-microscope-camera system in the front. For more details of this apparatus and method, see Swanson and Nelson (2019).

We report here images collected from CC2 during their growth as well as images of crystals grown by AY in a cloud chamber. The latter crystals were nucleated at the top of a tall (15-m) cloud chamber (Yamashita, 1971), fell while growing for about 3–4 minutes under relatively uniform conditions, and then collected post-growth in sub-zero silicone oil. Although they provide only a snapshot of a crystal's growth, the high-magnification imaging provides greater detail of the early growth stages as well as growth at higher supersaturations, thus complementing our CC2 results. In both the cloud chamber and CC2 experiments, the crystals grew in an atmosphere of air. Other crystal images were provided by Mark Cassino, Martin Schnaiter, and Art Rangno.

## 4 Observations and analyses

The following subsections survey observations made in CC2, including previously unreported "corner pockets", "planar pockets", and "elongated edge pockets".

### 4.1 Corner pockets on larger crystals during a growth−sublimation cycle

#### 4.1.1 Observations

In our CC2 experiments, we observed the appearance of 12 small pockets, one in each corner, after a thick prism crystal resumed growth after a period of sublimation. The crystal, shown in Fig. 3, remained at −29 °C with a supersaturation that began at about 0.5%, then spent less than an hour at a small negative value, then went back to about 0.5%.

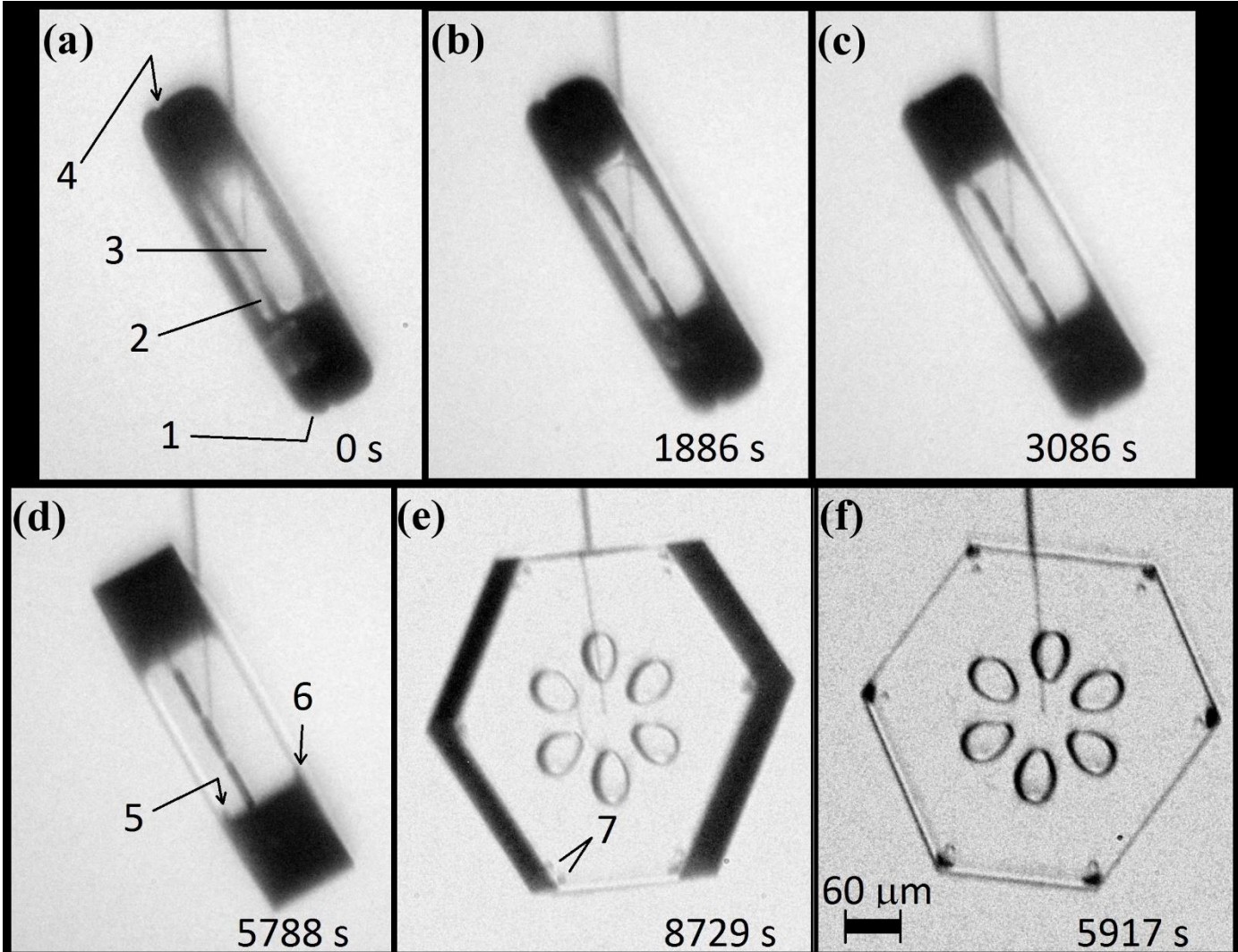

**Figure 3**: Corner-pocket formation at −29 °C after sublimation rounding. (a) Side-view of crystal at end of sublimation run. Marked point 1 shows a rounded basal-prism corner; 2, side view of interior planar air pockets; 3, view through two prism faces showing curved bounding edges (evidence from lack of sharpness); and 4, a perimeter groove bounding the same interior basal plane as the interior air pockets. (b)—(e) Subsequent sharpening of the basal-prism corner under growth conditions. Marked points 5 and 6 appear to show side views of the corner pockets. (e) The basal face partly turned into view, showing a corner-pocket pair near each prism–prism edge at 7. (f) Front view showing the 12 corner pockets (two pockets per prism-prism edge, each pocket near opposing basal faces). Line coming down from the top is the capillary, terminating in the crystal at the nucleation point. Supersaturation was constant at about 0.5%.

Consider the sequence in more detail. Figure 3a begins after the sublimation period, just as the growth condition has returned. The lack of sharpness viewed through opposite prism faces shows that the faces retain some curvature. At the edge, the rounding appears to have a radius of about 30 μm. As time elapses in (b)–(d), the boundaries sharpen, becoming fully facetted in (e) and showing six pairs of pockets near the corners in (f). The slightly rotated view in (e) shows that each pair consists of one pocket near each basal face (top and bottom), and these pockets may be barely discerned even in (d) at "5" and "6". This particular cycle is the second one we imposed on this crystal, but it shows the corner pockets more clearly than the first growth-sublimation sequence on the same crystal.

The corner pockets in this case occurred on a tabular crystal, but the tabular shape is not crucial to the pocket formation. In a case we consider later for a different phenomenon, we had 10 crystals of various aspect ratios, including a long column, all undergo a growth–sublimation–growth cycle, and all exhibited the corner pockets (e.g., on the nearly isometric crystals of Fig.

12b,d). All of the cases though have been on large crystals (~200–400 μm) at a temperature near −29 °C. In previous experiments (Nelson and Knight, 1998), we grew, sublimated, then grew crystals that were about 10x smaller (~15–40 μm) and at temperatures above −15 °C, yet never observed corner pockets. The literature shows cases that were not recognized as corner pockets. For example, similar corner pockets appear on a ~100 μm crystal studied by Kobayashi and Ohtake (1974) above −20 °C after a sublimation cycle. In that case, the radius of curvature at the corner was about 20 μm, but they show another case without corner pockets in which the corner radius was only about 10 μm. Also, Magono and Lee (1966) show a solid, thick plate (photo #30) with corner pockets. In this case, the crystal was about 150 μm across with a curvature at the corner near 20 μm adjacent to the upper basal. Near the lower basal, the curvature appeared a little smaller and the corner pockets were smaller. Thus, although the phenomenon can appear on a range of crystal shapes, the corner radius may need to exceed a certain value for the corner pockets to either exist or become resolvable with standard microscopy. At about one atmosphere pressure and temperatures near −20 and −30 °C, this critical radius may be between 10 and 20 μm, but the value may depend on temperature and pressure.

### 4.1.2 Basic mechanism

Existing views on normal growth via step motion cannot readily explain corner pockets on fully facetted crystals. With normal growth, each pocket must have at one time been a hollow (lacuna or concave feature) before closing-off to enclose the air. And standard hollowing theory (e.g., Kuroda et al., 1977; Frank, 1982; Nelson and Baker, 1996) predicts that hollows form around a local vapor-density minimum, not at a corner where the driving force for normal growth is instead a local maximum. Moreover, the standard theory relies upon step clumping on a facetted surface. We argue here that the pockets instead form via protruding growth adjacent to a rounded corner, similar to that in Fig. 2b',c'. But unlike the droxtal case, the rounding here came from sublimation.

Consider the stages in Fig. 4, with an oblique view at left and a cross-section through a corner at right. In (a), the crystal is a thick prism and fully facetted, representing a growth condition. In (b), the crystal has transitioned to a sublimation condition, thus rounding its corners and edges. Then, in (c), a growth condition resumes, causing the basal and prism facets to grow laterally, primarily via AST over the spreading edge-front where they bond. As the spreading edge becomes thicker (viewed in cross-section), this rate will slow because the same number of molecules must spread over a wider front region. This growing front becomes too wide in (d), and the AST flux of molecules builds up an overhang on the spreading facet edge, initiating protruding growth. Where the protrusions from two faces intercept, they merge, halting further protrusion there. This merging occurs first further back along the edge from the corner, but progresses to the corner at (e), sealing-off the corner pocket. Later, sublimation and deposition within the sealed-up pocket will round out its interior, making the pocket more spherical. This mechanism does not include normal growth because normal growth in the experiment was extremely low.

The case in Fig. 3 shows six dark corner pockets on one basal, six lighter pockets slightly further inside (radially) on the other basal. This difference may have arisen from having different degrees of initial rounding, or by one basal face having more basal-normal growth than the other. The side views show how the planar pockets are closer to the left basal face, indicating both that the left-side rounding may have a smaller radius and also that the right basal face has a greater normal growth rate. Further considerations of how normal growth may affect pocket formation is in §4.6.2.

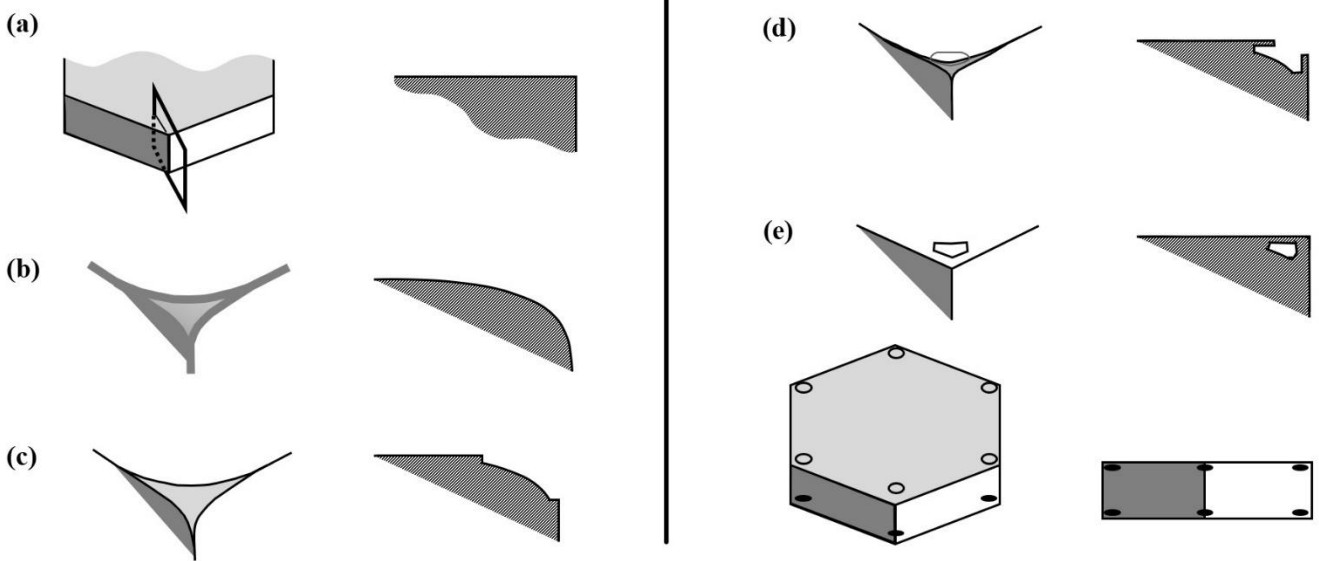

**Figure 4:** Corner-pocket formation after sublimation rounding. (a) Oblique (left) and cross-section (right) view of edge of a tabular crystal during growth. The top face is basal, the sides are prism. (b) Same views after net sublimation rounded the edge. (c) After growth conditions resume. Basal and prism have facet-edge fronts (same as Fig. 2b). (d) Protruding growth begins. (e) Corner pocket forms. Overall oblique and front view at bottom.

## 4.2 Planar pockets formed under constant conditions

The crystal in Fig. 3 exhibits another notable feature—its six thin, petal-shaped pockets. These "planar pockets" appeared well before the formation of the corner pockets, and did not require a sublimation event before formation. From the front (f), they appear typical of common center hollows (i.e., formed in face centers) that later closed up, but the side view (d) shows them to be unusually thin, or planar. That is, hollows often start by widening with a nearly circular rim shape (e.g., in hollow columns), whereas the hollows that preceded these planar pockets must have instead had a rim shape similar to a thick line segment before closing into pockets.

In Fig. 3a, the planar pockets appear to be in the same plane as the small notch marked "4". The notching suggests a disordered region, like the eroded region at the grain boundary near the center of bullet rosettes. However, the prism planes align on both sides of the notch, showing both sides have the same crystal orientation. Thus, the notch and plane must have a stacking fault, not a grain boundary, with the depth of the pockets suggesting that a region of faults may be present. Itoo (1953) called such crystals "twin prisms", and found them to be very common in a light precipitation at −30 °C. Kobayashi and Ohtake (1974) made a similar observation to that here, suggesting a specific type of stacking fault. A more recent study found that extended regions of stacking disorder are common when small water droplets freeze near −40 °C (Malkin et al., 2012), but are unlikely to form during vapor growth (Hudait and Molinero, 2016). The crystal of Fig. 3 began with a freezing event at the tip of the capillary, where the apparent stacking-disorder region intercepts, and then grew from the vapor. Thus, the argument for the source of the notch and planar pockets is consistent with these recent studies.

Another distinctive feature of these pockets is their near-perfect six-fold symmetry. Such symmetry of both the pockets and the crystal is unusual for a crystal grown at such low supersaturation. More typical cases for low supersaturation are shown in Figs.

7,9 and the literature (e.g., Nelson and Knight, 1996; Gonda, Sei, and Gomi, 1984). Reasons for their symmetry and their closing-off are argued in Appendix B.2.

## 4.3 Facet spreading on the basal face

### 4.3.1 Observations

In some crystals, we can observe the spreading of the basal facet when the partly sublimated crystal begins to grow. For example, the sequence in Fig. 5b–d shows an expanding ring on the basal face (though the exact position is harder to discern in (b)). The temperature and supersaturation were about −30 °C and 1%. When this ring reaches the perimeter, the crystal appears fully facetted and the corner pockets appear (arrows in (e)). Thus, the rings mark the expanding boundary of the basal face (as opposed to a macro-step on a growing face). The positions of these rings, simply estimated by eye, are marked in (f), with the

time interval (units of 5 min) between marked positions in the upper right. The markings show a significant slowdown as the facet perimeter approaches the crystal perimeter, and in this process, the facet perimeter becomes more distinct. The latter observation is consistent with a thicker height difference $h$ between the rounded surface and the facet upon reaching the perimeter, consistent with having a rounded edge, and lateral growth driven by AST. Also, one can see that the prism–prism edges appear to sharpen by (d), before the basal face fully spreads out. We saw similar behavior in other cases.

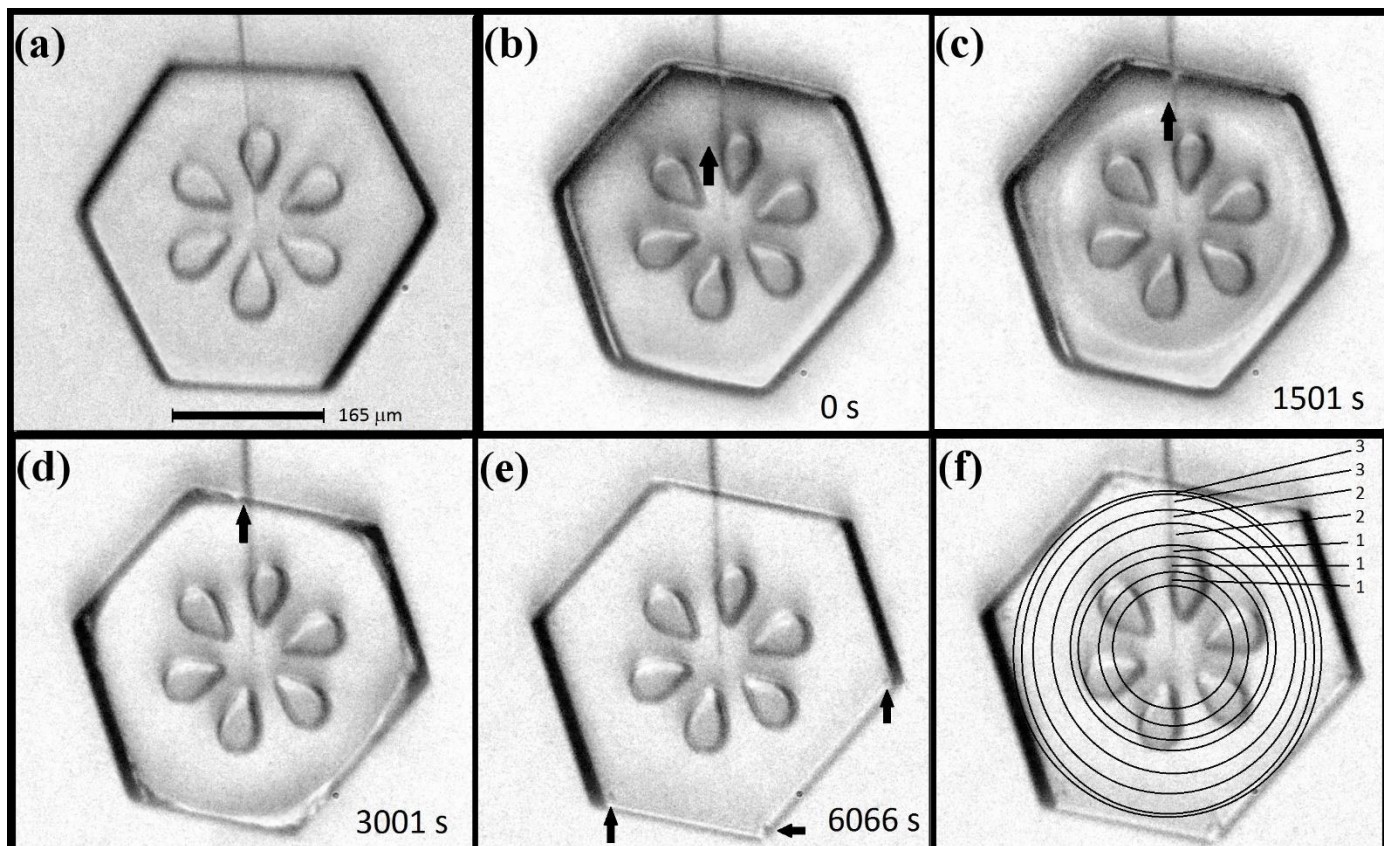

**Figure 5:** Expanding basal facet with corner pocket formation. (a) Crystal just before sublimation. (b)–(e) Same crystal, during a second growth period at −30 °C and 1% supersaturation, just after the sublimation period. The thick arrow marks a barely visible boundary of the basal facet, roughly forming a circle. In (c), two such circles can be seen, representing the boundaries on both

basal facets. The times of the images are in their bottom right corner. In (e), arrows mark three corner pockets. (f) Same image as (e), but with the estimated basal perimeters sketched with circles. Numbers at right are the times between the perimeter sketches in units of 5 minutes. Data is plotted in Fig. 6.

### 4.3.2 Test of AST-driven facet spreading

To test the AST-driven facet-spreading mechanism, we ran calculations for three possible mechanisms of lateral growth. Results are in Fig. 6. The first model (marked I), is normal growth of the lateral-growth front (i.e., **e–c** in Fig. 1b) driven by direct vapor flux. This case shows a resulting advance about two-orders of magnitude too slow. Also, the trend, which can only
be seen with a much higher supersaturation (not shown), does not capture the slowdown that begins within about 1000 s of the start. Model II is the AST-driven case, and this fits the data well provided that the calculation uses the inset trend of $h/x_s$ (normalized height of lateral-growth front). This profile of the growth-front height $h$ is difficult to compare to the crystal, as it requires frequent side views of the crystal that we did not obtain, but it is a reasonable fit to the initial cross-section profile. This profile is that of a flat facet out to a radius $r < a$, and a curved profile between $r$ and $a$ where the crystal had rounded during
sublimation. (Refer to Fig. A1 for further details.) A reasonable estimate of height $h$ upon reaching the edge is 1–5 μm. With this range, the fit in Fig. 6 (inset) predicts $h/x_s = 0.3$, giving $x_s$ = 3–17 μm at this temperature, which is comparable to the value of about 2 μm found by Mason et al. (1963). Model III is an approximate rate based on normal growth of the rough region beyond the lateral-growth front. It does not fit the data well, but is better than case I. Also, case III is sensitive to the profile of the rough region. Thus, the failure to fit the curve may be partly due to profile inaccuracy. Appendix A has details of all three model
calculations. A better test of the lateral-growth mechanism requires better data, such as interferometry data (e.g., Shimada and Ohtake, 2016) and possibly a model that includes processes in both mechanisms II and III.

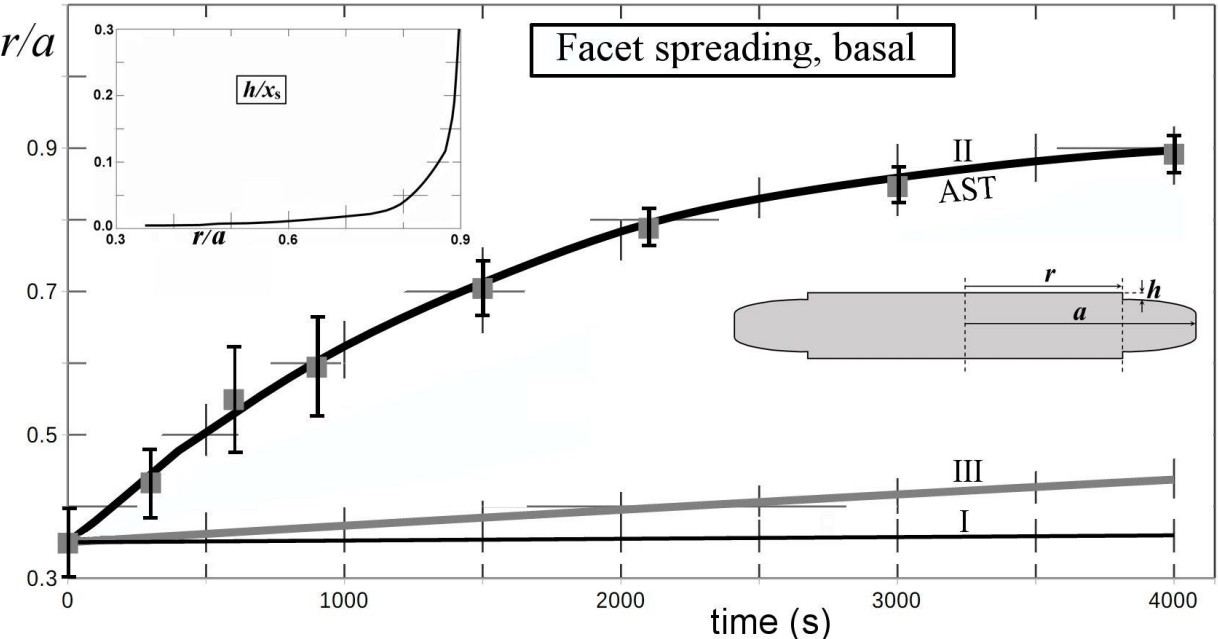

**Figure 6:** Facet spreading of basal from Fig. 5 (solid grey squares, error bars) with model fits I–III from Appendix A (curves). (I):
Normal growth of the facet edge. (II): Facet spreading from AST. (III): Normal growth of the rough, rounded region. Assumed supersaturation of 1% and temperature of −30 °C. For calculation details, see Appendix A. Crystal radius $a$ is the mean value out to the prism–prism edge. Hatch marks are truncated gridlines. Inset plot shows values of facet-edge height $h$ used in the fit for case II, $x_s$ is the surface migration distance. Inset sketch shows cross-section, basal faces top and bottom, with plotted variables.

Nevertheless, the observed behavior clearly shows that mechanisms I and III cannot explain the observed facet spreading. Only growth driven by a flux of surface mobile molecules, the AST mechanism, from the facet to the lateral-growth front is capable of fitting the observations.

## 4.4 Corner pockets on a non-symmetric thick plate

In another case, we ran a growth–sublimation–growth cycle on a tabular prism at −30 °C with unequal prism faces. In this case, shown in Fig. 7, the initial crystal in (b) is more rounded than that in the previous case, with a radius of ~30–40 μm. After regrowth (supersaturation below 1%), the facetted crystal emerged with larger corner pockets that are elongated along the edges (e). And, as with the previous case, the spreading of the basal facet slows down upon nearing the edge in (b) to (d). Later in the growth, in (e), a large basal hollow appears. But the larger size of the corner pockets in this case compared to those in the cases in Figs. 3 and 4 is consistent with the pocket size being larger for cases with larger initial corner radii.

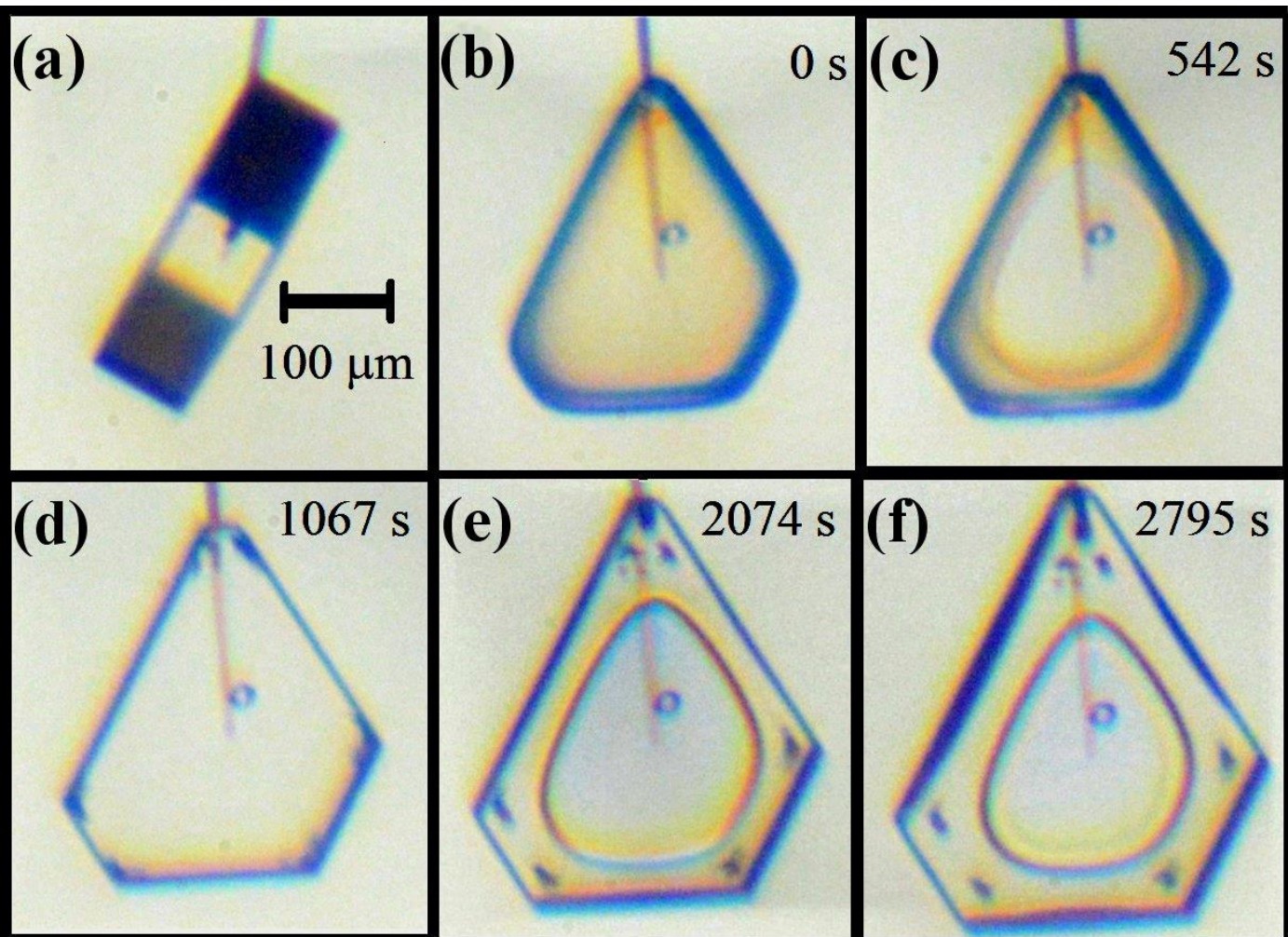

**Figure 7**: Corner-pocket and center hollow formation and changes on a non-symmetric crystal (−30 °C, 0.8% supersaturation). (a) About 2 h before sublimation. (b) Immediately after sublimation, growth period starting. (c) Basal facet spreading. (d) Clear corner pockets formed. (e) After normal growth, a center hollow on one basal and on top prism. (f) Further growth, some hollowing starting on wider prism faces.

## 4.5 Corner pockets on naturally formed crystals

Corner pockets such as those described here also appear on natural snow and ice crystals. The center of the snow crystal in Fig. 8a, collected and photographed at the ground, shows pockets "CP" in the corners of the central plate. Case (b) shows apparent corner pockets in a thin, solid tabular prism collected in-cloud. In (c), we see six pocket pairs near the center of another collected snow crystal. The mechanism in this case may differ from that in (a) because they appear on a two-level crystal. We also show other pockets further up a main ridge in the smaller inset at bottom. Thus, these do not appear to be the same corner pockets that we have discussed above. This type is discussed later, in Appendix B.4. For the cases in (a) and (b), we conclude that the crystals likely underwent a sublimation period to produce the corner pockets.

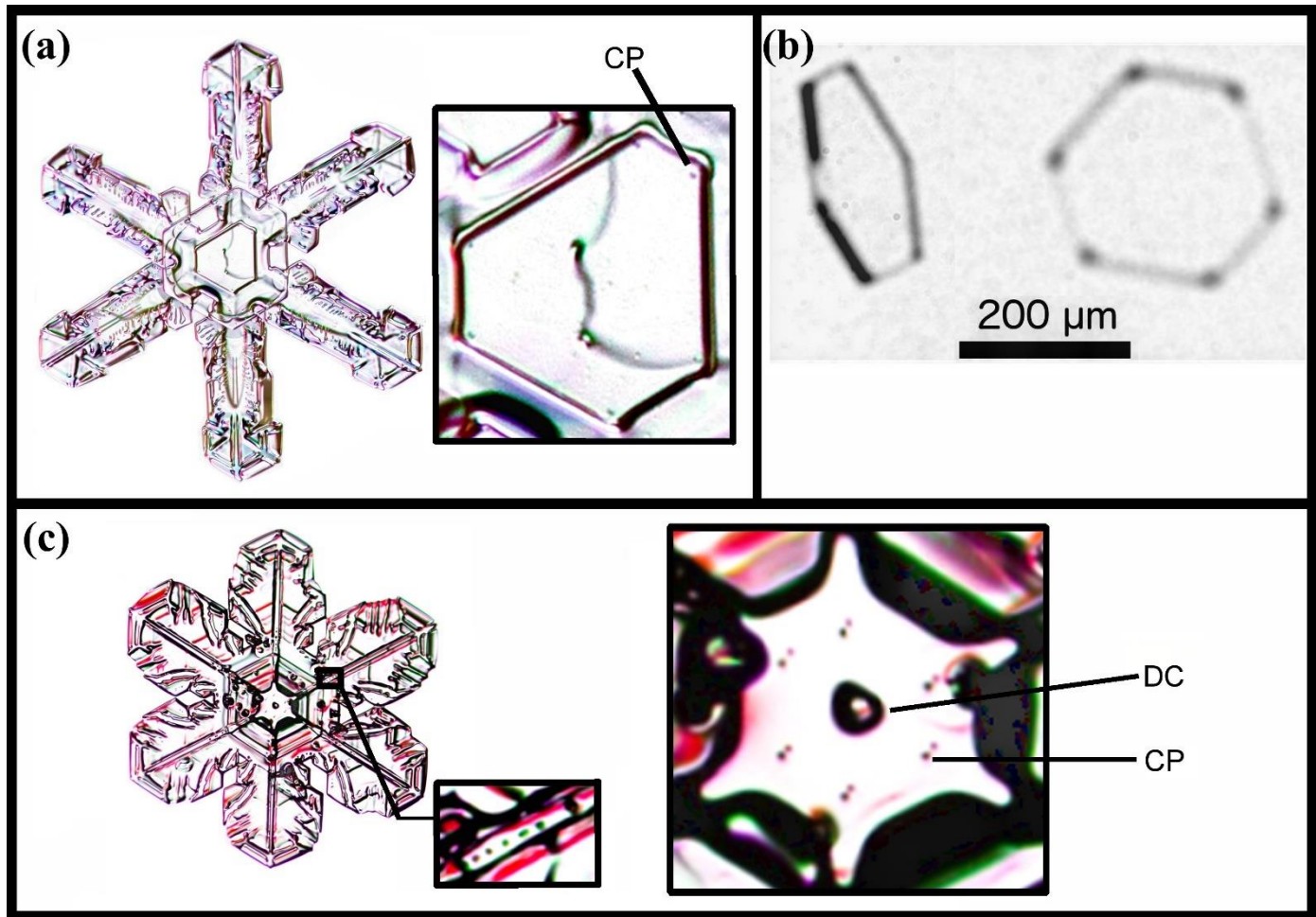

**Figure 8:** Corner pockets (CP) and droxtal centers (DC) on natural snow and ice crystals. (a) A narrow broad-branch crystal. (Curved line through crystal center is an imperfection on the glass slide.) (b) Crystal collected on the SOCRATES mission at -24.9 °C, viewed at two angles. Image courtesy of Martin Schnaiter. (c) Case of a wide broad-branch, large inset at right showing a close-up of center, small inset shows pockets along one ridge (nearly identical to pockets along the other five main ridges). Snow crystal images in (a), (c) courtesy of Mark Cassino.

## 4.6 Lateral growth on the prism faces and elongated edge pockets

### 4.6.1 Observations

Corner pockets vary in size and shape, with those in Fig. 7 larger and longer along the edge than those in Figs. 3 and 5. This elongation can extend along the edge, traversing nearly the entire edge, a case we call "elongated edge pockets". We show one example in Fig. 9. It begins from a sublimated, rounded form at 0 s. After 180 s, small prism facets started to appear (not shown).

These facets grow both normally and laterally as the other facets become defined. At 541 s, the edge at 'A', as well as the edges of face 1, extend slightly above the plane of the adjacent faces. By 1083 s, some normal growth can be discerned. From 1444 s, the two opposing edges of faces 2 and 3 become clear, and these edges approach each other at 'C' (2138 s), appearing to be facets. These two facets completely merge before 8448 s. Later, the final front and side view shows that this edge region has a

5 long pocket along this prism–prism edge marked 'E'. Thus, the merging of two lateral-growth regions created an elongated edge pocket between prism faces. As this is the only case we observed, it is hard to strongly argue a particular cause. One potentially important distinction from other crystals with unusual pockets is the greater amount of normal growth in this case. We account for such normal growth by including S-type lateral growth along with the P-type in a possible mechanism argued in the next section.

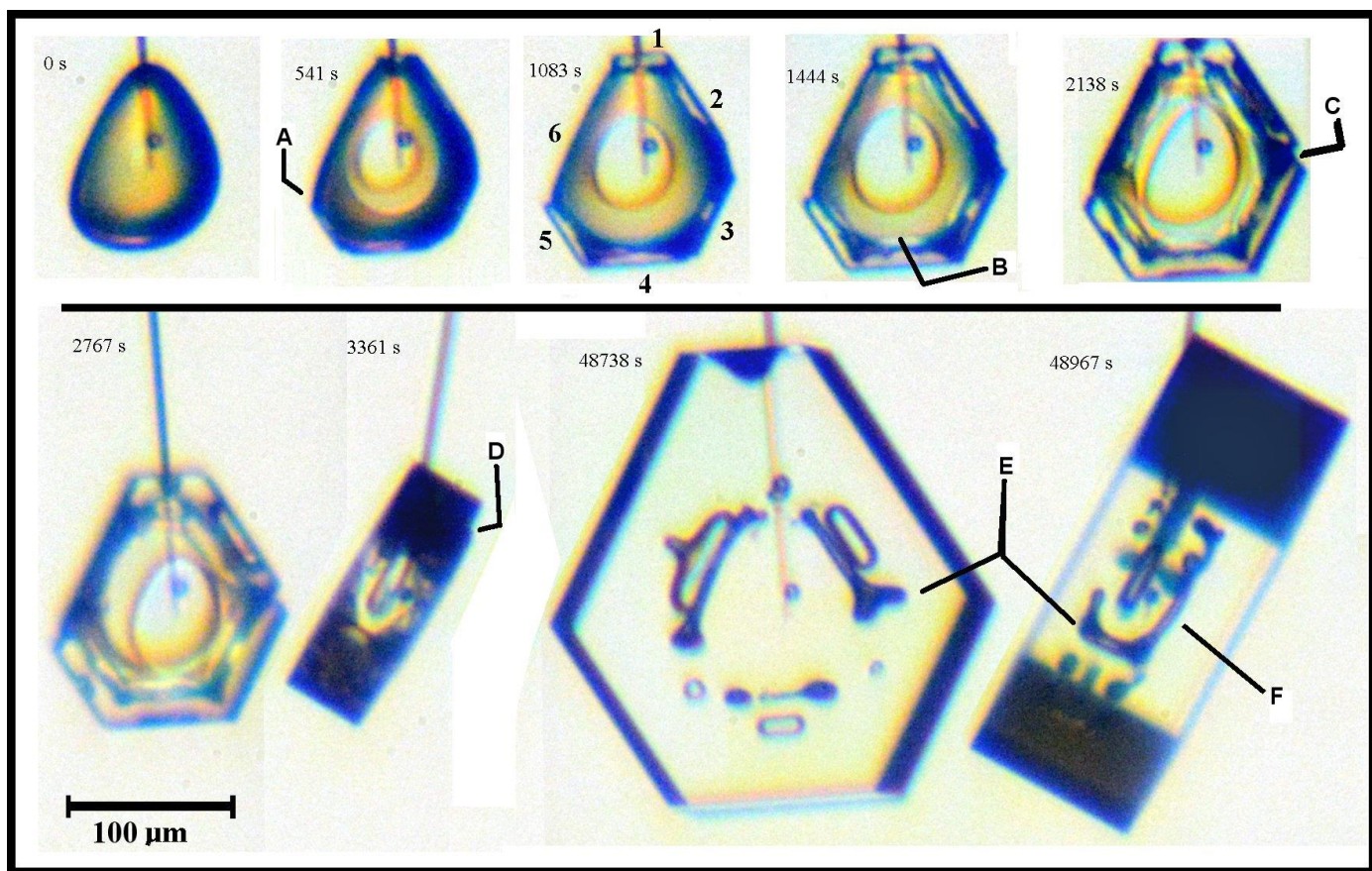

**Figure 9:** Elongated edge pockets and lateral face growth on a complex crystal growing under constant conditions (−30 °C, ~1% supersaturation). Time 0 s is just after sublimation, the crystal just starting to grow. At 541 s, **A** marks the edge of prism face 5, growing laterally over face 6. The prism faces are filled out by 1083 s and numbered clockwise from top. Face 5 growing

15 laterally over face 6. The **B** at 1444 s shows a boundary with straight edge. **C** at 2138 s shows two adjacent prism faces closing up via lateral growth (completed before 8448 s, leaving an elongated edge pocket). At 3361 s, **D** marks an interior edge of a thick layer on the bottom basal face. **E** marks two views of an edge pocket between prism faces (same as that tracked by **C**). **F** is an elongated edge pocket between the bottom basal face and prism face 2.

Such merging of straight-edged sections may be occurring on the basal face as well. By 1444 s, dark regions appear along

20 basal-prism edges, suggesting that the corners are connected by long pockets. Such an edge pocket is confirmed and marked 'F' in the final side view. However, unlike the prism–prism-edge case, the lateral growth involved in this feature's formation is unclear. Standard S-type lateral spreading of a thick layer on the old basal face, with edge boundaries parallel to the basal–prism edge, may explain this edge pocket. Two indications that such a thick layer may have spread as such are marked as 'B' and 'D'.

The dendrite in Fig. B13c shows similar edge pockets at sidebranch D, but the formation conditions are likely different. Nevertheless, the formation of elongated edge pockets in both cases likely require protruding growth even if the details of the mechanism differ.

### 4.6.2 Mechanism of edge, elongated-edge, and edge-pair pockets

5    The formation of edge and elongated-edge pockets should be similar to that of the corner pockets. For the elongated-edge pockets in Fig. 9, one difference from the corner pocket case is that the advancing front of the laterally growing facet is straight and parallel to the crystal edge. In the view marked 'C' in Fig. 9, these fronts appear to be prism facets indicating S-type lateral growth. Another difference may be the higher normal-growth rate (though still quite low). These differences suggest the mechanisms in Fig. 10.

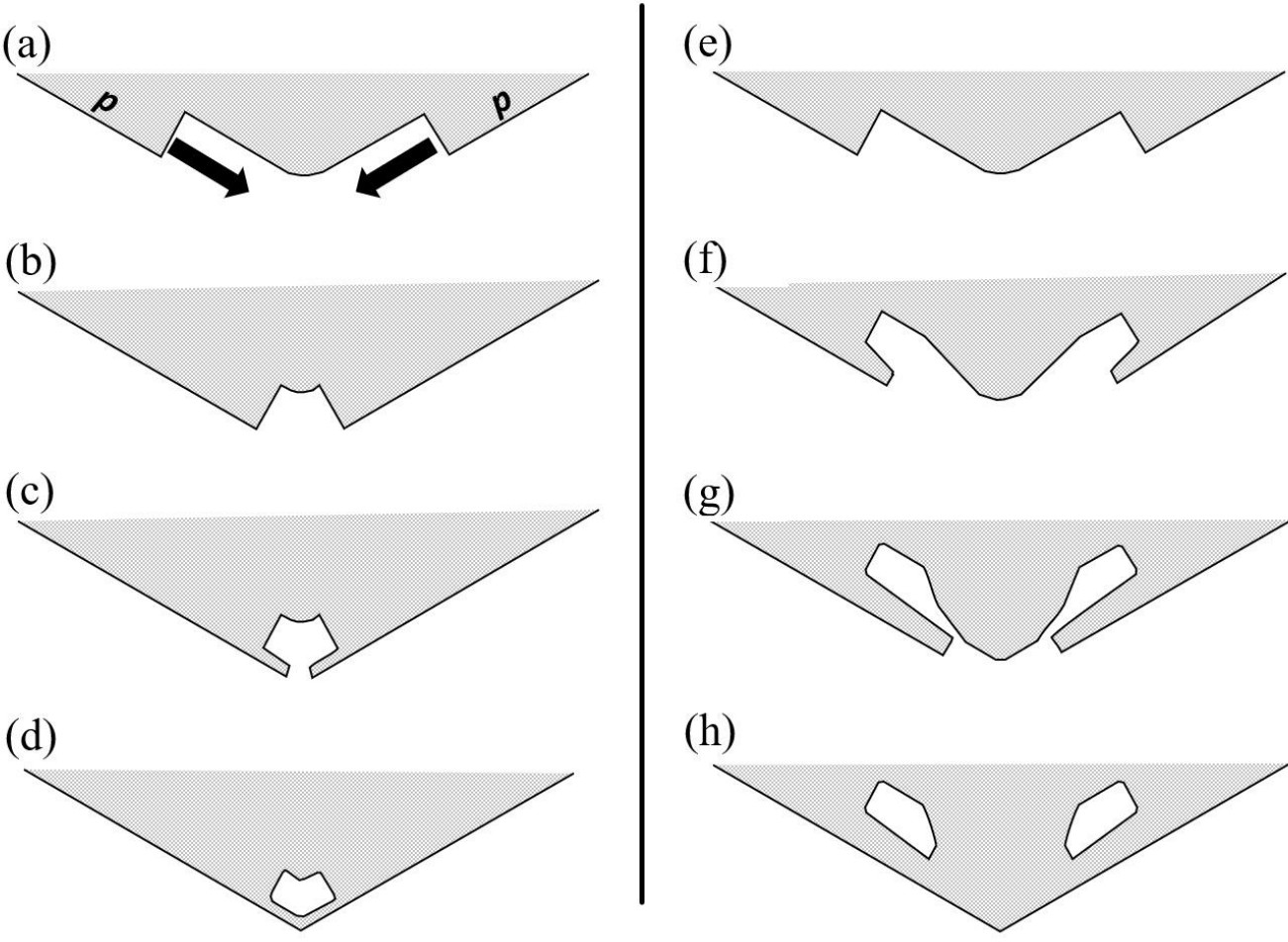

**Figure 10:** Edge-type pocket formation between prism faces. (a) Each prism face has a large advancing front or side-face growing laterally towards the edge. This lateral growth is marked by the solid arrows and is driven by both direct vapor flux and the AST flux. (b) The two large fronts are close enough to effectively 'shadow' the inside edge from the vapor flux. (c) Vapor gradients along front leads to protruding growth, driven by AST. (d) Protrusions merge, making an elongated edge pocket. Case (e)–(h) is similar except with greater normal growth. The two side fronts may also be non-crystallographic as in scallop-type rounded growth.

In Fig. 10a, the two new prism facets converge on an existing prism–prism edge. Their advancing fronts may be prism faces

20   (as in Fig. 9) or non-crystallographic. For a thicker, facetted front, the two fronts are cases of S-type lateral growth that intersect,

with their motion initially driven by both AST and direct vapor deposition to the front. But when the two fronts converge, the interior region would get increasingly shielded and shut-out from vapor (b) at the same time that the front height increases (the rounded edge means the base of the front recedes, increasing the front height). Thus, the AST should eventually dominate, producing two protrusions (c). Upon merging, they leave a pocket parallel to the edge (d). This pocket may be nearly equidimensional for an edge pocket on a thin tabular crystal, and elongated if the prism–prism edge is long. This enclosure would then be completely sealed up by protruding growth on the basal faces (not shown).

Near the edge, the advancing fronts may generate pockets before converging, generating a pair of pockets instead of one. Figure 10e–h shows such a process. Although all stages in this process have not been observed, Libbrecht (2003) shows a double-edge-pocket case in a thin plate grown at −15 °C, and Knight (2012, Fig. 3c) appears to show some that are more widely spaced at −5 °C. Bentley (1924) shows several cases (e.g., his figures 6, 32). Such cases may arise when even greater normal growth occurs with the protruding growth as sketched in Fig. 10e–h. Although the normal vapor flux may compete with the AST flux, it may also create vapor-density gradients that can favor protrusion formation on one face versus another. For example, if the case in Fig. 10e–h represents a thin plate, the vapor-density gradients (discussed in Appendix B.1) would favor initiation of protrusions on the prism faces as shown, but not necessarily from the AST flux from the basal. However, as argued in Appendix B.7, the AST flux from the basal should be larger for points nearly $\sim x_s$ back from the tip. Thus, the AST flux from the basal could produce protrusive growth away from the corner, but not at the exact corner. Thus, the corner can fill-out as shown due to both normal flux and AST flux from the basal. The result is a pair of pockets as shown in (h). This process requires that the initial stage (e) have a rounded prism–prism edge. Knight (2012) observed that the thin plates often began rounded, and scallopped, lacking any prims faces, and later became fully facetted plates (see Appendix B.7 for similar cases). Thus, this mechanism does not require a period of sublimation rounding.

### 4.7 Hollow close-off to center pockets and terracing

Under a wide range of growth conditions, a small hollow may form in the center of one or more crystal faces. Once such a "center hollow" begins, it can enlarge (in width) as it grows, eventually overtaking most of the face, or it can vary in width, perhaps even closing-off. In the case that the width enlarges, the hollow deepens and develops structure. Figure 11 has a crystal showing some hollows that oscillate in width and some hollows that close-off. The hollows are just forming at 4419 s on the center of the prism faces, as shown in (a), with wider hollows on the wider faces. But by 8210 s (b), different hollows have changed differently. On prism face "1", as marked in (a), the hollow has remained small. On face "2", the hollow width suddenly increased at some time between 4419 and 8210 s, but is now decreasing in width (i.e., the rim radius is narrower than that just inside the hollow). This sudden increase in hollow rim-size creates a flat terrace-like feature in the hollow marked "t", so we refer to this as hollow terracing. The initial formation of hollows on faces 2, 4, and 6 are also flat, consistent with their later terracing. On faces 3, 4, and 5, the hollows are gradually closing up, again, with the rim leading the way. Face 6 displays behavior like that of 2, except the hollow widths are more clearly decreasing before abruptly increasing. In addition to these, a basal face has a wide hollow in its center that slightly decreases in width from (a) to (b). All these trends continue for at least another 8000 s in (c), with the bottom three prism faces (3, 4, 5) now completely sealed center pockets (marked "c" on 3 and 4). The side view in (d) shows how just the basal face on the left (facing up) has hollowed, but has narrowed at the rim from (b) to (c).

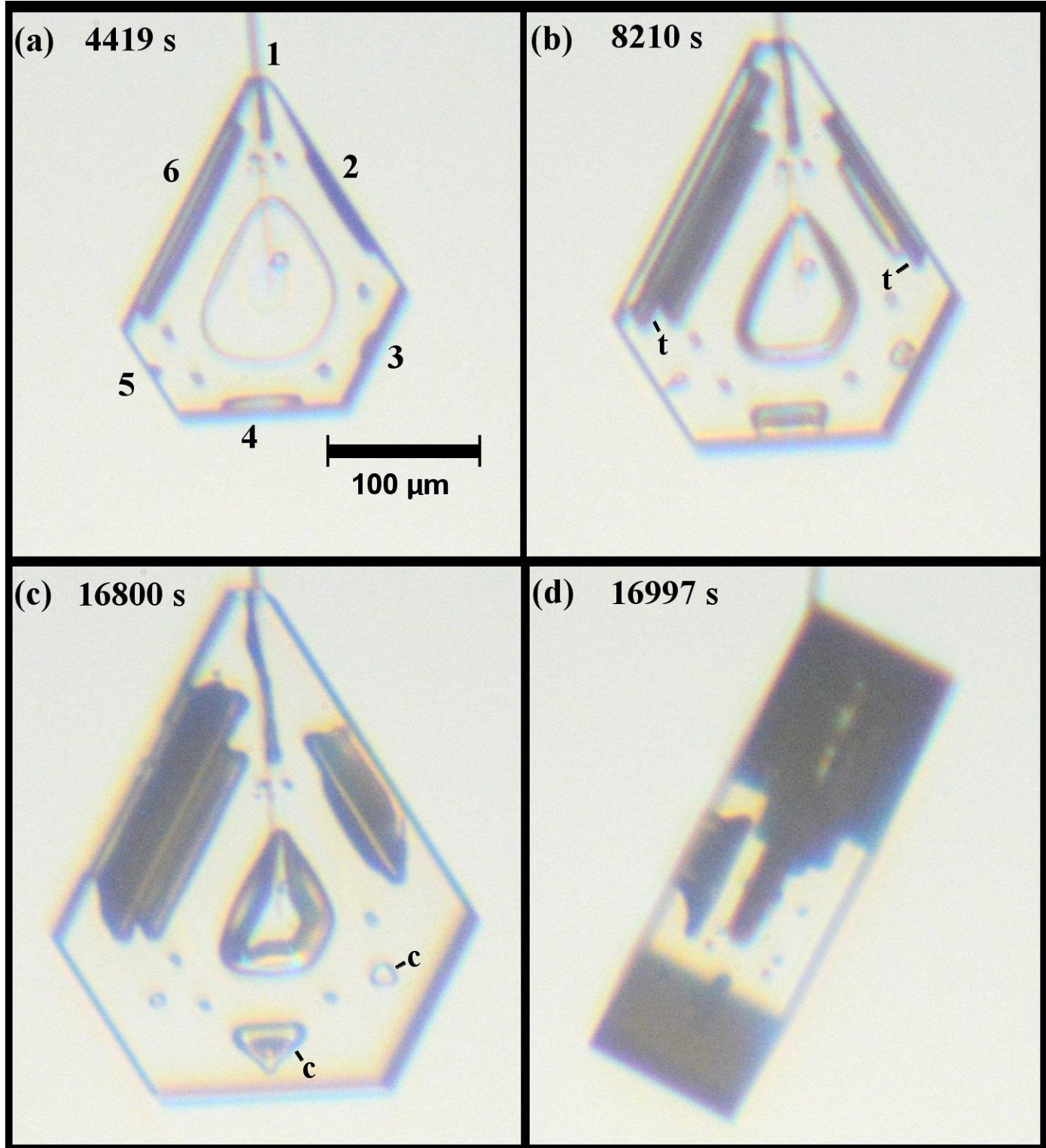

**Figure 11:** Center hollow development and center pocket formation. The crystal is the same as that in Fig. 7, but at later times under the same conditions. The scale in (a) applies to all images. Numbers in (a) label the six prism faces. "t" in (b) marks terraces, "c" in (c) marks two center pockets. Image (d) is a side view.

This oscillating-width nature of some of the prism hollows also occurs with basal hollows. In Fig. 12, we show two cases. In the top row, initially, in (a), the basal faces have hollows that 'fan open' at their start (e.g., "f" on upper left face), that is, have an increasing width during growth, but then later have nearly straight sides, indicating a constant rim diameter of the hollows. Soon

thereafter, the hollow rim suddenly widens, forming a terrace feature in (b) marked "t". A similar progression occurs in the crystal in the bottom row, with two such terraces forming on the face on the right in (d). Except for a brief sublimation period (note the small corner pockets), the growth conditions remained constant throughout the 47 hours of growth. Factors influencing the pockets and terraces in Figs. 11, 12 are likely complex, but in Appendix B.1 we suggest a simple model involving F- and P-growth to help explain them.

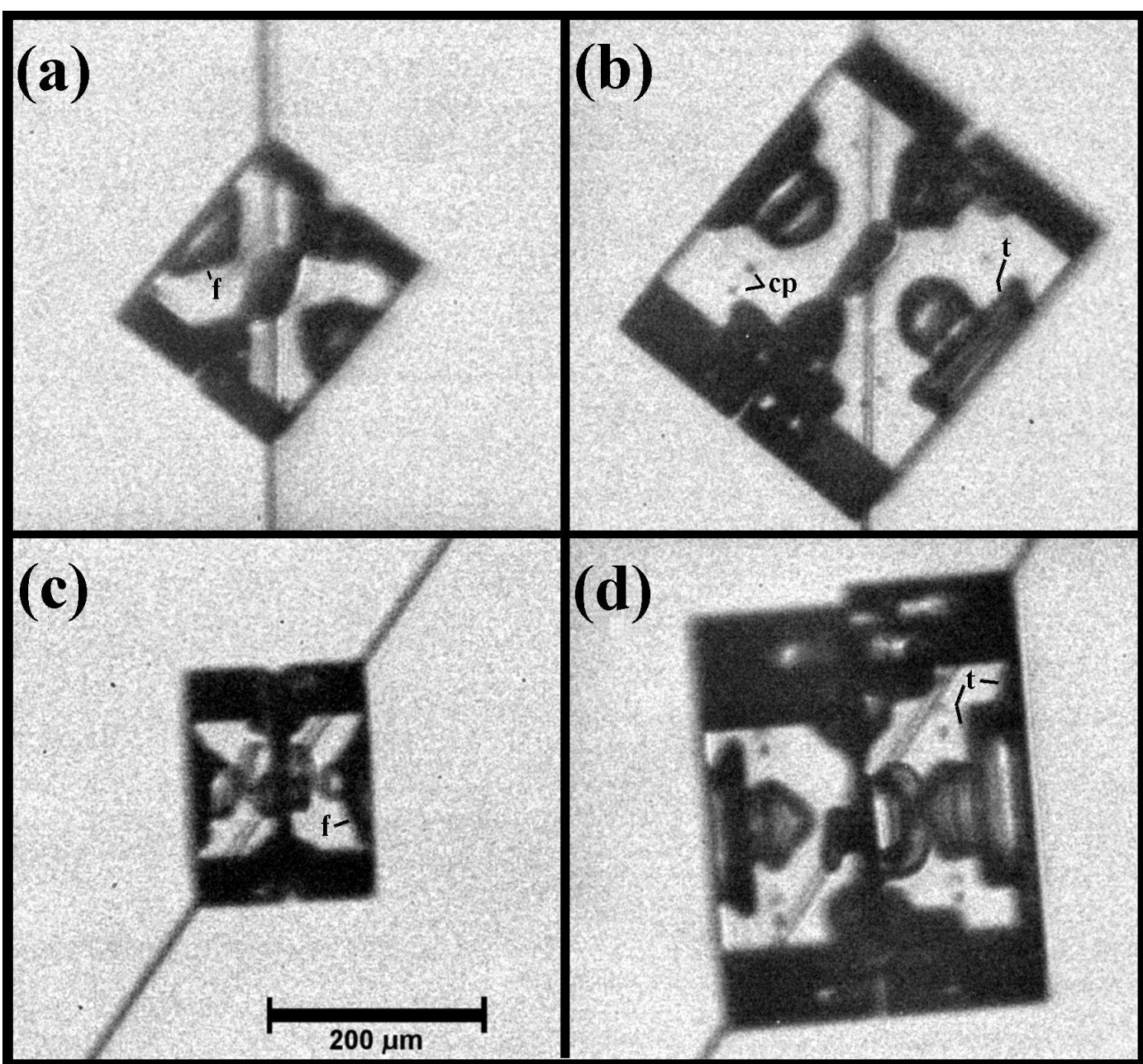

**Figure 12:** Center hollow terracing on twinned crystals grown at about −30 °C and 0.5% supersaturation. (a) Side view of crystal in middle of one capillary, basal faces pointing NW and SE. "f" marks "fanning-out" structure of a center hollow (ditto in (c)). (b) Crystal in (a), but 46 hours later. "t" marks terrace feature (ditto for (d)). (c) Different crystal under the same conditions, but on a different capillary. (d) Same crystal as (c), but 47 hours later. The small corner pockets (e.g., "cp") appear in (b) and (d) due to a brief sublimation period after images in (a) and (c). Scale in (c) applies to all images.

# 5 Discussion

## 5.1 Microscale mechanisms of facet spreading and protruding growth

The microscale mechanism of facet spreading involves AST. Our only requirement of AST is that it involves the migration of molecules, first over the edge of the facet, and second with their finding a high-density of growth sites on the other side. The first may occur via isolated molecules or as a more cooperative phenomena in a thicker disordered region (e.g., QLL), but either case may be consistent with the observations here. The second argues that the direction of this AST flux will largely be towards the side with the greater density of growth sites. The observations in Figs. 1, 5 show this lateral-growth front to be rough, thus indicating that the flux should be to this front. The reason for the roughness is not clear, but may be partly a result of the thermal roughening analyzed in BCF (1951). BCF argued that single steps should be rough, but surfaces should be flat except at or above their roughening temperature. Observations of steps on ice show them to be rough, even when collected into macrosteps (e.g., Kobayashi, 1967). The lateral-growth fronts may be rough for a similar reason even though they advance on a rough surface, not a facet, and may be thicker. That they may be significantly thicker than single steps might be connected to thermal roughening of facets. There have been reports of a roughening temperature near −2 °C (e.g., Elbaum, 1991), and thus ice facets at even lower temperatures may be close enough to roughening that macrosteps and other thin crystal regions such as the lateral-growth front can be rough even though larger faces remain facetted. Other causes of roughening are discussed below and in Appendix B.7.

For the microscale mechanism of protruding growth, two obvious questions arise: (1) How does a thin protrusion start? That is, instead of the AST molecules spreading out on the adjoining surface region to build-up a thick facet, why is the flux concentrated to a thin region? (2) As with F-growth above, why would the thin front of the protrusion have a high density of growth sites that can efficiently collect all the AST flux and continue protruding?

A possible answer to protrusion initiation (1) is a large facet-normal vapor-density gradient. Consider the qualitative features of the vapor-density contours as sketched in Fig. 2 (middle row, right side). This sketch is for the droxtal case, but it should also apply generally. Far from the surface, the contours are spheres, or circles in cross-section as shown as curve **1**. If the crystal was a roughened sphere, the contour curves nearer the surface would also be circles, but closer together, giving a radial gradient that is normal to the surface and strongest at the surface. But near a facet edge **m**–**e**, the contour curves bend such that further from the edge **e**, the normal gradient is zero (assuming zero normal growth) yet has a non-zero lateral gradient as shown by curve **3**. Right at the edge **e**, as well as near the roughened region beyond **c**, the contours are more nearly like that of the roughened spheres: nearly tracking the curvature of the surface. As a result, the vapor density at the surface rapidly decreases between **e** and **c** as shown by curves **3**–**5**. In such a case, the AST flux can build up nearer to **e** and not reach **c**, initiating the protrusion. Implicit in this argument is that sufficient air is present that the vapor mean-free-path is less than the distance **e**–**c** (otherwise the vapor density would have no appreciable gradients). Consistent with this argument is the observation that no cases of the corner pockets have been reported for small crystals and on crystals grown and sublimated in a pure vapor where such gradients are likely insignificant. Regardless, if one instead argues more generally that if we have a mechanism that answers (2), forming a high-density of growth sites in a thin region just over the edge of a facet, then a net flow of mobile surface molecules would not migrate any further than this thin region. If this migration on the rough surface has length scale $l_r$, then a region of thickness $t \sim l_r$ would start protruding. Thus, it becomes even more important to find a possible mechanism that answers (2); that is, why the edge is rough. Rough edges on thin-face regions have been observed in numerous cases as discussed in Appendix B.7. Thus, rough, thin protrusions may form and produce fast growth rates. However, it is not clear why only thin, and not also thick, protrusions would be rough.

A possible answer was proposed by Libbrecht (2003), who argued that thin plates must have a different structure at their leading fronts that leads to a high deposition coefficient (i.e., a high density of growth sites such as a rough edge), and then

suggested a type of nanoscale surface-melting effect. However, at nanometer sizes, the small radius of curvature may also increase the rate of sublimation, causing a compensating decrease in lateral growth rate. And though such a mechanism may help explain the fast-growing serrated dendrites at −2.0 °C and thin discs at slightly lower temperatures, it would be less likely at much lower temperatures, such as for the corner pockets observed here near −30.0 °C. Another possible answer is that the edge region consists of rough, high-index planes that essentially vanish on larger surfaces due to their rapid growth but cannot vanish on a thin protrusion due to a curvature effect. Other possible factors are considered in Appendix B.7, but clearly more experiments are needed to understand the mechanism of protruding growth as well as relations between thickness, roughness, and temperature.

## 5.2 General implications

### 5.2.1 How AST may help explain secondary features and habits

Ice growth in the atmosphere is affected by many processes. A first step towards an understanding is to identify which processes may play the dominant role in a given situation. We found above that AST appears crucial to understanding the observed facet spreading as well as the formation of corner and edge pockets. Although these exact situations may rarely occur in the atmosphere, AST itself cannot be "turned off", and thus AST-driven phenomena may have a key role in other situations as well. And as it turns out, there are numerous features on atmospheric ice crystals that are routinely observed yet have no clear explanation. Here we consider some of these features, proposing explanations that include F- and P-growth driven by AST. Detailed discussion and diagrams are in Appendix B.

**Center pockets** A center pocket is a center hollow that has closed-up. The closing-up involves growth lateral to the face in which the hollow sits, similar to that modeled in Fig. 6. As we found, the rate of the lateral growth via AST in that case was much faster than normal growth, making the closing up of hollows into center pockets a likely consequence of AST-driven P-growth. Such pockets are more common at lower normal growth rates, a situation that can allow such protruding growth to occur from opposite directions of a given facet. The normal-growth process, in contrast, does not explain the closing-up because the vapor impingement would be increasingly impeded in the narrow crevice region that closes up.

**Terracing and banding in hollows** Even when the hollow cannot close-up, facet spreading can occur on the inside surface of the hollows. When such spreading starts in a given region, adjoining regions sharpen. The vapor-density gradients near the sharpened region can then influence the facet spreading such as to amplify the effect. This may be the cause of the wide terraces in hopper crystals and the "band-like" lines in narrow hollow columns.

**Two-level planes with center droxtals** The center circle in some branched, tabular snow crystals around −15 °C have long been identified with the crystal's original droxtal, but it has never been clear how the circular form could remain as the droxtal grew outward via normal growth. Protruding growth from both basal planes has been observed in larger droxtals and likely also occurs in smaller droxtals such as those that are more common seeds of snow crystals. With AST-driven protruding growth of the basal faces, the basal planes can extend over the middle of the droxtal, leaving it behind in a vapor-shielded, still circular, region that hardly grows. This process likely explains the observed center droxtals.

**Capped and multiple-capped columns** When a columnar crystal moves into a temperature at which tabular crystals form, thin tabular extensions develop at the ends. These ends appear to start very thin, a situation in which AST should significantly contribute to their growth. Another driving process is likely the large supersaturation gradients near the tip, which may help to initiate the thin plates at the ends. But initiation of interior plates in the multiple-capped columns are much harder to explain without AST. Here, the appearance of a small basal face is all that is needed: AST from that face drives a small protrusion, and

as the face grows, the rate increases due to the larger collection area and the protrusion extending into a region of higher vapor density. In this way, a large plate can develop, even when starting from the side of a small rime droxtal.

**Scrolls** The scroll form involves prism faces that bend inward, extending in the directions of the a-axes and c-axis. If this was due to normal growth, then why would a face bend? Instead, their appearance suggests lateral growth in a prism plane, driven by AST. Once the thin plane thickens such as to be no longer primarily driven by AST, a new prism plane forms. Once a new prism plane forms, the extending growth changes direction such as to extend the new prism plane inward. In this way, the prism face bends inward, resembling a scroll. Nakaya et al. (1958) shows the form appearing around −9 °C, at relatively high supersaturation, consistent with that reported by Magono and Lee (1966). This temperature is close to that at which they reported the columnar habit changing to tabular. Scrolls also form below −20 °C, also being part of some polycrystalline types.

**Bundles of sheaths and needles** Needle crystals grow around −5 °C, even at sub-liquid saturation (e.g., Knight, 2012). In the atmosphere they often appear not as a simple needle shape, but instead in a bundled form. The sheath bundle is sometimes distinguished from the needle bundle, though they may be slight variations on the same form. Takahashi et al. (1991) reports sheath needles forming at −4.4 to −6.4 °C and liquid water saturation, with columns forming at immediately lower and higher temperatures. No other crystal form appeared in the columnar regime reported to lie between −4.0 and −8.1 °C. The non-bundled needle crystals observed by Knight (2012) had almost no normal growth of the prism faces, and thus it is hard to see how normal growth can explain the relatively wide diameters, and abrupt changes in diameter, of the bundled crystals. However, AST-driven lateral growth of prism faces could lead to extensions in a-axis directions, abruptly increasing the diameter and sprouting a new needle or sheath in the bundle. In this way, they are similar to the scroll forms.

**Trigonal crystals** These crystals have just three clearly distinguished prism faces, the other three being much smaller or indiscernible. A few possible explanations have been proposed for trigonal ice crystals, but they are either inconsistent with the observations of Yamashita (1973) or lack a specific growth mechanism. Yamashita (1973) argued that the trigonal forms grew from sub-micron droxtals, and more recently argued that AST-driven growth on an initial sub-micron prism face (more generally, of order $x_s$ or less) would dominate the droxtal, overgrowing the two immediately neighboring prism faces (Yamashita, 2014). The process may also promote the formation of next-neighboring prism faces, but even if the remaining three prism faces have equal chance of developing next, the sequence of events would lead to trigonal crystals in most such cases. These small crystals might not remain trigonal during growth, however, unless a mechanism exists that can maintain a stable trigonal structure. We suggest that the larger crystal with a small prism between two large prism faces would have vapor-density contours that allow slightly faster rates of layer nucleation on the small prism face. A small difference in layer nucleation rates would be amplified to a larger difference due to net AST from the large to the small prism, possibly stabilizing the trigonal form.

A recent review of ice growth from the vapor suggested that AST may be unnecessary for understanding ice growth forms (Libbrecht, 2005). The above examples suggest otherwise, instead arguing that many oft-observed secondary features may be inexplicable without the AST mechanism. Additional cases, including aspects of primary habit and rounding, are briefly examined in the appendix as well. The arguments are mostly qualitative; nevertheless, they serve to put very different growth forms into a common framework. They may also help stimulate new measurements of $x_s$, further observations, and more detailed modeling of these interesting crystal forms.

### 5.2.2 Implications for modeling and light scattering

To test the general magnitude of the AST role in ice growth, lateral-growth measurements are needed with greater precision than those given here. An interferometry study may provide sufficient precision of the lateral-front height and contour of the perimeter. For deducing the resulting $x_s$ values, the model introduced in Appendix A may be used. To test specific habit

mechanisms proposed here, we need better modeling—including vapor diffusion to realistic crystal shapes and relevant surface processes.

Presently, the most realistic crystal-growth model is that of Wood et al. (2001), but it is limited to hexagonal prisms. Some modeling approaches, such as cellular automata (Kelly and Boyer, 2014) and phase-field (Demange et al., 2017) simulate much

more complex shapes, but they unfortunately do not appear to include any of the relevant surface microscale processes directly. The list of relevant surface processes includes layer nucleation, defect-step sources, step clumping, and non-crystallographic regions (Nelson, 2005). To this list, we must now add that lateral growth processes with AST must be included.

Concerning light scattering from atmospheric ice, some studies have suggested that the outermost ice-crystal faces can introduce "roughness" that affects the visible-light scattering (e.g., Voigtländer et al., 2018). But in the crystal-growth field,

going back many decades (e.g., Frenkel, 1945; BCF, 1951; Woodruff, 2015), crystal faces are known to grow as atomically flat surfaces with nanoscale steps at low supersaturations, as occur in the atmosphere, except where hollows or branches sprout. Our experiments and observations are consistent with this well-established view of growth. However, the interior regions such as backsides, hollows, and pockets can show bumpier structures, and these interior regions are the more likely source of the "roughness" implied by the scattering results. The pockets, however, cannot be detected using the oft-used method (e.g., in Smith

et al., 2015) of examining ice-crystal replicas. In addition, for sublimation, our experiments showed no indication of rough features on the outermost surfaces (except the nanoscale roughness of a smoothly curved edge), such as those found in recent SEM studies (e.g., Magee et al., 2014; Pfalzgraff et al., 2010). In those experiments, little air was present, thus differing from atmospheric ice crystals. The presence of air had been argued previously to be important for the observed smoothly rounded shapes of sublimating ice (Nelson, 1998).

**6 Summary**

We have described here some previously unreported features on vapor-grown ice, including corner pockets, planar pockets, and elongated edge pockets, as well as provided more detailed observations of hollow terracing and hollow close-off. We argued that such features arose partly from facet spreading and protruding growth, both phenomena driven largely by surface transport across the boundary of a face to the advancing edge, a process we termed adjoining surface transport or AST. Several

quantitative models have been introduced that apply to lateral growth, including a model for center-pocket formation, and several qualitative models have been presented linking such growth to known secondary habits of snow crystals.

Our central point is that such lateral growth, long neglected in ice and snow research, may help explain a wide range of complex features and phenomena related to ice- and snow-crystal growth in the atmosphere, particularly when combined with normal growth. Protruding growth itself likely produces the two-level structure on many stellar snow crystals and also helps to

explain capped columns, multiple-capped columns, florid crystals, sheath growth, scrolls, sheath clusters, as well as various branch pockets and planar extensions. Lateral growth is also a likely factor in hollow terracing, banding, and close-off to make center pockets. Finally, the AST process itself likely contributes to the growth rates of sheath and dendritic crystals where it may substantially increase the growth rates and round-out the shape of the leading tip or corners. Finally, AST may also affect layer nucleation rates and explain trigonal forms.

As for immediate practical applications, we may infer the occurrence of an undersaturated cloud region via the observation of 12 corner pockets in a collected crystal, with the positions of the pockets providing the crystal size and aspect ratio at the time immediately after sublimation. Corner pockets may also form whenever a change in growth conditions leads to a transition between rounded and facetted growth, such as on branch backsides. Similar inferences of crystal conditions based on other

crystal features will likely be revealed in subsequent experiments. Thus, gaining a greater understanding of the formation of hollows, pockets, and various thin protrusions may lead to a more detailed knowledge of cloud conditions, and conversely, lead to better predictions of their occurrence in models. In turn, the improved predictions may improve the modeling of radiative transfer through ice-containing clouds. With such widespread potential applications, the phenomenon of AST-driven lateral and protruding growth deserves greater study.

## Appendix A: Lateral facet growth models for tabular crystals

We introduce the three mechanisms for the lateral growth of a face for the fits in Fig. 6. Referring now to Fig. A1, assume $r$ marks the radial edge-front of the face with height $h$. The rate $dr/dt$ is affected by I) direct vapor deposition to the edge-front, II) AST flux from the top basal face, from within $x_s$ of $r$, and III) normal growth of the rough region laying between radial position $r$ and the radius $a$ of the crystal. Cases I and II involve a face edge-front of height $h$, whereas $h$ is assumed zero for III. In case III, the position $r$ is the intersection of the curved face and the basal-face position $z = c$, marked with a dot in the sketch.

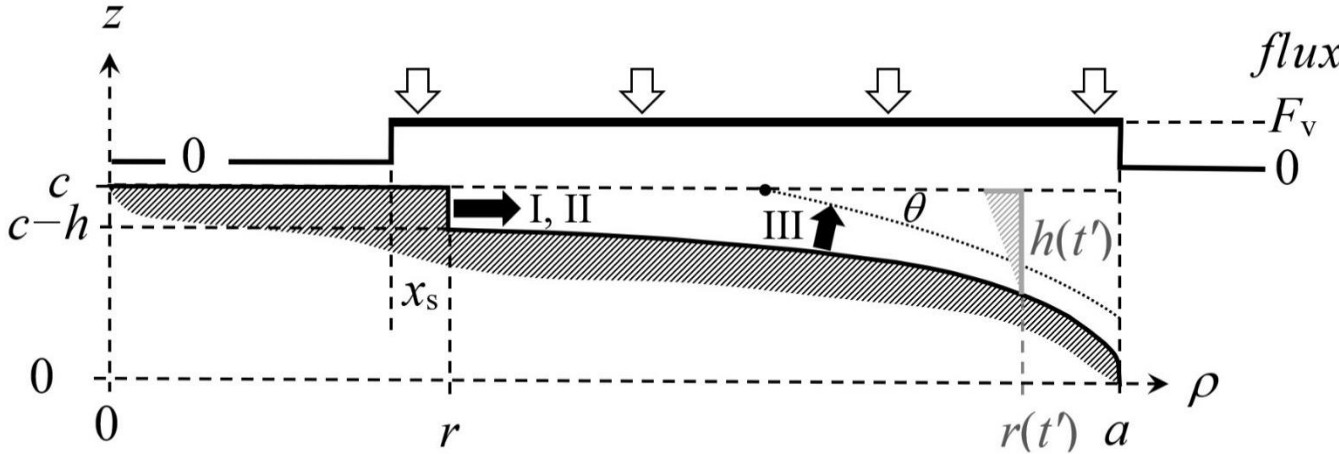

**Figure A1:** Facet spreading models. Dark shading shows the surface region of one quadrant of the crystal cross-section at a given time $t$. The flux calculation treats the crystal as a thin disc of radius $a$ with flux $F_v$ uniform between $r-x_s$ and $a$, zero elsewhere (upper plot). The face edge-front at radial position $r$ has height $h$ for mechanisms I and II. At a later time $t'$, the value of $h$ is larger (light shading) due to the advancement to $r(t')$, making a larger distance between the rough surface and basal surface plane. For III, the edge-front is instead assumed to lie at the intersection of the laterally growing face at $c$ (dashed line) and the dotted curve, intersecting with angle $\theta$.

All three mechanisms for $dr/dt$ depend on the vapor flux to the face, so we first estimate this flux. Assuming zero normal growth of the face (the basal face in this case), the flux normal to the face must be zero out to within $x_s$ of the face-edge at $r$. Beyond this point we assume a uniform flux $F_v$ (#/m²s) in the $z$ direction out to the edge of the crystal and then zero beyond. For the flux calculation, the crystal is assumed to be an infinitesimally thin disc of radius $a$. As done for other uniform-flux calculations (e.g., Nelson and Baker, 1996), the value of $F_v$ is determined self-consistently through an assumed surface response (deposition-coefficient function) to the vapor density at the surface, with the vapor density depending on $F_v$. The rates for I–III, top-to-bottom are

$$\frac{dr}{dt} = \begin{matrix} \Omega F_{\mathrm{v}} \\ \Omega F_{\mathrm{v}} \frac{x_{\mathrm{S}}}{h}\left(1 - \frac{x_{\mathrm{S}}}{2r}\right) \\ \Omega F_{\mathrm{v}} \frac{1}{\sin(\theta)} \end{matrix} \quad , \tag{A1}$$

where $\Omega$ is the volume occupied by a water molecule in ice (mass of molecule/mass-density), and $\theta$ is the angle between the rough surface beyond $r$ and the basal face (see Fig. A1). To calculate this angle for case III, we assume the rough surface to be the perimeter of an expanding ellipsoid of the same, fixed aspect ratio of the crystal. Such an assumption is unlikely to be accurate in detail, but nevertheless predicts angle $\theta$ to increase with $r$ as we expect. For the calculations, we use the treatment of the ellipsoidal coordinate system in Moon and Spencer (1961), and do not give the details here. For cases I and III, the flux is assumed to be in the normal direction right at the surfaces (edge $h$ and rough region, respectively) in Eqs. A1, even though the flux is assumed as along the z-axis for the calculation of $F_{\mathrm{v}}$. For case II, the prefactor comes from Eqs. B4 and B5, with the second factor in parenthesis arising from the curvature of the disc. The value of $h$ is not known from the measurements, and thus is treated as a fitting parameter here and then compared with the initial crystal profile. It only remains to determine $F_{\mathrm{v}}$.

In a stagnant atmosphere of air, the vapor density $N$ surrounding an infinitesimally thin disc of radius $a$ has flux $D\partial N/\partial z$ at the surface. For the first step of the calculation, we assume this flux is uniform over the entire top surface (i.e., $0 \le \rho \le a$). It is convenient to shift $N$ and making the variables dimensionless as

$$\begin{aligned} \Delta N' &\equiv \frac{N(\rho', z') - N_\infty}{N_\infty} \\ F_{\mathrm{v}}' &\equiv \frac{\partial \Delta N'}{\partial z'} \\ \rho' &\equiv \frac{\rho}{a} \\ z' &\equiv \frac{z}{a} \end{aligned} \quad , \tag{A2}$$

with $N_\infty$ the far-field vapor density. To determine the normalized flux $F_{\mathrm{v}}'$, we first assume it is known and solve for $\Delta N'$. In Nelson (1994), it is shown that

$$\Delta N'(\rho', z') = -F_{\mathrm{v}}' \cdot h_{\mathrm{td}}(\rho', z') \quad , \tag{A3}$$

where the thin-disc basis function $h_{\mathrm{td}}$ is an integral of Bessel functions. (This function is defined the same in A3 as are the analogous basis functions $h$ for the cylinder (Nelson and Baker, 1996; Nelson, 2001) and $Q$ for the hexagonal prism (Wood et al., 2001).) At the surface ($z' = 0$), this function simplifies to

$$h_{\mathrm{td}}(\rho', 0) \equiv \begin{matrix} {}_2F_1\left(\frac{1}{2}, -\frac{1}{2}, 1, \rho'^2\right) & \rho' \le 1 \\ \frac{2}{\pi} E(\rho'^2) & \rho' > 1 \end{matrix} \quad , \tag{A4}$$

where ${}_2F_1$ is the hypergeometric function and $E$ the elliptic integral. The curve is roughly bell-shaped about the origin, where it equals one, then nearly equaling $1/2\rho'$ for $\rho' > 1.5$. However, in the facet-spreading case, the flux is non-zero only in the thin ring $r - x_{\mathrm{s}} \le \rho \le a$, not the entire thin disc. So, we consider now the "thin-ring" basis function $h_{\mathrm{tr}}$ defined as

$$h_{\mathrm{tr}}(\rho', z', r', x'_{\mathrm{s}}) \equiv h_{\mathrm{td}}(\rho', z') - \frac{1}{\gamma} h_{\mathrm{td}}(\rho'\gamma, z'\gamma) \quad , \tag{A5}$$

where

$$\gamma \equiv \frac{a}{r - x_{\mathrm{s}}} \equiv \frac{1}{r' - x'_{\mathrm{s}}} \quad , \tag{A6}$$

always exceeds one. Eq. A6 defines $r'$ and $x_s'$. You can readily show that the derivative of $h_{tr}$ normal to the surface $(\partial/\partial z')$ gives a non-zero value at the surface only in the ring $r - x_s \leq \rho \leq a$ (or $1/\gamma \leq \rho' \leq 1$), where the value equals $-1$. For the edge-front, the relevant part of the function lies at $\rho = r$. We plot $h_{tr}$ at this position in Fig. A2. As the facet spreading situation is most similar to this thin-ring case, we use only $h_{tr}$ from here.

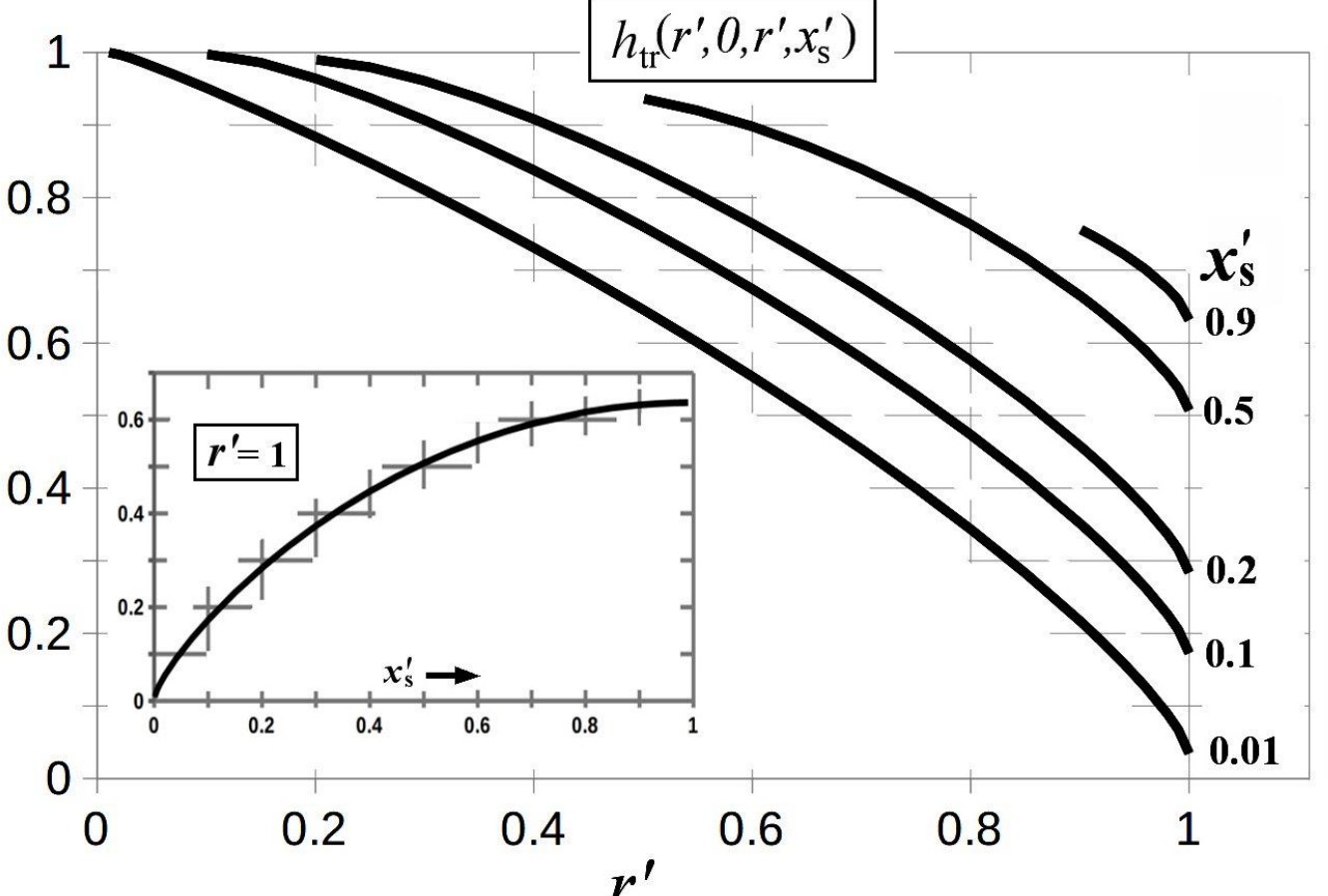

**Figure A2:** Basis function $h_{tr}$ for the thin ring. The function is evaluated at $z' = 0$ (i.e., the surface) and $\rho'$ at the position of the growing face edge-front ($r'$). The five curves are for the $x_s'$ values given at right. Inset plot shows the dependence on $x_s'$ when the edge-front reaches the crystal perimeter at $a$ (i.e., $r' = 1$). Hatches show the grid.

As the face edge-front $r'$ moves towards the crystal perimeter, the area that collects vapor decreases. This behavior is reflected in the decrease in $h_{tr}$ as $r' \rightarrow 1$ for all values of $x_s$. Each curve for a given $x_s'$ value begins at $r' = x_s'$ because the vapor-collection region starts at $r' - x_s'$, which cannot be negative. And when this starting point at $x_s'$ increases, the function decreases because $h_{td}$ decreases away from the origin. Exactly at the rim, where $r' = 1$, the only region of vapor collection is the ring of width $x_s'$ (Fig. A1). Thus, in this case, $h_{tr}$ approaches zero as $x_s' \rightarrow 0$, as shown in the inset plot; that is, a thin ring of growth hardly depletes the surrounding vapor.

Calculating the flux requires the surface-kinetic expression for the flux. Assuming a rough surface, the flux at the edge-front is one-fourth the vapor mean speed $v$ times $N(r',0) - N_{eq}$ (see, e.g., Nelson and Baker, 1996), which can be rewritten as

$$F'_v = a'(\Delta N'(r', 0) - \frac{N_{eq}}{N_\infty}\sigma_\infty) \quad , \tag{A7}$$

where $a' \equiv va/4D$ is approximately the crystal radius divided by the vapor mean-free-path. From Eqs. A3 and A7, one can eliminate $\Delta N'$ to derive

$$F_v = \frac{\frac{v}{4}N_{eq}\sigma_\infty}{1+a'h_{\text{tr}}(r')} \quad , \tag{A8}$$

where $h_{\text{tr}}(r')$ is shorthand for $h_{\text{tr}}(r', 0, r', x_s')$, which is plotted in Fig. A2. This expression is used with Eqs. A1 to plot the curves in Fig. 6.

The method of linear superposition of basis functions, as shown in Eq. A5, can be extended by adding more terms to properly treat the case of rough growth in the region $r-x_s \leq \rho \leq a$. That is, instead of a single ring of uniform flux with deposition coefficient unity, one can break the ring into many smaller rings, and then sum the terms. Nevertheless, the treatment here should capture the essential features of the diffusion field $\Delta N'$ and should be suitable for the present measurements.

For protruding growth, the behavior of $h_{\text{tr}}$ for $r' = 1$ (inset, Fig. A2) is relevant. For example, having $x_s = 5$ μm with a face radius $a = 100$ μm gives $h_{\text{tr}}$ of only $\sim 0.08$. Moreover, this value will decrease further as the protrusion grows due to $a$ increasing at fixed $x_s$. Having such low vapor depletion will not only speed up the lateral growth, but may also allow the protrusion to nucleate layers more closely, possibly aiding a roughening transition. Of course, this treatment assumes no normal growth of the face and no direct vapor flux to the edge in the radial direction. Such modifications can be added. The resulting expression will be similar in form to Eq. A8, but with added terms in the denominator that reduce the flux.

**Appendix B: Secondary features and habits**

Further evidence that lateral facet growth is a common factor in ice growth comes from examination of various secondary features and secondary habits. We examine here several cases, but as ice-growth from the vapor in air is often complex, having several competing processes on complex shapes, each case requires a different approach and presently can only be treated in a qualitative fashion. Most of these features and habits appear inexplicable with normal-type growth processes only, and only a few of them have even seen attempts at explanation. Addressed here are

- Center pockets, terracing, and banding from hollows (B.1)
- Center pocket variability (B.2)
- Two-level formation and center droxtals on planar forms (B.3)
- Corner pockets on rounded tabular backsides (B.4)
- Capped columns, multiple-capped columns, and florid crystals (B.5)
- Rounding of plates and tips of fast-growth forms (B.6)
- Tip shapes of sheath and sharp needles (B.7)
- Scroll crystal features (B.8)
- Bundles of sheaths and needles (B.9)
- Protruding growth on branch backsides and ridge pockets (B.10)
- AST contributions to trigonal formation and primary habits (B.11)

**B.1 Hollow close-off to center pockets, hollow terracing and banding: mechanisms**

Center hollows show variable behavior. Concerning their formation, Libbrecht (2005) and other authors (e.g., Gonda and Gomi, 1985) have referred to the process as an instability. In the standard treatment, however, the hollow occurs when the gradient in

supersaturation needed for uniform growth can no longer be compensated for by the step density (e.g., Kuroda et al., 1977; Frank, 1982). In other words, normal growth of the entire facet becomes impossible, which is different from being unstable. In this "impossibility" case, one expects i) the hollow initiation and shape to be nearly identical on identical faces in a nearly uniform environment as well as ii) being highly reproducible when other crystals grow under the same conditions. On the other hand, if hollowing is merely an unstable phenomenon, then a sufficiently uniform, constant condition may be expected to circumvent the hollowing indefinitely. Conversely, if hollows do form, then their initiation and shape should differ between identical faces due to minute differences in conditions. We suggest here that inclusion of lateral growth processes predicts qualities of unstable growth at low supersaturations, leading to hollow close-off and terracing features.

The standard facet-impossibility approach seems qualitatively successful in some cases of middling supersaturation (e.g., Nelson and Baker, 1996) and at relatively high supersaturation where the hollow tends to keep enlarging in width (e.g., hollow columns) or advance into branches (e.g., dendrites) in a generally consistent, repeatable fashion on all identical face types. But at low supersaturations, the hollow often varies in width, getting wider, then getting narrower, and may even close-off into a center pocket. Large changes in width also occur at middling-to-high supersaturations (e.g., Smith et al., 2015), but are much more pronounced at the low supersaturations here. Gonda and Koike (1983) also observed the closing-off of hollows during growth at one atmosphere and supersaturations up to 33% at −30 °C. At low supersaturations, otherwise identical faces can have different patterns of hollows and pockets. Thus, at least at low supersaturations, the hollow phenomenon does seem to have some qualities of an instability.

In the cases shown here, the center hollows vary considerably even though the conditions are nearly constant. For example, the hollow's size and shape can vary considerably between different faces of the same crystal in Figs. 11 and 12. Also, a given hollow's width can change suddenly, often showing periodic terrace-like features, and sometimes closing-off completely, leaving a center pocket. Such behavior suggests a complex process involving competing influences and a possible instability. Here we describe a simplified mechanism for such an instability between normal and protruding growth, and argue that the growth behavior of adjacent faces may influence hollows, particularly at low supersaturations, possibly leading to the above-mentioned observations.

First, consider the initial hollowing. The overall driver of hollowing of a surface is lateral supersaturation ($\sigma$) gradients across the surface. These gradients are influenced by growth on all crystal faces such that, for example, normal growth on the basal face produces a decrease in surface supersaturation, starting from a high value in the middle of the prisms $p$, as sketched in Fig. B1a, to the $p$–$b$ edge, to the smallest value in the center of the basal. Normal growth only on an adjacent prism faces produces the opposite gradient on the basal (dotted lines). In general, normal growth occurs on all faces, and thus the contributions from both sets of contours in (a) are superimposed with a weight in proportion to the normal growth rates (Nelson and Baker, 1996). The normal growth rate of a given face is proportional to the areal flux $F_v$ to the surface, which is mainly the vapor-diffusion flux

$$F_v = DN_{eq}\nabla\sigma \cdot \hat{n},$$
(B1)

where $D$ is the vapor diffusion constant, $\hat{n}$ is the surface normal, and $N_{eq}$ is the equilibrium vapor density at the local surface temperature. This means that a larger growth rate of a face implies a larger normal gradient, which should positively correlate to a larger lateral surface gradient. Hence, taken together, the surface gradient in supersaturation that leads to hollowing, say on the basal face, will be weaker at low normal growth rates of the basal and also weaker at high normal growth rates of the adjacent prism faces. That is, the growth on one face influences the lateral gradients on the other faces.

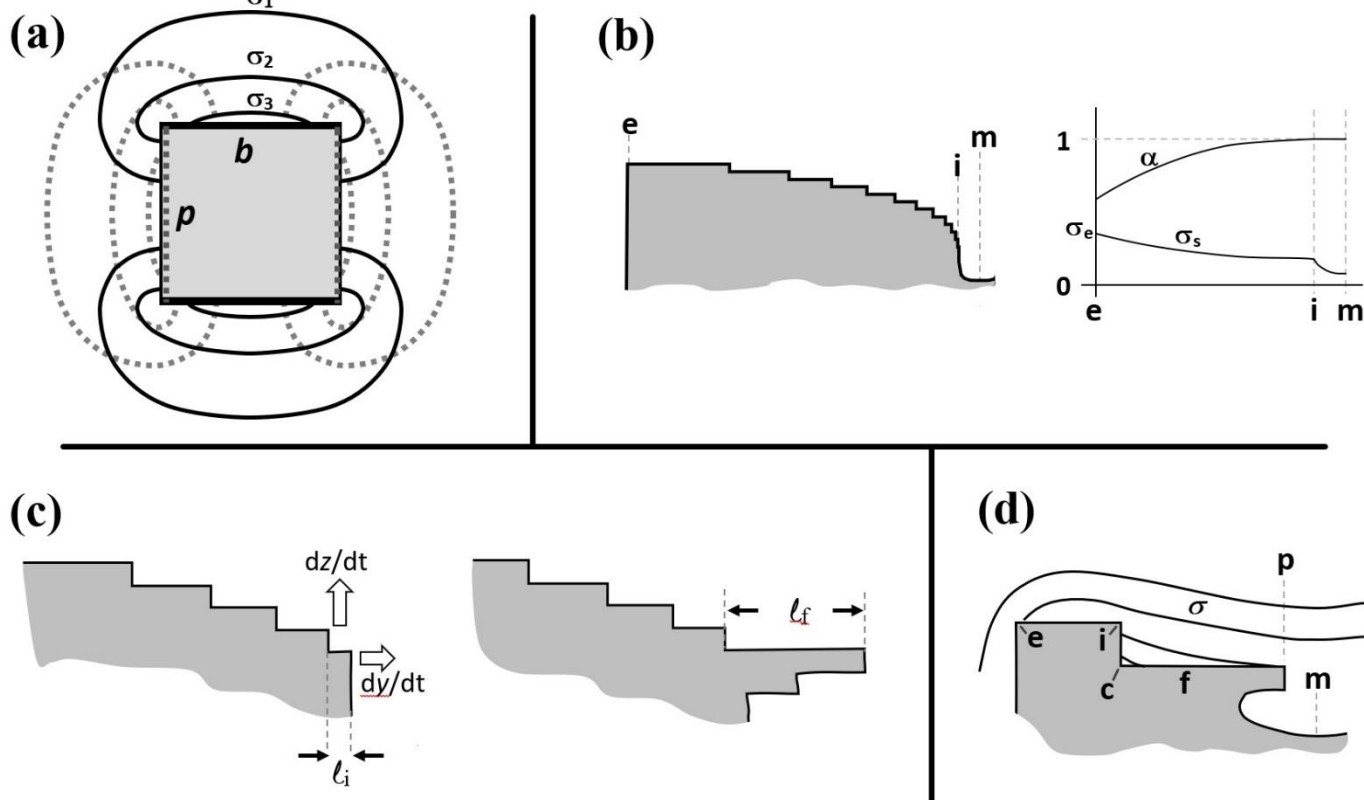

**Figure B1:** Processes involved in hollow close-off and terrace formation. (a) Basic patterns of supersaturation (equivalently, vapor-density) contours on a solid prism for growth only on the basal faces *b* in solid, dark curves with $\sigma_1 > \sigma_2 > \sigma_3$. The same for growth only on the prism *p* in dotted grey curves. The general case has a linear superposition. (b) Standard picture of hollow formation. Steps coming from edge at **e** become closer towards the inside edge **i** at right. Face middle is **m**. (Other half of face, as well as steps on side face, are not shown.) The plot on the right side shows the deposition-coefficient function $\alpha$ at top, with the surface-supersaturation $\sigma_s$ trend sketched below. (c) Close up of step-clumping region near inside edge **i** with step separation $l = l_i$. Protruding growth can occur towards the right (*dy/dt*), normal growth upward (*dz/dt*). Protrusion shown at right, last step length now $l = l_f$. (d) Supersaturation contours when terrace **f** forms via lateral growth. Inside corner is **c**, protrusion is **p**.

The lateral surface gradient in supersaturation leads to hollowing via its effect on the surface steps (e.g., Frank, 1982; Nelson and Baker, 1996; Wood et al., 2001). Briefly, as sketched in Fig. B1b, steps originate from the crystal edge **e** and flow towards the face center **m** on the right. The sketch on the right shows the trends of vapor supersaturation along the surface and deposition coefficient function α. Near the edge, the vapor supersaturation $\sigma_e$ is relatively high and the steps are relatively far apart, but there is a relatively high fraction that desorb, which is described by its low deposition coefficient $\alpha_e$. As the steps move toward **m**, they slow down and become more densely packed, thus increasing the local deposition coefficient. At the edge **i** of the hollow, essentially all the incident molecules reach a step and the steps are clustered together to the point that they hardly move. A wall of steps builds up here, at the step-clumping region (SCR), forming the edge of the hollow. (Neshyba et al. (2016) proposed a more detailed model of step dynamics for ice with a thick surface-disordered region, but it is not yet clear how a hollow would develop in that model.)

However, after the hollow forms, the local supersaturations may change. This change could be due to either a change in external conditions (e.g., temperature or supersaturation), a change in growth rate of a face due to a changing activity of the step source, or simply the increasing size of the crystal. For example, an increase in crystal size will generally decrease $\sigma_e$. Regardless of the cause, consider now the sketch in Fig. B1c in which a slight change in local conditions near the hollow edge **i** has

occurred, causing a slight increase in the local step separation $l$. The normal growth rate $dz/dt$ at **i** is the step height $h$ divided by the step-passage time $\tau$ (time between successive passings of a step at **i**). The latter time is the step separation divided by the step speed $v_s$. However, for a protrusion of thickness $nh$ ($n \geq 1$) that starts into the hollow, the protruding growth rate $dy/dt = v_s/2n$, with the factor ½ due to the AST flux coming only from the top side. Comparing the two rates,

$$\frac{dy}{dt} = \frac{l}{2nh} \cdot \frac{dz}{dt}. \qquad\qquad\qquad\qquad (B2)$$

When the hollow first forms, $l$ may be of order $h$. But an increase in $l$ causes $dy/dt$ to increase, further increasing $l$ and thus further increasing $dy/dt$. Eventually $dy/dt$ may greatly exceed $dz/dt$. Hence, in this very basic description, some change in step spacing near the hollow may become unstable, leading to a protrusion that can continue to grow, eventually sealing-off the hollow into a center pocket. Initially, there will also be some direct vapor-flux to the side of the hollow at the lip, but this
contribution to protruding growth would vanish due to shielding from the opposite side as the pocket closes off.

This basic treatment neglects lateral supersaturation gradients and advancement of the crystal face. Briefly, these factors make sealing-off of a hollow less likely at higher growth rates because the initial protrusion will become left behind in a lower supersaturation region as the rim grow higher. As the supersaturation drops, the protrusion grows slower, amplifying the effect. Also, as the protrusion grows inward, the supersaturation should decrease (except in the case mentioned next), thus hindering or
possibly preventing the instability. Thus, the above suggests a hollow instability at low growth rates, but not at high rates.

Concerning terracing, if the SCR develops nearer the rim and becomes elevated as in Fig. B1d, the interior region **f** may flatten via facet spreading. Such F-growth would be aided by a reversed surface supersaturation gradient; that is, if the inside corner **c** becomes isolated in an effective vapor shadow (height **i**–**c** exceeding the vapor mean-free path), then the steps in region **f** will speed up as they go towards the higher supersaturations near the center. This would produce an interior face that spreads
laterally, flattening region **f** into terrace features such as those in Figs. 11,12.

The growth of crystals with many terraces has been called skeletal or hopper growth, but the structure differs between that in relatively squat hollows and that in narrow columns. Referring to Fig. B2, we call the former as terraced (a) and the latter as banded (b). A sequence showing banding during growth at atmospheric pressure, −30 °C, and 8.8% supersaturation is in Gonda et al. (1985). Schnaiter et al. (2018) shows the banding in hollow bullet rosettes from clouds and Nakaya et al. (1958) shows
numerous cases on hollow columns and sector-like forms. The bands are much denser in the latter forms. Indeed, terracing and banding are very common in natural snow and hoarfrost.

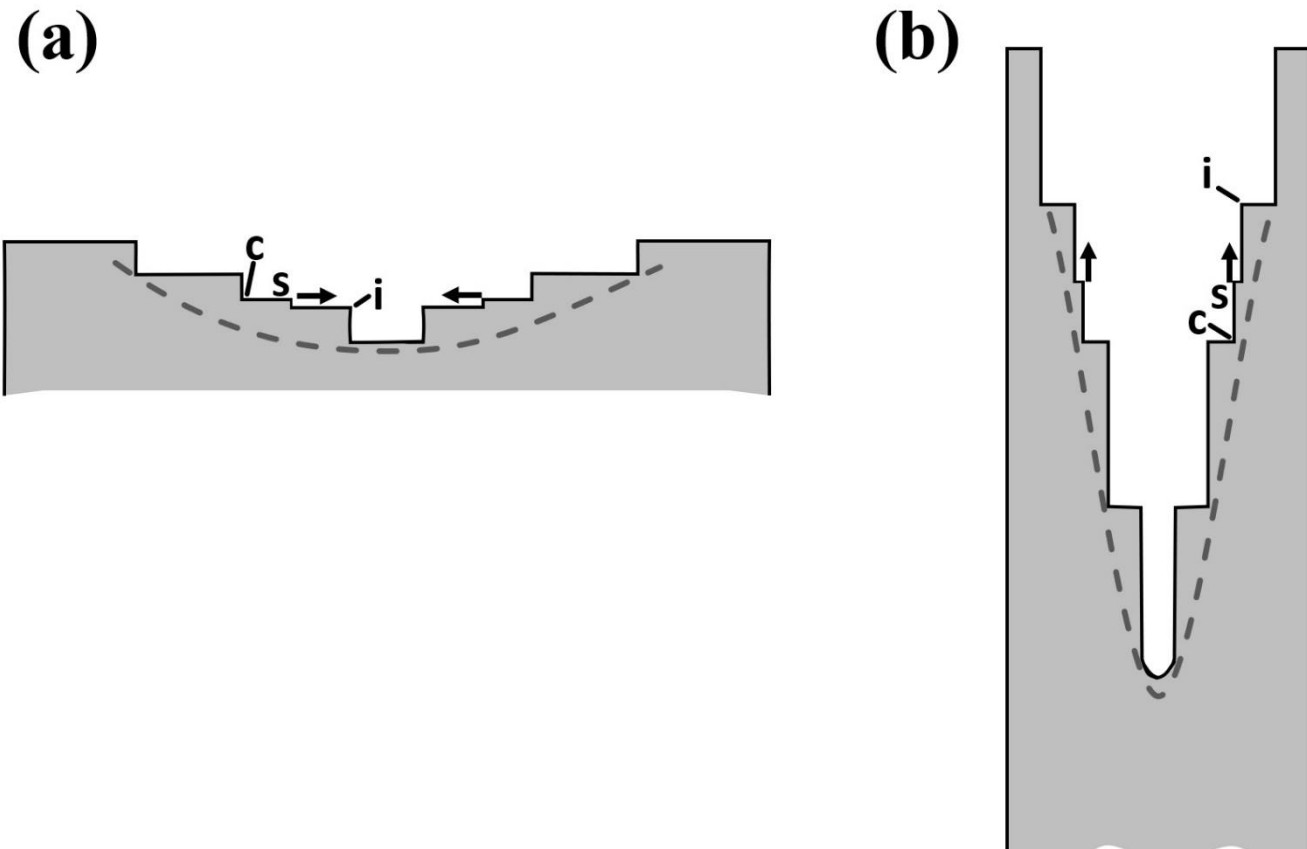

**Figure B2:** Terracing and banding in hollows. Sketches are cross-sections of top half of crystal, the dashed lines representing previous, smooth profiles. (a) A hollow on a squat crystal with terraces (may be either basal or prism face). On a sufficiently recessed terrace, the surface supersaturation increases towards the center from **c** to **i**, and thus single step **s** speeds up as it traverses the face, flattening the terrace. (b) A hollow on a long column or sheath with bands. The large macrosteps with inside corner such as **c** may produce the observed banding. A single step **s** starting between **c** and edge **i** would speed up upon approaching **i**, flattening the band region.

Concerning the formation of a terraced hollow, initially, the inner surface may be smoothly curved, as shown in the dashed line in Fig. B2a. Starting from such a smoothly curved surface, the closely spaced single steps may clump into large step clumps as described by Mason et al. (1963). In that process, once two single steps become close enough to essentially lock together, they move more slowly, allowing steps further back to catch up. Mason et al. did not include supersaturation gradients, but such gradients may promote the clumping. In this way, a step-clump of two quickly becomes a clump of three, and the clumping continues. (Velikov et al. (1997) describes a more complex interaction for step-bunching.) Once a sufficiently tall clump forms, any single step **s** between an inside corner **c** and an edge **i** would speed up as it approached **c**, flattening the terrace as described above.

The banding in a narrow column in Fig. B2b similarly starts with a smoothly curved interior surface, and may form a macrostep by the same step-clumping process. But in this case, a single step **s** flows from the center out towards the higher supersaturations at the rim. Thus, a step clump at **i** does not need to be high before the flattening effect becomes large because in this case the step is speeding up due to the higher supersaturation even without an edge at **i**. As with the terracing case, as **i** grows, it sticks out into regions of higher supersaturation, meaning that the next single step may travel even faster. Thus, a later step overcomes a previous step, quickly building up a larger macrostep, which would appear as a band in the hollow. The

hollowing may start with a single band, with new bands forming via the same process as the crystal grows, leading to a series of nearly equally spaced bands.

This treatment suggests that step sources and dynamics, F- and P-growth, the moving interface, and the shape of the supersaturation contours all likely influence hollow structure, leading to their highly variable behavior even under constant growth conditions. A similar process of banding may also apply to the 'cross-rib' features (Nelson, 2005) on the backsides of branches on broad-branch and sector-plate crystals. The suggested mechanism in that study was instead changes in temperature or supersaturation around a crystal. These changes would cause the width of the branch to vary, and the same process may also produce some terracing and banding in hollows by temporarily changing the rim width **e–i** in Fig. B1d. In a changing environment, more than one mechanism may alter the hollow structure.

**B.2 Cause of pocket-size variability**

Hollow sizes and shapes are highly variable under low normal growth rates (e.g., on different faces of the crystals in Figs. 11 and 12). Such variability is uncommon at high growth rates, so we outline here a few factors that may play larger roles at the low growth rates at low supersaturations.

One factor is the greater variability in the normal growth rates. In contrast to high-supersaturation growth, the step-sources at low supersaturations are thought to be crystal defects such as dislocation outcrops and stacking faults. The dislocation activities will in general be different on different faces of the same crystal, and between different crystals, and also may change during growth. But other factors may lead to greater variability at low supersaturation. For example, the greater relative role of lateral growth processes at low normal growth rates leads to the phenomena described in the previous section; that is, the interplay between the surface influence and the bulk vapor-diffusion influence may allow more complex nonlinear feedbacks on growth, leading to a greater chance of unstable behavior. This variability may be increased by the variation in dislocation activity, which would have a larger influence at low supersaturation because the surface has a larger direct influence on growth rates under these conditions. Finally, after a given duration, the smaller crystal sizes at low growth rates mean that variability in the initial droxtal size and properties would have a relatively larger influence on the later crystal form.

In contrast to the other low-supersaturation crystals shown here, the six planar pockets in Fig. 3 are remarkably similar. The reason for the pocket symmetry is likely due partly to the equal normal-growth rates of all six prism faces. This symmetry in the growth rate must arise from having the same step source on all faces. Given that the crystal has an apparent stacking fault or stacking-disordered region that intersects all prism faces, the obvious step-promoting defect would be the fault. Fault-generating steps had been proposed by Ming et al. (1988) via a mechanism in which the fault yields a lower barrier to layer nucleation. Thus, we suggest that the six pockets open-up at the same time because the step-generation mechanism is the same stacking-fault mechanism on all six faces, producing the same normal growth rates on all faces. (That the fault could both be a source of growth and a location of hollowing is harder to explain, but possible given that the steps would start from the prism–prism edge, not the hollow location.) Concerning the hollow closing-off to form pockets, two factors occur during growth that will likely change the step-separation near the pocket, thus determining whether they close-off: i) The edge supersaturation decreases due to the larger crystal areas, and ii), the relative position of the pocket-opening changes on the prism face due to one basal face growing faster than the other (i.e., the side-view in Fig. 3d shows greater advance of the right basal face than the left). As both of these factors will be equal for all six prism faces, the closing-off should occur at the same time, leading to the identical nature of the six pockets. Finally, note that the groove region, which is sublimation-rounded like the edges, did not produce a pocket during re-growth like the corners. The reason for this may be the much smaller radius of curvature in the former case.

**B.3 The droxtal center on two-level planar crystals**

The two-level structure occurs in planar crystals P2–P4 (Kikuchi et al., 2013) grown near water saturation around −15 °C. Some such crystals show a circle at the center, suggesting that the original droxtal largely remained unchanged. But it has been unclear how the original droxtal's basic shape could remain intact as vapor deposition "filled-out" the crystal via normal growth of the faces. The answer appears in Fig. B3, which shows how this circle shape can remain when the top and bottom basal planes extend by protruding growth. (Takahashi and Mori (2006) also show several cases of such sprouting.) The process of formation via protruding growth has been described by Yamashita (2014), but we include it here due to its close relation to other phenomena we describe.

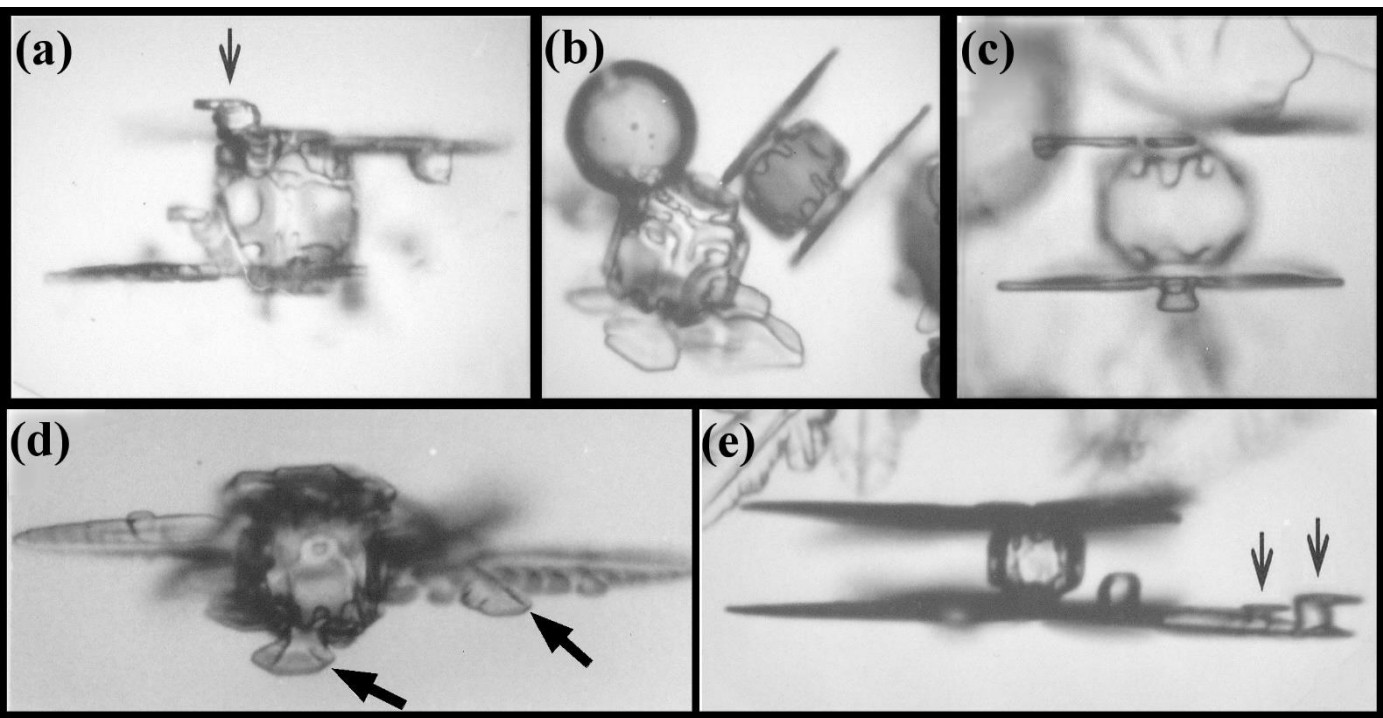

**Figure B3:** Protruding growth on large droxtals (~40–70 μm) to form the two-level structure near −15 °C and liquid-water saturation. Black arrows in (a) and (e) show protruding growth on smaller rime droxtals. Black arrows in (d) shows complex structure of branch and sidebranch backside. (From the cloud chamber, courtesy of A. Yamashita.)

The overall process is sketched in Fig. B4. In (a), the droxtal has just frozen. The latent heat release raises the droxtal's temperature to about 0 °C, but if the droxtal is relatively isolated, its temperature returns to the ambient value within a fraction of a second. Then, the surface quickly depletes the nearby air of vapor, driving down the surface supersaturation to a value that greatly suppresses layer nucleation on the basal faces (Nelson and Knight, 1998). Thus, after the basal facets form on the top and bottom of the droxtal (a–b), they mainly spread via F-growth. The growth continues as protruding growth in (c), as was the case for corner pockets. But, unlike the corner-pocket case, protruding growth does not occur on the prism faces, and thus the basal protrusions extend out from the boundaries of the initial droxtal, creating two levels. A possible reason for the lateral growth rate being larger on the basal than the prism may be the proposed larger $x_s$ value on the basal in this temperature regime (Mason et al., 1963). Sketch (d) shows one level grows more than the other, which is due to asymmetry in the vapor-diffusion field around the falling crystal (Fukuta and Takahashi, 1999). As shown in Fig. B3a, and found much earlier by Nakaya (1954), branches can also occur on both levels.

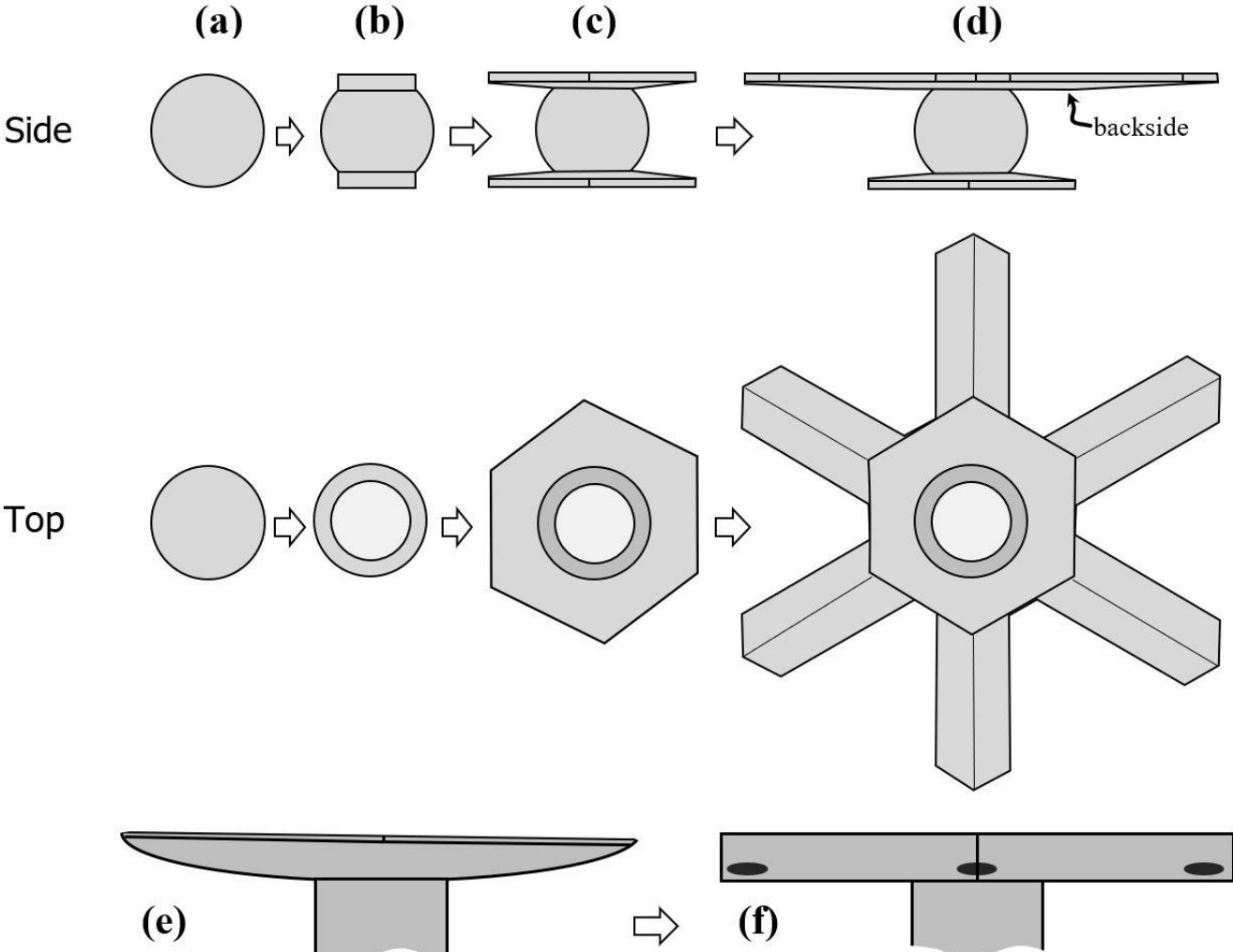

**Figure B4:** Facet spreading and protruding growth on a droplet or column under constant or varying conditions. (a)–(d) follow the description first given by Yamashita (2014). Here, the growth on a frozen droplet leads to protruding growth and a two-level snow-crystal. For sketches in (d), compare the side view to images in Fig. B3, and the top view to the snow-crystal in Fig. 8c. (e) Cap on one end of frozen droplet or column, showing rounded backside. (f) Same cap after a decrease in growth rate, allowing corner pockets to form. Details of the process between (e) and (f) are not shown, but would be the same as that shown in Fig. 4 except that only one side of the crystal is rounded. This latter process may explain the observed line of pockets on the inset image in Fig. 8c.

Such abrupt sprouting can explain the small center circle observed in some branched snow crystals (e.g., Fig. 8c). Not all branched, two-level crystals show such a "droxtal center". The two levels in these other cases likely arise instead via the hollowing-type ("lacunary") process observed by Yamashita (1979) (see also Frank (1982), Nelson (2005)). Bentley (1924) wrote that "at least half" of the 4200 crystals he had photographed in 40 years had such a center circle, and argued that it was the frozen droplet (droxtal) upon which the crystal formed. What factors may cause the droxtal sprouting in some cases but not others? Yamashita (2014) observed sprouting on larger-than-average droxtals. Large initial droxtals may favor sprouting because their larger areas would depress the surface vapor density more than that of a much smaller droxtal, essentially shutting off the normal growth on the basal face. This effect of size may also lead to a greater vapor-density gradient where the protrusion starts, particularly at higher ambient supersaturations that would tend to produce higher supersaturation at the face edge. In addition, the

AST flux should be higher when the basal face has larger $x_s$ values. Mason et al. (1963) found $x_s$ to peak in −9 to −15 °C, a temperature near which such two-level crystals sprout. Also, high supersaturations likely promote such sprouting: In the recent vertical wind-tunnel experiments of Takahashi (2014), nearly all the images of planar snow crystals clearly show the center droxtal as described here. In those experiments, the crystal nucleated and grew in a droplet cloud of various liquid-water

contents, and thus grew near liquid-water saturation. The mean droplet diameter (before freezing) was 8 μm, but inspection of the images indicates that the droxtals that sprouted the two-level crystals had a slightly larger diameter (~9–13 μm). Thus, both relatively high initial supersaturations and relatively large droxtals may favor two-level initiation via protrusive growth.

**B.4 Corner pockets on rounded tabular backsides**

The snow-crystal image in Fig. 8c appears to show corner pockets on both levels of a two-level crystal. In addition, the smaller

inset shows a sequence of small circles along the centerline of a branch, a pattern reproduced on the other branches as well, ruling out the possibility that these are rime. Similar series of circles appear in crystals #11, 12, and 22 in Bentley (1924). In most cases, the circles appear before the crystal branches sprout. These circles may be corner pockets by the following mechanism. During growth, the backside of the plates and branches of two-level crystals show rounded features even without sublimation (c.f., Fig. B3d, Shimada and Ohtake, 2016). A short slowdown in growth may allow facet spreading and protruding

growth from the rounded backside region via steps similar to those shown in Fig. 4b–e, but on one side only. The protruding growth from the side is more difficult to picture in this case due to the positioning on a ridge. Nevertheless, the basic process may be like that sketched in Fig. B4e–f. If correct, the existence of each pocket marks the time when growth temporarily slowed down.

**B.5 Capped columns, multiple-capped columns, and florid crystals**

Capped columns (CP1a, CP1b) and multiple capped columns (CP1c) form when a columnar crystal quickly moves into a high-supersaturation region with temperature in a tabular regime. For example, a column growing near −8 °C can be quickly lifted in a vigorous updraft to the thin planar regime starting below about −10 °C. The resulting form is similar to the droxtals with two levels (e.g., Fig. B3), but with a column between two basal extensions such as a thin plate or dendrite. The left crystal in Fig. B5 shows two thin end caps and another thin, but shorter, central plane in between. At the right is a case where the central plane is

longer and all three plates are thicker than those in the other crystal. The planes in this case may have passed back into the columnar temperature regime. These two crystals are multiple capped columns.

A previous model of capped-column formation involves an extreme form of hollowing in which the step-clumping region (SCR) on the prism faces forms near the step origin at the basal–prism edge (Nelson, 2001). However, that model cannot readily explain the central thin basal extensions that occur in the multiple-capped column (type CP1c). Instead, the similarity to the two-

level case in Fig. B4 suggests an AST contribution to cap formation. If so, what is the source of the originating basal plane on the interior of the column for the CP1c case?

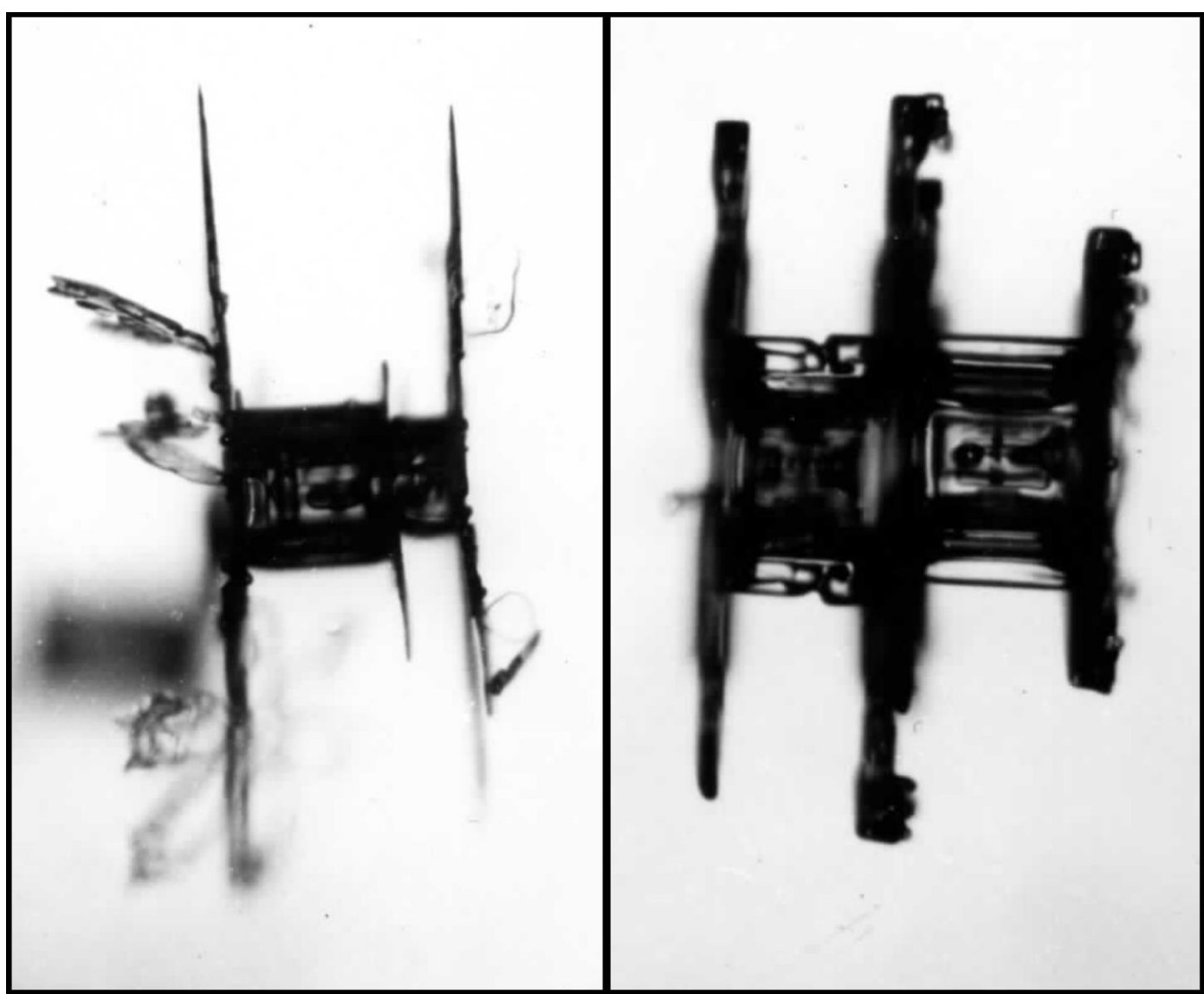

**Figure B5:** Multiple-capped columns from the Magono–Lee collection (Magono and Lee, 1966).

The basal protrusions on rime in Fig. B3a,e (thin arrows) suggest one possible source of an interior basal extension. As sketched in Fig. B6a (top), a rime droxtal could develop a basal face aligned along that of the column (assuming the droplet freezes with the same orientation). The face would grow laterally and then protrude via AST as shown in (b). Once the basal extension starts, it can grow both outward and around the column. If the two end caps have a head-start on growth, they would deplete the nearby vapor, making the rime droxtal nearest the center the more likely one to have sufficient vapor to develop a significant basal extension. (Otherwise, two basal extensions may form relatively close together, competing for vapor until one grows significantly larger, stunting the other.) The image labeled CP1c in Fig. 1 of Kikuchi et al. (2013) shows other rime droplets along the column, suggesting this mechanism. Without the rime, it may be unlikely that a high density of new layers could nucleate in the middle of the column, produce an SCR, and sprout a new plate.

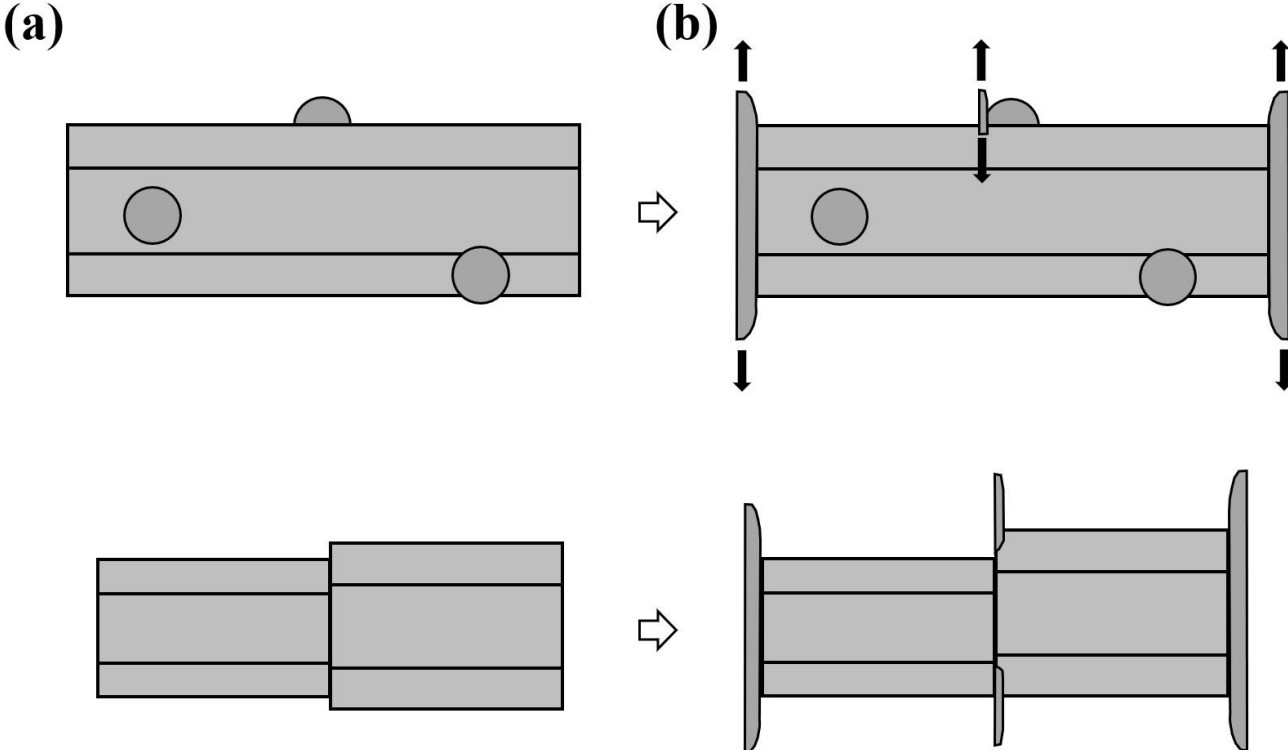

**Figure B6:** Protruding growth leading to multiple capped-column crystals. (a) A columnar crystal either with rime droplets (top) or some sort of break feature (bottom). (b) Moving into a tabular-growth temperature leads to protruding tabular growth. Black arrows in rime case show that the protruding growth extends outwards in all directions within the plane.

In addition, the examples of CP1c in Fig. 5 show no rime on the column, but a shift in the column that provides an interior basal plane upon which to sprout the extension. Figure B6, bottom, shows how this case may proceed. Although we know neither how common such crystals are nor how the interior planes arise, we show here one such case in Fig. 9. Such an interior basal plane could also arise from a small bundle-type column such as a bundle of needles (C1b) or bundle of sheaths (C2b), which are

discussed in Appendix B.9. Other cases include a column with a small double-twin (e.g., §4.10 of Kobayashi et al. 1976), crystals such as those in Fig. 12 (possibly twins), columns with prism hollows, or a column that underwent previous transitions in the tabular regime.

Similar to the capped columns, the protruding growth process may also influence the initiation of tabular extensions on 'florid' crystals or side planes (Bacon et al., 2003). The base crystals (before sprouting) are squatter than the columns for the capped

columns, and sometimes polycrystalline. See type P8b 'complex multiple plates' (Kikuchi et al. 2013). Also similar to the capped columns are the bullets with plates (CP2c), which probably form by the same processes. The left crystal in Fig. B5 shows smaller planes on the end caps in different directions. Such new crystal orientations likely formed on rime droplets that froze in these other orientations, then extending like those in Fig. B6b (top).

**B.6 AST contribution to normal growth rates of thin plates, dendrites, needles, and sheaths**

To estimate the contribution of the AST flux to lateral growth of several growth forms, consider a simple treatment based on the BCF (Burton et al., 1951) model of crystal surfaces. Assume here that the region over the face edge (i.e., the lateral-growth front) is rough, and thus this region can be treated as BCF do for a step edge, that is, as having an equilibrium concentration of mobile

surface molecules. Assume further that, as suggested by step-motion experiments (Hallett, 1961), molecular migration over the edge encounters no barrier. In this case, straightforward use of BCF gives a flux of molecules $f_L$ (per edge length) that equals

$$f_L = x_s \frac{v}{4} N_{eq} \sigma_e , \qquad (B3)$$

where $x_s$ is the surface migration distance, $v$ is the mean molecular speed in the vapor, $N_{eq}$ is the equilibrium vapor density, and
$\sigma_e$ is the vapor supersaturation near the edge. This result suggests that we can view the adjoining face region within $x_s$ of the edge as a collection region of molecules impinging from the vapor. Assuming that this face edge has $n$ adjacent facets from which to draw the flux, and the edge has thickness $t$ over which the AST flux is distributed, the effective flux (per area) $F_{AST}$ is

$$F_{AST} = n \frac{x_s}{t} \frac{v}{4} N_{eq} \sigma_e , \qquad (B4)$$

an amount we compare to the direct vapor flux $F_v$ (e.g., Nelson and Baker, 1996),

$$F_V = \frac{v}{4} N_{eq} \sigma_e . \qquad (B5)$$

where the deposition coefficient is assumed to be unity (effectively rough surface), consistent with the assumption of a step edge in the derivation of Eq. B3. Thus, as a crude estimate, the ratio of AST flux to standard vapor flux is just $n \cdot x_s/t$, with $n \sim 1$ or 2 depending on whether the thin edge is bound by one facet, as in a dendrite branch, or two facets, as in a thin disc. In general, $n$ can vary between 1 and 2 when two faces meet on one edge, such as the two prism faces along the edge of a sheath, and may
exceed 2 in the case of a thin whisker. In the next two subsections, it will be convenient to view the total flux as equivalent to having an effective supersaturation of $\sigma_{eff} \equiv \sigma_e \cdot (1 + n \cdot x_s/t)$.

    Concerning $x_s/t$, early estimates of $x_s$ from step-motion experiments on the basal face gave a range of values, depending on temperature, of about 1–6 μm (Mason et al., 1963; Kobayashi, 1967). A more recent measurement gives a value of about 5–10 μm at −8.6 °C (Arakawa et al., 2014). For the thickness $t$, a recent study shows that the tip region of a dendritic snow crystal has
a tapered tip (Shimada and Ohtake, 2016; 2018). The measurement does not give a precise value of $t$ exactly at the edge, but an average value within ~10 μm of the edge gives about 0.3 μm. Thus, the estimates of $x_s/t$ here suggest that the AST flux could be up to ~3–60x the direct vapor flux in certain cases. The resulting increase in growth rate from this flux would be less than this factor due to the vapor-diffusion process. Further analysis (to be included in part II) suggests that a partial barrier to migration over the edge may exist, and the resulting reflection of some admolecules at the edge would reduce this AST flux. Nevertheless,
such a flux could significantly increase the rate of growth (maximum dimension) of thin planar growth forms such as the dendrite (P3b) and fern (P3c) crystals, as well as sheath (C2) and needle (C1) forms. For a thin disc, a more complete calculation is at the end of Appendix A.

## B.7 Rounding of plates and tips of fast-growth forms

Fast-growing crystals have leading growth fronts that can appear rounded, and some thin tabular crystals can be disc-shaped or
scalloped. For example, the sheath extensions in Fig. B7 include some with tips that appear rounded and some that appear flat-faced. Similarly, Knight (2012) observed sheath-needles with rounded tips. Sei et al. (2000) shows rounded sheath-needles sprouting from prism corners that later flatten upon becoming larger. An increase in supersaturation would then cause smaller, round tips to sprout on the larger, flat tips. Round tips also seem to appear on the faster-growing dendrites near −15.0 °C (P3b,c). Gonda and Nakahara (1996) even found such rounding on dendritic crystals grown with their basal face against glass. (However,
for the fast-growing dendrite cases, the small scale of the tips makes it hard to discern small facets with limited image resolution, and a slight amount of rounding may quickly occur in brief undersaturated conditions, so the phenomenon is not well-established yet.) Away from the tip, rounding of the side vertices has been attributed to SCR forming due to decaying gradient in surface supersaturation (Nelson, 2001; Frank, 1974). In slower-growing tabular crystals, Keller et al. (1980) observed disc crystals

faceting as they became larger and thicker, as did Knight (2012). In our previous experiments (Nelson and Knight, 1996), we also saw small disc crystals develop facets as they grew (Nelson, 2014). Also, Yamashita and Asano (1984) grew rounded tips of "serrated" dendrites at about −2.0 °C, and noted that they were thinner and grew faster than the thicker, facetted tips around −3.0 °C. Thus, rounding can occur in several situations. But, as argued in Nelson (2001), the supersaturations are too low for the phenomenon to arise from kinetic roughening, so we ask if AST may contribute to such rounding.

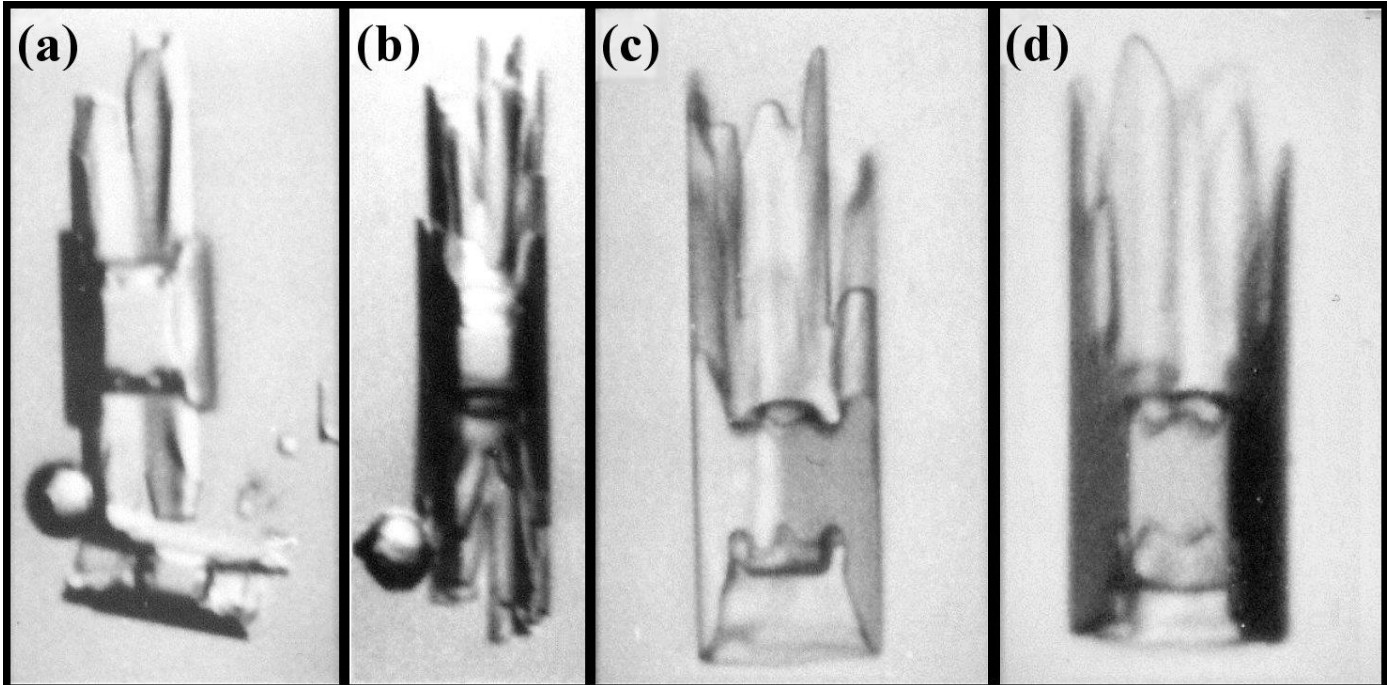

**Figure B7:** Initial sheath protrusions sprouted from droxtals of ~40–70 μm diameters at −4.8 °C (a,b) and −5.9 °C (c,d) and liquid-water saturation. (From the cloud chamber, courtesy of A. Yamashita.)

Consider first a thin tabular crystal with two basal faces, focusing on the region near a prism–prism edge as sketched in Fig. B8a. As shown in the top sketch, the collection area for AST flux at the tip **c** has an angle of just 120° compared to the 180° further down (at least a distance of ~$x_s$) at **e**, making the AST flux ~2/3rds the value at the tip. If the total molecular flux is dominated by the AST flux, then this effect would move the point of highest total flux away from the tip. Equivalently, we can view position **e** as having higher effective supersaturation than **c**. Thus, as shown in the bottom sketch, the point of new-layer nucleation has moved down from the tip to **e**. Moreover, if the edge region is effectively rough, then the regions of higher effective supersaturation will advance faster than regions with lower values, changing the vertex or tip shape until a steady-state shape emerges. Such a shape would be rounded, as shown in the sketch. Thus, if the crystal edge is nearly rough, then AST flux can cause the corner or tip to round as steps travelling from their source at **e** to further down the tip toward **c** can readily clump. In this case, $n = 2$ due to AST flux from the top and bottom basal faces. Thus, the edge plan view may be similar to that shown at the top in Fig. B8b.

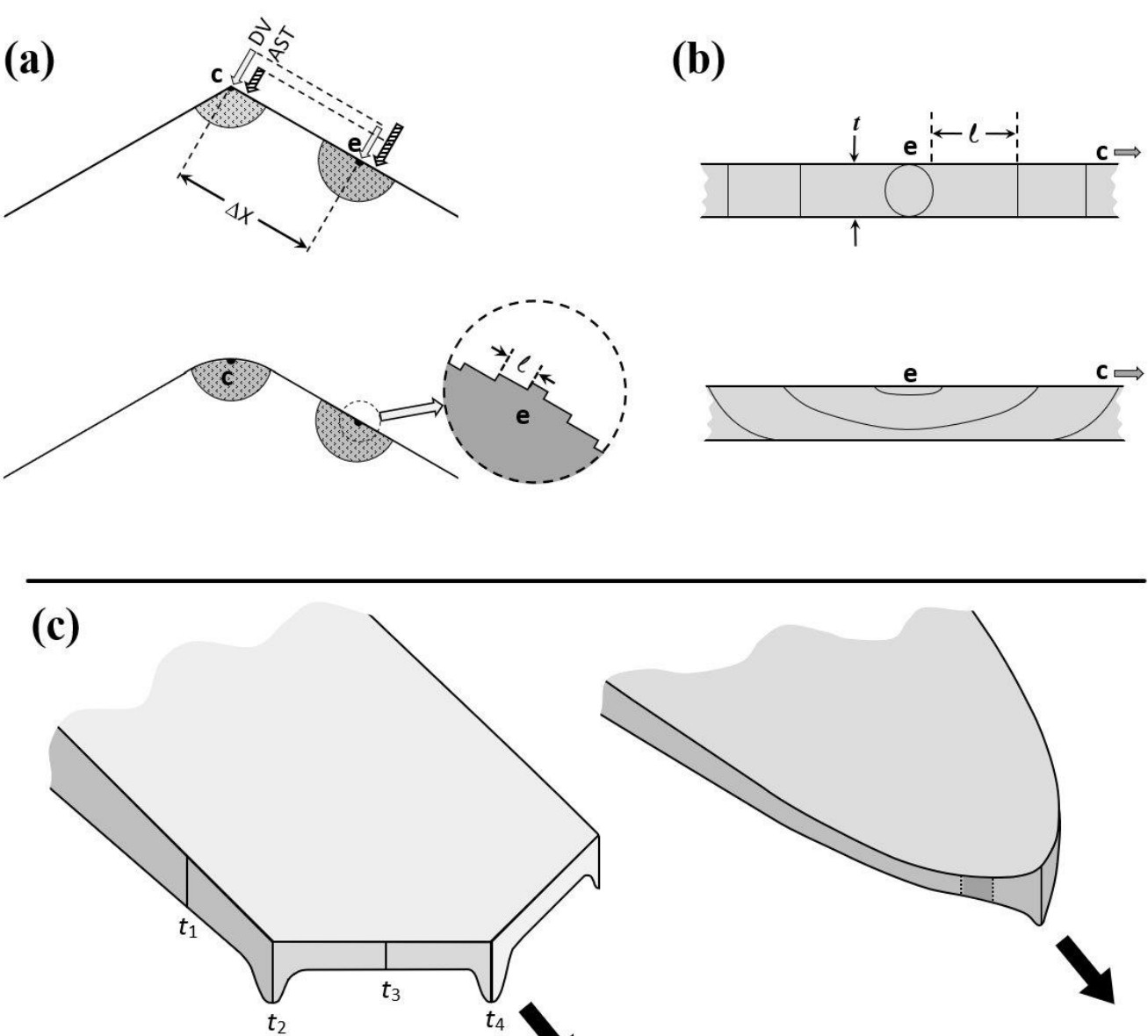

**Figure B8:** AST-induced rounding. (a) Direct vapor flux DV and AST at the corner **c** and edge **e** of a thin tabular crystal. Shaded regions are effective collection regions for AST flux, arrows show relative magnitudes. Inset at bottom is shown in top case of (b) from an edge-on view. (b) Nucleation and step flow on edge region of thin crystal in (a), plan view of edge. Top sketch is symmetric case. Layer-nucleation point **e** of (a) has steps moving away with initial separation $l$. Bottom sketch shows asymmetric case leading to step separation $l \sim t$. (c) Thin-branch case with non-crystallographic backside. Thickness $t$ is greater at the vertices $t_2$ and $t_4$ and thus has less growth from AST flux. Rounded case at right with small prism face bound by dotted lines.

A dendrite branch has a different tip shape, with the edge prism faces bound by just one basal face, the other being non-crystallographic. Thus, the edge plan view may be more like that at bottom in Fig. B8b. In this case, a second factor may cause the vertices to round. As sketched in (c), the leading prism faces are expected to be thicker at the vertices, that is, $t_2 > t_{1,3}$ and $t_4 > t_3$. (Although the thickness variation right at the perimeter was not detected by recent interferometry studies (Shimada and Ohtake, 2016; 2018), the formation of a main ridge and side ridge, which are clearly visible in nearly all tips, appear to require such an increase in thickness.) Their formation mechanism, as described in Nelson (2005), relies upon the influence of the direct

vapor flux. But if their growth is also significantly influenced by AST flux, and they are thicker at the vertices, then Eq. B4 shows that their growth rates will be slower. In this case, they may advance more slowly at the vertices, leading to the rounding shown in (c). In this latter case, a small region of prism facet remains (dark patch in (c)), but may be indiscernible in photomicrographs.

These rounding mechanisms depend upon the prism faces being nearly rough. One possible factor is illustrated by the bottom sketch in Fig. B8b. Here, layers nucleate at one basal–prism edge and can reorient parallel to the edge with a separation $l$ of roughly the crystal thickness $t$, which may make the edge effectively rough. For a face to essentially collect all surface ad-molecules, the step spacing $l$ need only be smaller than $x_s$. But if the spacing becomes significantly smaller than $x_s$, the normal growth rate becomes proportional to the local effective supersaturation (as all the flux is incorporated into the crystal), meaning

that a decrease in this supersaturation would cause the surface to respond by slowing, producing rounding. Such a case is akin to the kinetic roughening that is driven by high supersaturations (Elwenspoek and van der Eerden, 1987), but in this case the supersaturations are relatively low.

   Although AST may be a key factor in some of these cases of crystal rounding, rounding in general on vapor-grown ice is more complex than our simple treatment here. In different situations, crystal rounding on vapor-grown ice may involve a combination

of the above AST mechanism, thermal roughening, a thin solute layer, and perhaps yet undiscovered factors.

### B.8 Tip shapes of sheath and sharp needles

   Other aspects of tip shape may also influence the normal growth rates of needle crystals. Needle crystals (C1a) are long, thin, columnar crystals with "tops shaped like a knife edge" (Kikuchi et al., 2013) that form near −5 °C. In their initial growth, they appear as narrow prism planes that sprout from the edges of the basal plane, similar to the sprouting of basal planes during the

formation of two-level crystals examined above (§B.3). Examples shown in Fig. B7, as well as in Sei et al. (2000), suggest that they initiate via P-growth. Knight (2012) observed both sheath-needles and, less often, a newly reported type he called sharp needles. The sheath needles would grow at a rate of about 0.3–1.0 μm/s, whereas the sharp needles grew about twice as fast, about 1.5–2.5 μm/s. The sharp needles also had a smaller diameter and appeared more nearly round in cross-section. Why did they have different tips and why the distinct growth rates?

The different tip shapes may be the reason for the bimodal growth rate. If each side (prism plane) of the needle tips are of order 10 μm or less, the AST flux can be significant, perhaps even dominant. The net effect of this flux should depend on the ratio of the collection area (on the adjoining prism faces) to the growth-front area. Consider the two tip shapes at the bottom of Fig. B9. The left sketch, showing a 120° interior angle, is like the sheaths in Fig. B7, whereas the right sketch shows a 60° angle like that of "iii" in the right scroll of Fig. B10b in which one prism face is missing. If the arrows represent the AST flux, then the right

case of Fig. B9 has almost twice the AST flux to the growth-front area at the tip (dashed circle) than the left case; that is, $n$ is nearer 1 on the left, but nearer 2 on the right. Another factor is the influence of the backside of the tip on the surrounding vapor density. The backside, being non-crystallographic, is effectively rough and thus efficiently draws in vapor. Neither the front-side prism faces ($p_1, p_2$ or $p_1, p_5$) nor the tip are collecting much mass from the vapor. But for a given length of needle, the open sheath on the left has a greater backside area than the proposed sharp needle on the right. This backside may dominate the mass-uptake,

just as it appears to do on dendritic crystals (Nelson, 2005). With the smaller area for the sharp-needle case comes a smaller mass uptake, and with a smaller mass uptake, a higher surface supersaturation and higher normal growth rate. In this way, the sharp needle should grow significantly faster than the sheath needle, as was observed. An approximate calculation (to appear in part II) finds these effects cause an increase in rate by about 50%. A possibly larger effect, though harder to accurately model, may come from the direction of the supersaturation gradients between the two needle cases. This gradient is likely larger in the sharp-needle

case due to the smaller interior angle, which would have the effect of sharpening the needle, which in turn could greatly increase its growth rate.

Consistent with our hypothesis that the two case have different interior angles, Knight (2012) found that when the temperature of the sharp needle was raised to slow down growth, it thickened enough to discern that it had a triangular shape (Fig. 4 of Knight, 2012). This 60° interior angle of the proposed sharp needle is a feature of trigonal crystals. In Appendix B.11 below, we argue that this angle is stable in the columnar regime, which includes the needle case.

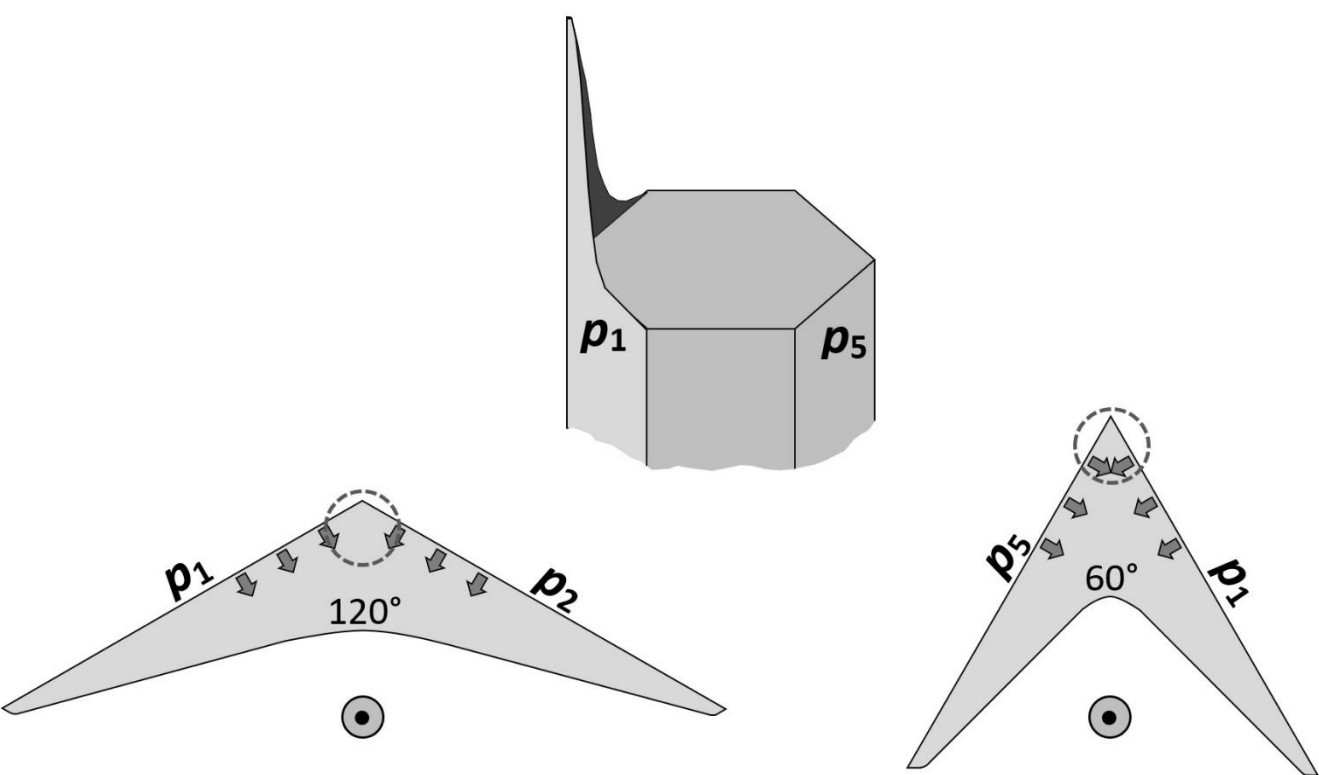

**Figure B9:** Needle-tip cross-sections. Prism face $p_1$ through $p_6$ run clockwise around crystal sketch in middle-top. Bottom shows views looking straight down. Left: case from sheath with prisms $p_1$ and $p_2$. Right: case from sheath initially with prisms $p_1$, $p_6$, and $p_5$, but middle prism $p_6$ vanished. Arrows show AST flux from adjoining prism faces.

### B.8 Scroll crystal features

A perplexing growth feature is the scroll (C3c). With a scroll feature, thin prism-face "sheets" or side-planes tend to curl inward while maintaining a prism orientation, somewhat resembling a paper scroll as shown in Fig. B10. (For formation sequences, see Figs. 3 and 5 in Sei et al., 2000.) In the atmosphere, the original Nakaya diagram (e.g., Nakaya et al. 1958) shows scrolls forming at and above water saturation near −7 °C, which is consistent with later diagrams by other authors. Later, Nakata et al. (1992) found scrolls to be very common features of certain polycrystals (Gohei twins, CP7a). But perhaps one of the earliest descriptions of a scroll is in Seligman (1936), where he finds large examples in crevasse hoar. However, with the aid of a standard macro lens on a common digital camera, one can observe them frequently in hoar frost. A description of their formation is briefly mentioned in Higuchi et al. (2011), but their proposed mechanism differs from that presented next.

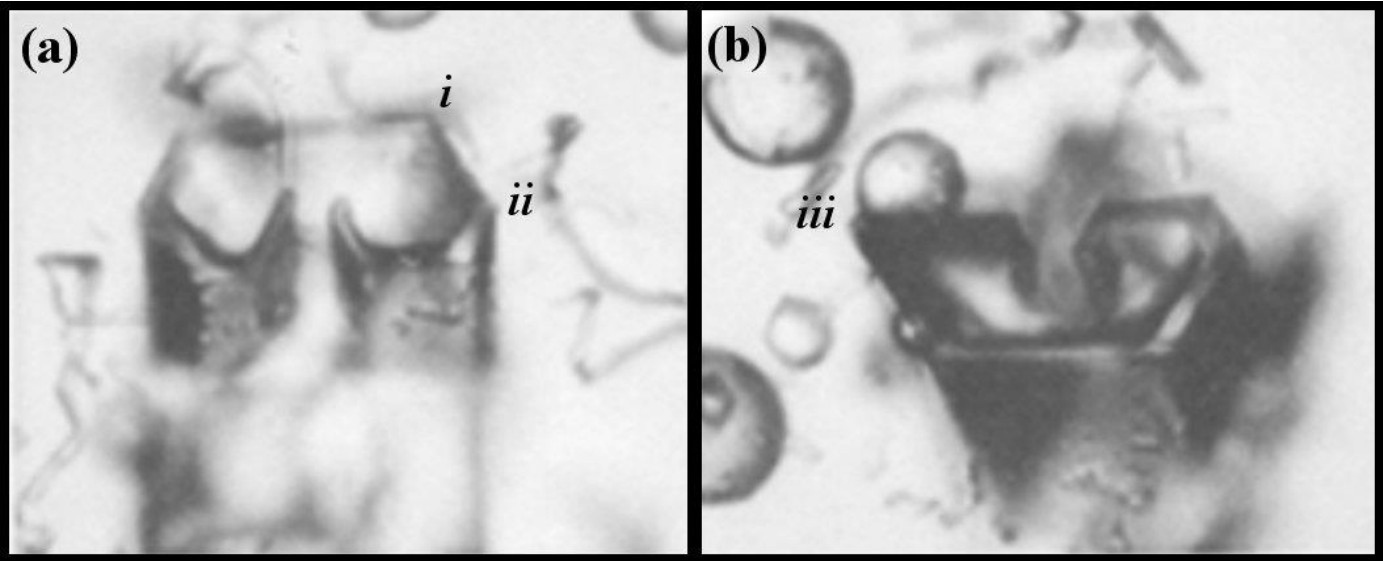

**Figure B10:** Scrolls formed on crystals of approximate diameter of 60 µm. (From the cloud chamber, courtesy of A. Yamashita.)

The scroll may start as a protrusion, like a sheath, except on a larger-area basal face as shown in Fig. B11a. When this protrusion is thin, it can grow rapidly because the AST flux is depositing onto a small-area region at the growth front. (Growth may be more rapid normal to the page, but we focus on the side growth.) This rate is marked by the large arrow pointing across prism $p_2$ in (a). But normal growth is also occurring on the backside (or inside), causing it to gradually thicken. (In this description, we assume that the backside, or inside, of the scroll is mostly non-crystallographic.) This normal growth rate is marked by the smaller arrow pointing down, towards the basal interior. On the leading front of the protrusion, direct vapor flux is also contributing to growth, but when the feature is thin, this flux is overcome by the larger AST flux from prism $p_2$ (Eq. B4). Eventually, this protrusion thickens enough to reduce the protruding growth rate, at which point prism facet $p_3$ can form and spread there, essentially ceasing the lateral growth of $p_2$ as shown in the sketch. Now the process starts on the edge of these new prism faces, causing new protruding growth at 60° to the old protrusive growth (right side), and the process repeats, later curling around another 60° into a scroll-like shape. Each new "wing" of the scroll may have a smaller area due to it moving towards lower vapor density, allowing the edge-front faceting to start sooner. Such a process would produce a curling feature. In brief then, the mechanism involves normal growth, F-growth, and P-growth.

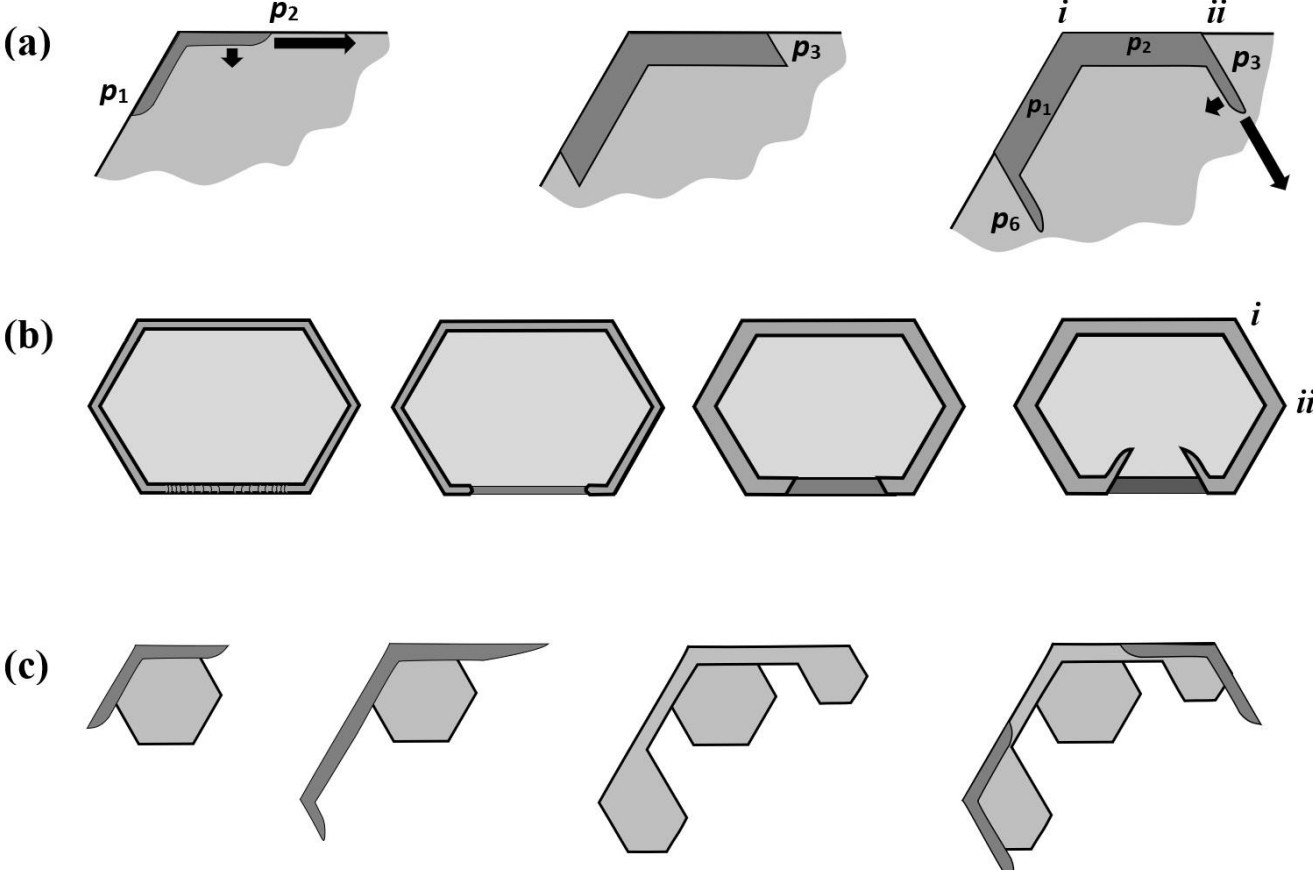

**Figure B11:** Scroll and sheath bundles via protruding and normal growth. (a) The basic process, viewed along c-axis (normal to basal). The darker region is a sheath protrusion from prisms $p_1$ and $p_2$ going up out of the page. Arrows indicate relative rates of growth in width (via AST) and thickness (via normal growth). In middle sketch, protruding region has thickened, slowing the rate of protruding growth, allowing significant lateral growth on the ends that leads to new prism facets $p_3$ and $p_6$. In the right sketch, protrusion growth occurs on these new prism faces. The process can continue (not shown), generating facets $p_4$ and $p_5$. Compare *i* and *ii* vertices to ones in Fig. B10. (b) A complete sheath has its rim broken at the bottom (left), due to a vapor-density gradient or asymmetry. The process in the next three sketches follows the same process as explained in (a). Vertices *i* and *ii* correspond to those in Fig. B10. (c) Possible source of sheath and needle bundles from protruding-growth overshoot. Leftmost sketch follows from start of case (a) except protruding growth overshoots the base. (Overshooting is exaggerated to clarify the concept.) In middle sketches, the edge of the protrusion thickens and facets as in case (a). Far right, process repeats.

A scroll may also initiate from a cup-type crystal form or sheath (C2a) after part of its rim forms a break, which may in turn be due to an axial asymmetry in the nearby vapor-density field. As sketched in Fig. B11b, the sides of the break (at bottom) could then curl around via the same process as that sketched for (a). Supersaturations high enough to produce cup crystals are rare in the atmosphere, but are very common near hoar frost. As hoar cup crystals tend to be closely clustered, and thus having large local variations in vapor density via crystal competition, this initiation process may be likely for hoar scrolls. Large examples of this type are in Knight and Devries (1985). This mechanism may have produced the crystal in Fig. B10a (compare vertices *i* and *ii*).

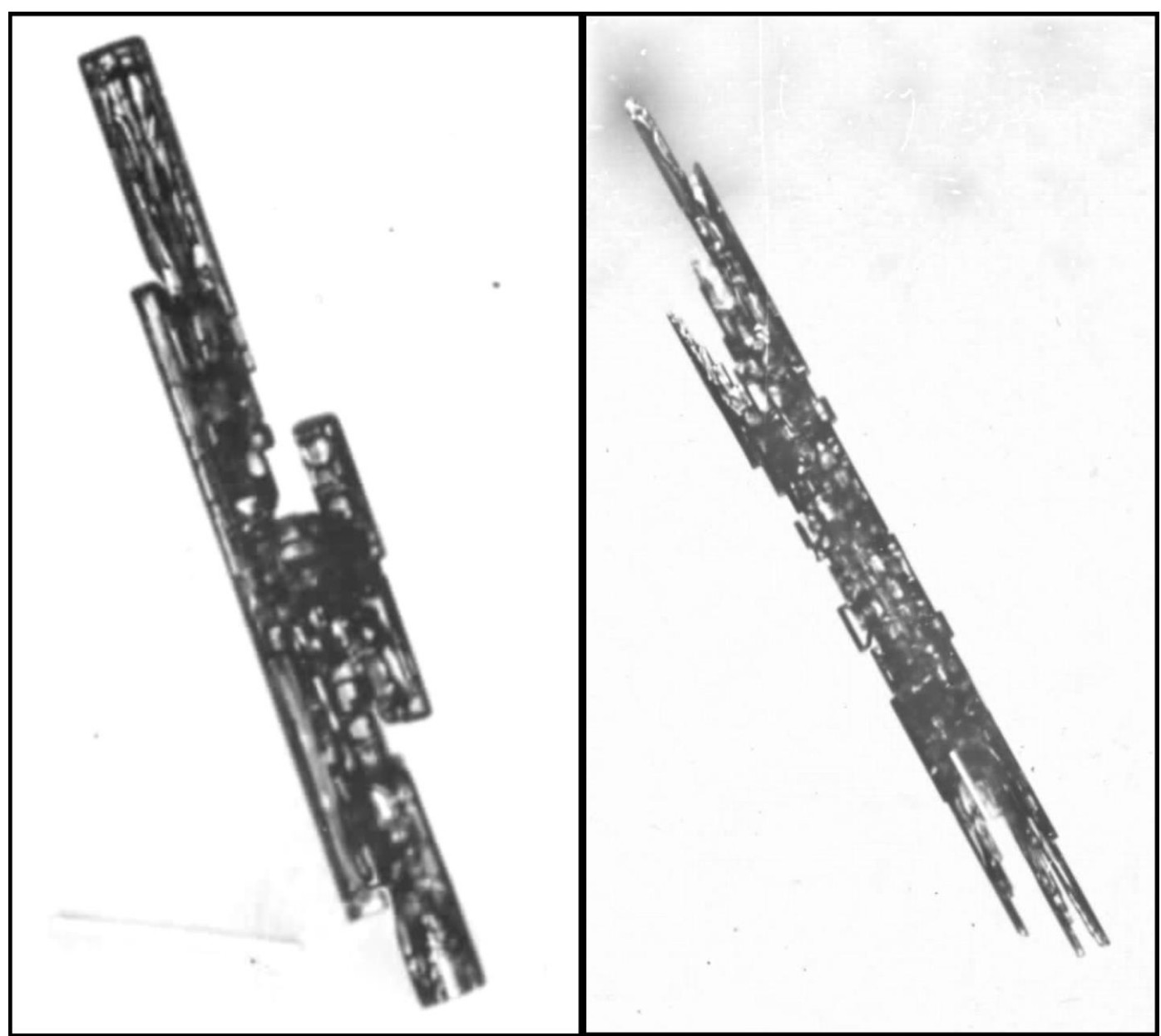

**Figure B12:** Bundle of sheaths (left) and bundle of needles (right) from the Magono–Lee collection (Magono and Lee, 1966).

## B.9  Bundles of sheaths and needles

Another perplexing growth form is the bundle of sheaths (C2b), and similarly, the bundle of needles (C1b). These habits form in
5  a narrow temperature regime near −5 °C where growth is almost exclusively in the c-axis direction (e.g., Takahashi et al., 1991).
However, unlike the needles, these crystal forms have widened significantly perpendicular to the c-axis. Such widening is hard to
reconcile with our knowledge of growth driven by layer nucleation, which may effectively shut off all normal growth of the
prism faces. Two mechanisms may overcome this nucleation barrier. One is riming. A rimed drop on a prism plane may then
sprout a new sheath protrusion along the c-axis. The second mechanism is sketched in Fig. B11c. In this mechanism, the
10  protruding sheath widens as in the scroll form in (a), but overshoots the base crystal, thus advancing the crystal width
perpendicular to the c-axis. The protrusion will thicken, and may then develop a new prism face by the mechanism suggested in
(a) for the scroll. In this way, a new sheath or needle can develop to the side of the original, producing a "bundle". Such
overshooting of the prism planes in this case is analogous to the overshooting of the basal planes in the two-level crystal (§B.3).

## B.10  Protruding growth on branch backsides and ridge pockets

The backsides of branches on tabular crystals appear to be largely non-crystallographic during growth under constant conditions. But they are not gently curved; rather, various ridges and ribs are common, which show up as dark interior lines in images. However, when part of a relatively fast-growing crystal branch slows down, due to either a change of conditions or to the gradual drift toward the crystal interior as the outer parts grow out, the ridges and cross-ribs may form planar protrusions. For example, such protrusions appear relatively common on the slower-growing planar crystals in Takahashi (2014) at temperatures of −12.5 and −16.3 °C. Libbrecht (2006) refers to them as "aftergrowth plates", though they form while the crystal is growing. Examples are marked with arrows in Fig. B13a. Also, fairly common are long pockets that we call ridge pockets. Ridge pockets include the main-ridge pockets aligned towards the vertex and coming in a pair, as well as the side-ridge pockets aligned toward each side and generally having numerous pairs. Several examples appear in Fig. B13b. This image shows main-ridge pockets at A (enlarged inset upper left) and side-ridge pockets at B.

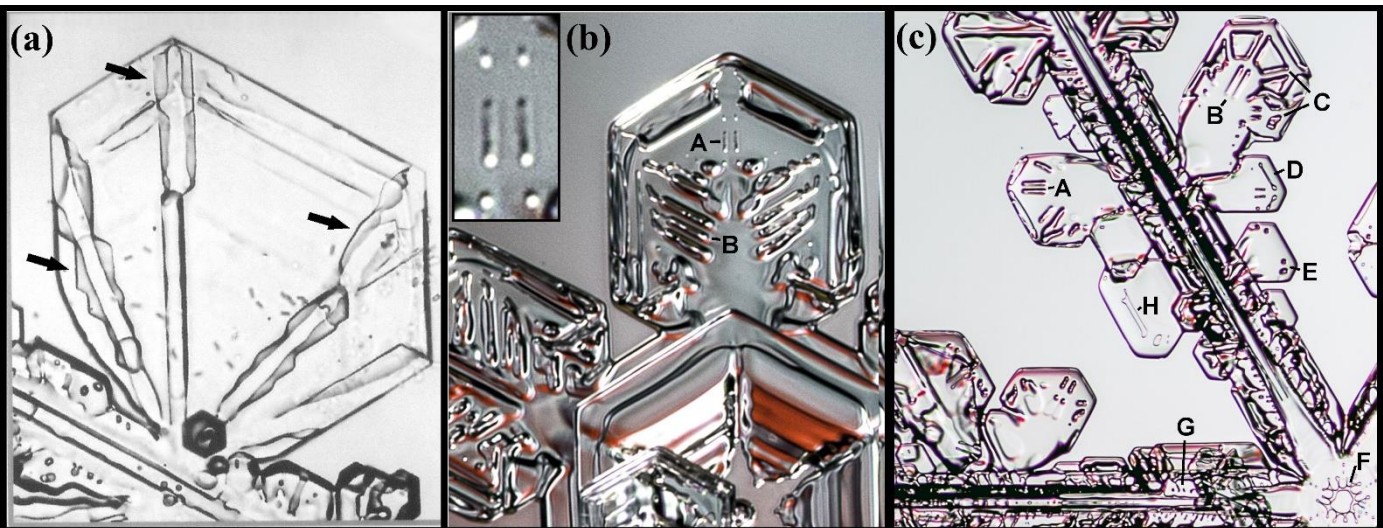

**Figure B13:** Protruding growth on branch backsides and branch pockets. (a) Sidebranch of dendrite. Arrows show protrusion features on ridges. (b) Branch of broad-branch crystal. Inset is close-up of main ridge pockets A on one branch, B marks side-ridge pockets. (c) Branch of dendrite. Sublimation has apparently exposed several pocket features A–H described in text. Image in (a) from large hoar crystal grown in an unforced air-flow cloud chamber of Yamashita and Ohno (1984). Crystals in (b) and (c) from snow crystals collected at ground level (courtesy of Mark Cassino).

The dendrite branch in Fig. B13c shows numerous pocket features. (This crystal has undergone significant sublimation just before imaging, and thus most of the pockets appear to have re-opened. But it provides clear examples of pockets seen on other crystals.) Here, A and B mark main ridge pockets. C marks an example in which pockets can form where a main ridge meets a thickened cross-rib. At bottom is a pocket and at top is a position that may produce a pocket by such an intersection. D appears to be an elongated edge pocket (§4.6). E and G appear to be side-ridge pockets, and F is a center ring pocket (Yamashita, 2018). H may be a center pocket or an edge pocket like D. The main ridge pockets at B fade towards the main branch, indicating that the base of the pocket has a downward slope towards the tip. In images such as these, which show only one view, it is difficult to be sure that the features are pockets as opposed to channels or small hollows. Nevertheless, they appear to be common and whether or not they are pockets or channels, their effect on light scattering in clouds may be similar.

The ridge pockets may form as sketched in Fig. B14. The branch backside often has a main ridge from the tip, shown in cross-section at (a). As pointed out by Frank (1982), the ridge produces a vapor-shadowing effect leading to the two parallel channels to both sides in the next sketch. These channels are clearly seen in images of most branched snow crystals and was noticed much

earlier by Nakaya (1954). With a local growth slow-down, which may arise simply due to the branch tips growing further out, thus depleting the crystal inner regions of vapor, facet spreading may start to dominate, transitioning to protruding growth in the next two sketches. This may continue and eventually close-off the channels making the two main ridge pockets at the bottom in (a). The mechanism for side-ridge pockets is sketched in (b). The process is like that of the main ridge, except side ridges can form close together, creating a central channel gap. Protrusion growth at the crest starts bridging the gap, and completes the pocket at the bottom. These sketches do not show the 3-D structure of the branch, in which the top surface slopes downward towards the thinner outermost prism faces at the tip. Thus, the protrusions likely also grow from the interior region towards the tip (as in the corner-pocket case), eventually covering the entire backside.

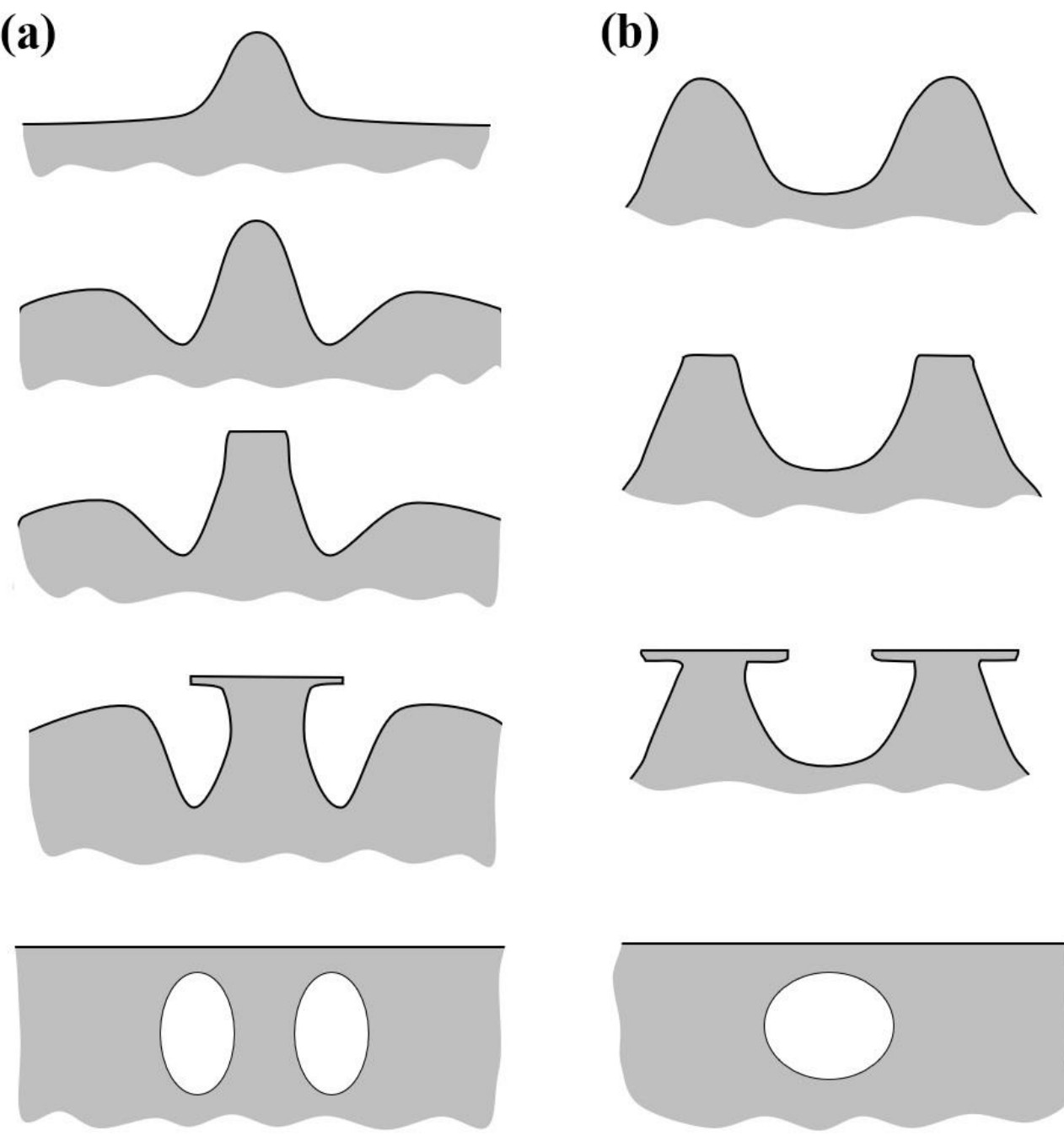

**Figure B14:** Development of ridge channel pockets. (a) Adjacent to main ridge. (b) Between side ridges. Compare to A and B in Fig. B13b.

## B.11 AST contributions to trigonal formation and primary habit

Trigonal crystals have only three clearly observable prism faces, a striking feature that begs for an explanation. Another crystal type with three-fold symmetry is the scalene hexagonal, which is shown together with the trigonal type in Fig. B15. Bentley (1901) finds both types rare in precipitation, but Heymsfield (1986) reports on a very cold cloud in which roughly 50% of the crystals had three-fold symmetry. In the laboratory, Yamashita (1973) found that numerous trigonal and scalene forms would result from seeding with an adiabatic-expansion method that creates sub-micron ice nuclei, but only hexagonal crystals would result from nucleating cloud droplets of much larger size. This finding may explain the difference in Bentley's and Heymsfield's findings because the latter observations were of crystals that likely formed on sub-micron droxtals. In Yamashita's sub-micron seeding experiments, the crystals grew at temperatures down to about −26 °C. The trigonal forms appeared stable when columnar; for example, about 10–20% of all crystals were trigonal in the columnar regime above −10 °C. But the trigonal tabular forms appeared to transition to the scalene hexagonal at small sizes, with the latter types occurring in over 40% of the crystals at all temperatures except around −12 to −18 °C. In all cases the trigonal were more common than rhombohedral and pentagonal forms.

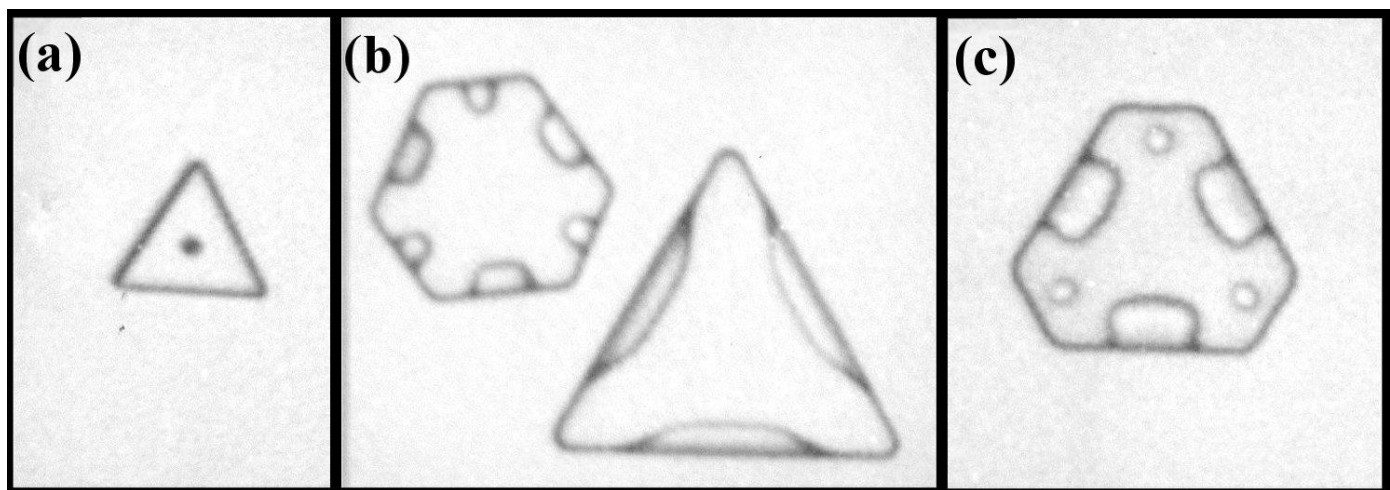

**Figure B15:** Trigonal and scalene hexagonal. (a) Trigonal. (b) Both types. (c) Scalene hexagonal. All crystals grown for about 310 s at −10.2 °C in the cloud chamber (courtesy of A. Yamashita). Diameters are about 15–35 μm.

A recent review proposes three possible explanations for trigonal formation (Murray et al., 2015). In one, they suggest that stacking disorder can lead to growth of trigonal forms, but do not give a specific formation mechanism. Stacking faults in cubic crystals can produce trigonal forms (e.g., Millstone et al., 2009), but the mechanism involves growth on alternating re-entrant corners that have not yet been shown to occur in stacking-disordered ice. Also, it is not clear how such a mechanism would explain Yamashita's observations above. Our observations here (discussed in §B.2) suggest that regions of stacking disorder may instead lead to near-symmetric hexagonal forms (e.g., Fig. 3). The second explanation involves having equivalent dislocation step sources on just three alternating prism faces (e.g., Sei and Gonda, 1989; Wood et al., 2001). Such a mechanism cannot explain the preponderance of trigonal over rhombohedral and pentagonal forms. The third explanation involves aerodynamic

factors (Libbrecht and Arnold, 2009) that would influence habit more for tabular and larger crystals. Such an explanation also appears inconsistent with Yamashita's findings above, specifically the greater stability of the trigonal form on columnar habits and the transition from trigonal to scalene hexagonal as the tabular forms grew larger. Instead of these proposed explanations, we suggest a closer look at the growth mechanism, focusing on two factors: a mechanism for their initial formation in sub-micron

droxtals, and a mechanism for their stability as they grow larger.

A possible explanation for the formation of an initial trigonal habit from a sub-micron droxtal is sketched in Fig. B16a–d. When the sub-micron droplet freezes, one prismatic plane must start forming first. Assume, as in sketch (b) that it is $p_1$ on the left side. If the crystal is smaller than $x_s$, facet spreading of $p_1$ may be dominated by AST yet increase in rate as the face expands. This rate would increase in proportion to the increase in area because all the vapor impinging on the area can migrate to the edge.

Thus, facet spreading would greatly favor the first face that develops. Moreover, this growth may overshoot, and effectively bury, the neighboring faces $p_2$ and $p_6$. If the next face that develops is either $p_3$ or $p_5$, then face $p_4$ would be similarly buried as shown in (c). As $p_3$ and $p_5$ expand, the crystal fills out as a trigonal form with only $p_1$, $p_3$, and $p_5$ faces as shown in (d). In this way, if after step (b), the three remaining faces were equally likely to form next, then the likelihood of a trigonal would be twice that of a crystal with four prism faces. Experimentally, the trigonals formed much more frequently than those of the

rhombohedral and pentagonal (except at −4.2 °C), and thus the formation of the second prism face may depend on the formation of the first. For example, the region marked **u** in sketch (b) may immediately develop into $p_3$ (ditto for $p_5$ on the other side), leading to trigonal in all cases. Or, a small crystal with four prisms may be unstable compared to one with three. Regardless, the basic mechanism would lead to trigonal crystals in most cases.

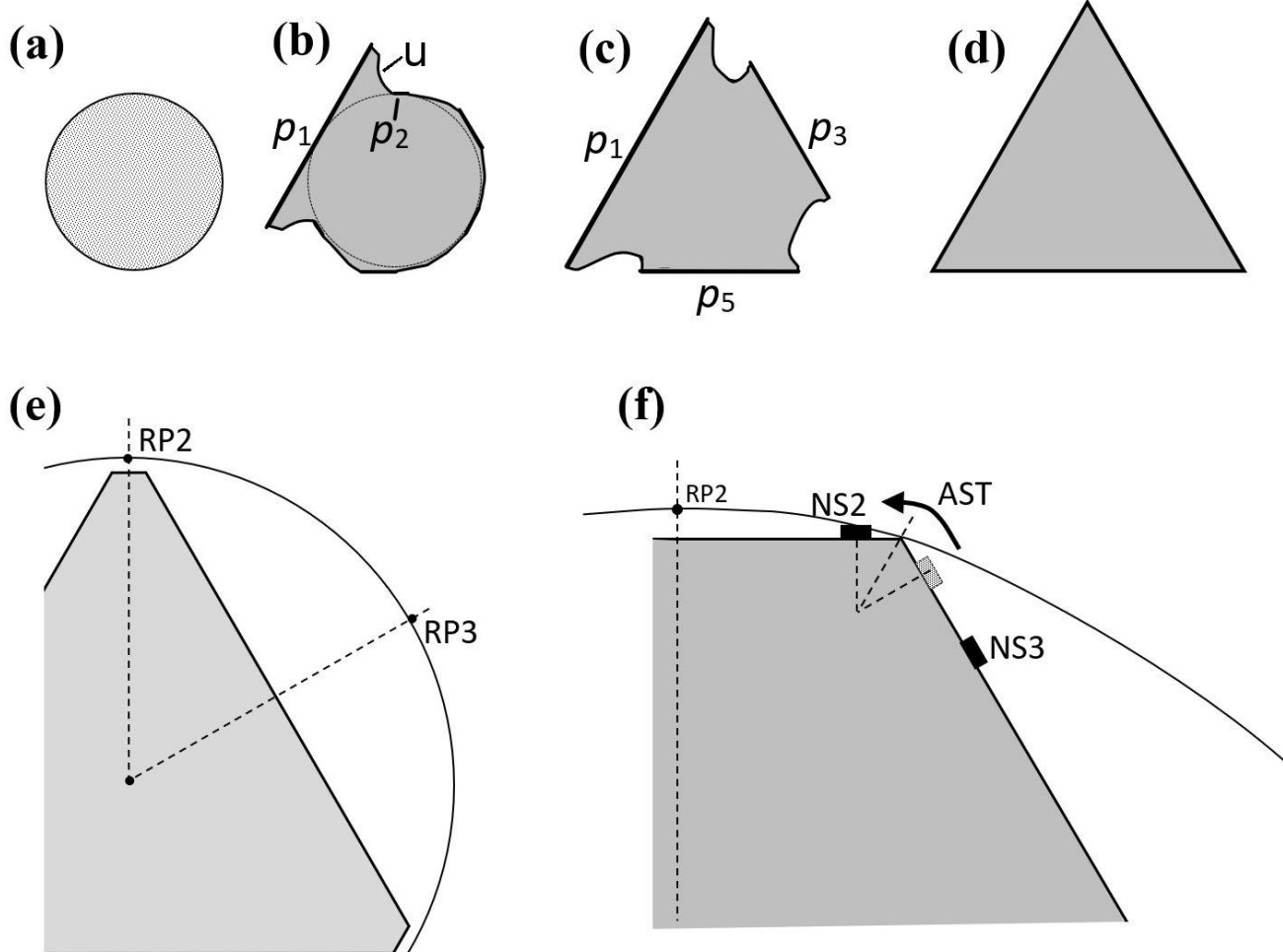

**Figure B16:** Formation and growth of a trigonal crystal. (a) Droplet before freezing. (b) Upon freezing, prism face $p_1$ forms before the others, growing laterally via AST, stunting neighbors $p_2$ and $p_6$. The region "u" may have a range of slopes, sometimes lining up along $p_3$. (c) Prism face $p_3$ or $p_5$ develop next, also stunting $p_4$. (d) Trigonal forms. (e) Vapor density contour around nearly trigonal form (scalene hexagonal) with small prism faces. RP2 and RP3 are points on $p_2$ and $p_3$ where a reflection-symmetry plane crosses the contour. (f) Close up of corner between prisms $p_2$ and $p_3$. Solid boxes are the layer nucleation sites NS2,3. Shaded box on $p_3$ is the nucleation site in the absence of AST.

The trigonal form may then be maintained via an effect of AST and the vapor-supersaturation contours on layer nucleation. The supersaturation contours around the crystal, viewed in the plane of Fig. B16e, should have the same symmetry as the crystal. In the sketch, the contour is a circle, but in general needs to only have reflection symmetry about the dashed lines from the center to points RP2 (reflection-symmetry point, face 2) and RP3 as shown. (Far from the crystal, the contour will be a circle, but closer to the surface, the lines will bend closer to the surface as suggested in (f). The consequence is the asymmetry in the contour about the vertex between $p_2$ and $p_3$, as shown. Assuming that the growth is via layer nucleation, with layer nucleation points near to, but not exactly at, the vertex, then such points on either side of the vertex will experience different vapor supersaturations. In particular, point NS2 on face $p_2$ will have greater supersaturation and thus nucleate new layers at a faster rate than at the corresponding point on face $p_3$. This factor will increase the normal rate of growth of $p_2$ over that of $p_3$.

This factor, though small, can be amplified by AST. Consider that the rate of layer nucleation depends on the density of surface-mobile molecules, not the adjacent vapor density directly. Thus, as indicated in (f), the slightly faster production of new layers at NS2 will draw a net AST flux from face $p_3$, thus reducing the surface ad-molecule concentration there. As a result, the

layer nucleation point NS3 must move further away from the vertex, as shown in the sketch, further reducing the layer-nucleation rate on face $p_3$. In this way, the normal growth rate of $p_2$ can significantly exceed that of $p_3$ even with a relatively small vapor-density asymmetry, driving its area lower. The difference in growth rates between the faces may lead to the smaller face becoming relatively smaller or larger, depending on the ratio of the rates. (With a little trigonometry, you can readily work out the condition for a relative decrease.) But if the face area shrinks, the effect here may increase, causing further shrinkage; conversely, if the face area grows, the effect may weaken, leading first to the scalene hexagonal and then to fully hexagonal. This AST effect on the relative layer nucleation rates between adjoining faces was previously proposed by Frank (1982) to explain the abruptness of primary-habit change with temperature. As he suggested, it should apply in general to the basal–prism edge as well, influencing the primary habits (aspect ratios) of snow crystals in general.

About the higher stability of columnar trigonal forms, consider the magnitude of the effect. The magnitude should depend on the size of the mean migration distance $x_s$ on the prism faces compared to the crystal size. When the value of $x_s$ is a significant fraction of the large-face diameter (e.g., 0.1 or more), the effect is likely to be stronger as NS3 is pushed further from the vertex. In contrast, when $x_s$ is much smaller, then the shift of NS3 will be insignificant. The values of $x_s$ for the prism face are unknown, but Mason et al. (1963) argued that they should be relatively large in the columnar regime (compared to the tabular regime). Such a trend in $x_s$, if verified by experiment, could the explain the higher stability of trigonal columns as well as the transition to scalene hexagonal for the tabular case. The instability of the tabular case here is also consistent with the argument that an imposed gradient in supersaturation has little effect on the direction of tabular branches (Nelson and Knight, 1998), that is, prism faces adjacent to a supersaturation maximum should grow at the same normal growth rate (unlike the case in Fig. B16f) because in the tabular regime, $x_s$ for the prism face would instead be relatively small.

In addition, the AST flux from the basal to the prism should be smaller towards the narrow prism in a scalene hexagonal crystal than to the wider prism (following §B.7 above). This effect would further de-stabilize the tabular trigonal and scalene hexagonal forms, particularly for the thinner tabular crystals, but have less effect on the columnar crystals. Concerning the role of the vapor mean-free path, this mechanism for the stability would have a vanishing role when the vapor mean-free path exceeded the crystal size. But in an atmosphere of air, this condition would require crystal diameters below a few tenths of a micron. Finally, a greater sensitivity of the layer nucleation rate to supersaturation would increase the influence of the AST flux, strengthening the mechanism. In general, the supersaturation in a cloud is higher when the first crystals nucleate, and is also higher at the crystal surface when the crystal is small, but as each crystal grows and more crystals nucleate, the supersaturation drops. This effect also predicts a transition from trigonal to scalene hexagonal, and eventually, to hexagonal. However, if the crystal develops branches while still scalene hexagonal, the nearly three-fold symmetry should remain as the branches grow independently of one another. This may explain the large, nearly three-fold symmetric branched crystals in Bentley's collection (Bentley, 1924; Bentley and Humphreys, 1962).

Finally, consider what would result if, instead of $p_3$ or $p_5$ developing after $p_1$, that $p_4$ developed in Fig. B16c. In this case, one can argue that the resulting crystal would have two large-area prisms $p_1$ and $p_4$, with the latter smaller than the former, and just two other equal-sized faces $p_3$ and $p_5$. That is, the shape in cross-section would be an isosceles trapezoid. Such a shape falling with $p_4$ side down could generate the suncave Parry arc in a thin cloud. Upon growing larger, the $p_2$ and $p_6$ faces would likely develop, and then regardless of whether the falling orientation had $p_4$ side down or up, a suncave Parry arc would result (e.g., Westbrook, 2011). A sampling of crystals from a Parry-arc display found no evidence of the trapezoid form (Sassen and Takano, 2000), so such a form may transition to the six-sided form while still small.

*Author contributions*. The experiments with the capillary apparatus were done by JN and BS, Text, figures, and calculations were prepared by JN, with input from BS.

*Competing interests*. The authors declare that they have no conflict of interest.

*Acknowledgements*. The National Science Foundation provided support for this research through grant #1348238 from the Division of Atmospheric and Geospace Sciences (AGS). We also thank the Laucks Foundation for research funds, equipment, and laboratory space. JN thanks Charles Knight for discussions about the distinction between lateral and normal growth. We thank Prof. Akira Yamashita (AY) for kindly supplying the images for Figs. 1, B3, B7, B10, B13a, and B15. We also thank Art Rangno for supplying the original digital copies of the Magono–Lee collection shown in Figs. B5 and B12. Mark Cassino and Martin Schnaiter also kindly supplied useful photographs. The concept of protruding growth is from AY, as is the basic mechanism of trigonal initiation on sub-micron droxtals, and two-level formation. JN also thanks AY for numerous discussions of protruding growth and pocket formation. Some crystal images were processed in ImageJ using auto-contrast and background subtraction.

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
