# Peer review of "Lateral facet growth of ice and snow I: observations and applications to secondary habits"

_Atmospheric Chemistry and Physics, 2019_

## Referee Comment (RC1) · Anonymous Referee #1 · 26 Apr 2019

The authors studied the formation mechanisms of air pockets and other secondary habits in snow crystals. The topics coincide with the scope of Atmos. Chem. Phys. Discuss., and the secondary habits of snow crystals are interesting from the fundamental viewpoint. However, first of all, this manuscript is too lengthy (26 figures are shown in the manuscript of 51 pages in total), and too many subjects are included in a mixed-up way. Therefore, I need to say the presentation quality is poor. Second, the authors insist that the formation of corner air pockets (the main subject in this manuscript) cannot be explained by the traditional growth mechanisms based on lateral step motion, and by the morphological instability based on the inhomogeneous distribution of vapor density. Then the authors conclude that the lateral-type (protruding) growth, which is the key mechanism in this study, is a novel concept. However, I cannot agree with such

authors' claims (for details, see the comment 2). Hence, I believe that the scientific significance of the present manuscript is also poor. Since the amount of revisions is significantly large and the conceptual revisions are necessary, I do not recommend the publication of this manuscript.

Following 1-3 are major comments. 1. Too lengthy: one paper should have one main claim. Then, I believe that the following topics should be presented in separated papers: # the formation mechanism of the corner air pockets, # quantitative discussion about the kinetics of the lateral-type (protruding) growth, and # secondary habits other than the corner air pockets (these topics can be also moved into supplementary information)

2. The formation mechanism of the corner air pockets: the authors mentioned that the normal growth via step motion and the standard hollowing theory based on the morphological instability caused by the inhomogeneous vapor density cannot explain the formation of the corner air pockets. Then, I shall explain the formation of the corner air pockets by the traditional concepts. The key is the morphology of a snow crystal at the beginning of the growth. 1) When a starting crystal is fully faceted, the local vapor density becomes maximum at the corner of the crystal, providing a hollow not at the corner but at the center of the crystal face, as the authors explained. 2) In contrast, when a starting crystal is partly rounded, the layer-by-layer growth of the faceted face (located at the center of the crystal) proceeds. Then a spreading edge appears as shown in Fig. 1b (marked by e) and Fig. 8c. Since the spreading edge shows an angular shape and the corner of the crystal is still rounded, the local vapor density at the tip of the spreading edge becomes higher than that at the rounded corner, providing the overhang as shown in Fig. 1d and Fig. 8d. After once the overhang was produced, the overhang is developed spontaneously (the authors call this process the protruding growth), and the corner air pocket is formed. These processes never violate the traditional concepts of the layer-by-layer growth and the morphological instability. In addition, the authors emphasize the importance of the diffusion of admolecules on

the crystal surface (the surface diffusion of admolecules). Then the authors named this process "adjoining surface transport (AST)". I fully agree with the importance of the surface diffusion for the formation of the overhang and the subsequent protruding growth. However, the concept of the surface diffusion of admolecules is very traditional (firstly proposed by Frank and coworkers in the 1960s, and then experimentally proved by the growth of various crystals). Therefore, the authors should clearly show what is the authors' novel concept and what is not.

3. Throughout the manuscript, the authors should clearly explain what is the authors' new finding and what is not, with respect to phenomena and formulas as well.

Followings are minor comments. 4. The term "droxtal": since many readers (including me) are not familiar with this term, the authors need to explain it properly at the beginning.

5. The section 1.2 gives the impression that the authors do not fully understand the fundamental growth mechanism of crystal growth. The concept of the surface diffusion of admolecules (AST) is widely accepted in the crystal growth of wide variety of materials: not only for the metal whiskers, but also for semiconductor crystals, molecular crystals and ice crystals as well (as studied by Hallett, Mason et al, Kobayashi, and Asakawa et al.) Hence, for me, the application of the surface diffusion to the lateral and protruding growth by Yamashita (2015) does not look a significant revision, since the lateral and protruding growth can be explained easily, as shown in my above-mentioned comment 2. By the way, it is impossible to obtain the reference Yamashita 2015. The page numbers of the references Yamashita 2013 and 2016 should be 165-176 and 393-400, respectively: the page numbers of 23-33 and 15-22 are those in the issues 60 (3) and 63 (5).

6. The authors should show the schematic illustration of the new crystal-growth apparatus (CC2) in this study, since the reference Swanson and Nelson 2019 is still in preparation.

7. In the section 3.4, from Fig. 4, I cannot understand the difference between the expanding boundary of the basal face and growing macro-steps. The authors also should clearly explain the kinetic models I, II and III in the main text (of a separated paper), since the quantitative discussion has no meaning without obtaining the complete understanding of the models. By changing the value of h/xs arbitrarily, one can easily fit the experimental data. Hence, here the authors need to explain the causes of the change in the value of h/xs (I believe that the cause of the change is the evolution of macro-steps) and also whether the change is appropriate or not. The authors also need to discuss the values of h and xs.

8. In the section 3.9, the authors should explain the impossibility and instability much more in detail.

9. The sections 3.10-3.22: If these sections have scientific significance, the authors should explain them in separated papers. If their scientific significance is not so large, the authors should move them into the supporting information.

---

## Referee Comment (RC2) · Anonymous Referee #2 · 30 Apr 2019

The paper "Air pockets and secondary habits in ice from lateral-type growth" presents extensive collection of micrographs of growing ice. I am very pleased by its esthetical beauty. I think it should be published for its experimental value, regardless the theoretical explanations. The explanation of air packets formation is elegant. I cannot judge on its correctness. Nevertheless, I have some suggestions steaming mostly from the fact that I am not an expert in the field of ice grow from the vapor phase and thus I would welcome some introduction and generalizations. That may be a case for most of the readers, though. I suggest the paper be published after considerations the comments.

I would think that such extensive work deserves broader introductions and connections to what "general ice knowledge" may cover. I started form Hobbs: Physics of ice Ch. 8: I found the description of the growth in the direction of c and a-axis easier to understand

then to consider the basal face and prismatic face in presented manuscript. Can both description be shown in the pictures?

In Hobbs (ch. 8.3) the linear growth (here named normal) is defined as normal to crystallographic face. Is the here discussed lateral growth perpendicular to both c-axis and a-axis? Would not that be more exact definition than that given on the line 5 of page 2?

I think, that schema in Figuer 1a suggests that the droxtal has 8 prismatic phases –should not there be only 6 of them?

I had some previous knowledge of "snow morphology diagram", where temperature and humidity is decisive for the shape of snowflakes. Thus I was surprised that the current paper does not describe the humidity in details or does not attempt systematic study of the influence of temperature and humidity. Is reasonable to suppose the dependence? Is AST necessarily needed for the observations or would the vapor deposition normal to a-axis be sufficient?

It would be nice to shortly connect current observations to those of "classical" snowflakes formations. Is there AST mechanism needed there?

I would appreciate if some discussed term are more explained and/or shown in the pictures (droxtal, adjoining facets, basal and prismatic facets in Figure 2).

I think the abstract should be modified according the final content. Currently, I find some disagreement between it and the content of the manuscript. Also the name of prof. Yamashita in the abstract does not seem appropriate to me.

---

## Referee Comment (RC3) · Anonymous Referee #3 · 7 May 2019

This paper reports an experimental study of ice growth from vapor in air, with a focus on the formation of air pockets and secondary habits. It ultimately looks to explain a wide variety of experimental observations on lateral-type growth and looks to relate the observed behaviour to a surface flux of water molecules, which the authors call "adjoining surface transport" (AST). As a non-expert this specific field (experimental studies of ice growth from vapor), I found this paper rather difficult to read, and I felt I learned very little in reading it. I found it to be not well written, and a bit of a jumble of data and ideas with no clear narrative. Hence, I think it could be considerably improved by shortening (i.e. less would be more here) and organizing the material better. Also, the paper often seems to read more like a review, where it was often unclear where the authors' work started and ended. Perhaps after considerable revision with work might

be appropriate for publication.

I was also unable to make sense of their physical model. It has now be very well established that the surface of ice features a layer water molecules that exhibit mobility (this is sometimes referred to as the quasi-liquid-layer – QLL; see Rosenberg, Phys. Today 2005, 58, 50; Li and Somorjai, J. Phys. Chem. C 2007, 111, 9631; Björneholm et al., Chem. Rev. 2016, 116, 7698 ). While different experiments report different thickness for the QLL, they generally agree on its presence, and that its thickness (and other properties) are strongly temperature dependent. One would expect any surface flux to then depend on the thickness of the QLL and the mobility within the layer (which will also be strongly temperature dependent). Thus, AST should exhibit strong temperature dependence, and one would expect to observed "protruding growth" only when the effect of AST is large compared with the rate of vapor deposition (which will depend on the level of supersaturation). Yet, the authors do a rather poor job of characterizing the conditions, temperatures and supersaturations, in the reported experiments. Moreover, in a carefully designed set of experiments (when one varies one of these, for example), it should be possible to see the effect become manifest. I would find such a set of data much more convincing. As a minimum, the authors should do a much better job of describing the conditions for each experiment, and then comparing and contrast the behaviour on the basis of these conditions.
* * *

---

## Author Comment (AC1) · 27 Jul 2019

**Reply to the reviewers**

We thank the reviewers and editor for their time evaluating our manuscript. Here we reply to their helpful suggestions, pointing out changes made to the revised manuscript. First, the main issues:

The major criticism of reviewers 1 and 3 was the length or organization. We responded by reducing the length of the main body of the manuscript by about half. Our other large change is to the introduction, which reviewer 2 expressed a need for more general background. So, we broke the original Introduction into Introduction and Background, adding general information in Background. Except for Appendix A, which is needed for the test of facet spreading, all theory sections are moved to a separate manuscript. The remaining parts have been significantly reorganized, and much rewritten, to make reading easier. All detailed discussion of the secondary habits was moved to the appendix, and some figures have been revised to help clarify the content. Finally, the title has changed to better reflect the content.

> Shown below are the complete reviewer comments, separated by our replies with a red dashed line and made distinct with larger font. Also, in yellow highlighting are changes in the revised manuscript.

**Anonymous Referee #1**

The authors studied the formation mechanisms of air pockets and other secondary habits in snow crystals. The topics coincide with the scope of Atmos. Chem. Phys. Discuss., and the secondary habits of snow crystals are interesting from the fundamental viewpoint. However, first of all, this manuscript is too lengthy (26 figures are shown in the manuscript of 51 pages in total), and too many subjects are included in a mixedup way. Therefore, I need to say the presentation quality is poor. Second, the authors insist that the formation of corner air pockets (the main subject in this manuscript) cannot be explained by the traditional growth mechanisms based on lateral step motion, and by the morphological instability based on the inhomogeneous distribution of vapor density. Then the authors conclude that the lateral-type (protruding) growth, which is the key mechanism in this study, is a novel concept. However, I cannot agree with such authors' claims (for details, see the comment 2). Hence, I believe that the scientific significance of the present manuscript is also poor. Since the amount of revisions is significantly large and the conceptual revisions are necessary, I do not recommend the publication of this manuscript.
* * *
Our reply:

Thank you for your comments. We address these issues below. In brief, the manuscript is reorganized, with a main body half as long. The issue with comment 2) seems to be based on a misreading as explained below.
* * *
Following 1-3 are major comments.

1. Too lengthy: one paper should have one main claim. Then, I believe that the following topics should be presented in separated papers: # the formation mechanism of the corner air pockets, # quantitative discussion about the kinetics of the lateral-type (protruding) growth, and # secondary habits other than the corner air pockets (these topics can be also moved into supplementary information)
* * *
Our reply:

We followed your suggestions in part by moving most theory subsections to a separate manuscript. The secondary habits discussion is a crucial part of our main claim and remain in this paper, but due to their length we moved them to the appendices and simply summarize the findings in the main body. They remain in the paper because they give further support for the importance of lateral growth and provide numerous tests of the phenomenon, but do not interrupt the flow. More importantly for us, they may satisfy readers curiousity about how some very unobvious crystal forms can arise.

Our main claim is that lateral growth should be included in any complete model of ice growth. We stated this in the introduction and in the conclusions, but neglected to explicitly state it in the abstract. This omission has been corrected. Appearing at the end of the abstract:

Although these suggested mechanisms may presently lack quantitative detail, the overall body of evidence here demonstrates that any complete model of ice growth from the vapor should include lateral-type growth processes.

The title was also changed to better reflect the content.
* * *
2. The formation mechanism of the corner air pockets: the authors mentioned that the normal growth via step motion and the standard hollowing theory based on the morphological instability caused by the inhomogeneous vapor density cannot explain the formation of the corner air pockets. Then, I shall explain the formation of the corner air pockets by the traditional concepts. The key is the morphology of a snow crystal at the beginning of the growth. 1) When a starting crystal is fully faceted, the local vapor density becomes maximum at the corner of the crystal, providing a hollow not at the corner but at the center of the crystal face, as the authors explained. 2) In contrast, when a starting crystal is partly rounded, the layer-by-layer growth of the faceted face (located at the center of the crystal) proceeds. Then a spreading edge appears as shown in Fig. 1b (marked by e) and Fig. 8c. Since the spreading edge shows an angular shape and the corner of the crystal is still rounded, the local vapor density at the tip of the spreading edge becomes higher than that at the rounded corner, providing the overhang as shown in Fig. 1d and Fig. 8d. After once the overhang was produced, the overhang is developed spontaneously (the authors call this process the protruding growth), and the corner air pocket is formed. These processes never violate the traditional concepts of the layer-by-layer growth and the morphological instability.
* * *
Our reply:

Thank you for the suggested mechanism. We agree that the vapor-density gradients should promote protruding growth and that "traditional concepts" are involved. In the first paragraph of §3.6 (which the comment addresses), we are referring to hollows forming on a fully facetted face as being incompatible with the observed corner pockets. To reduce chances of the same misunderstanding, we changed the wording (now section 4.1.2):

Existing views on normal growth via step motion cannot readily explain corner pockets on fully facetted crystals. With normal growth, each pocket must have at one time been a hollow (lacuna or concave feature) before closing-off to enclose the air. And standard hollowing theory (e.g., Kuroda et al., 1977; Frank, 1982; Nelson and Baker, 1996) predicts that hollows form around a local vapor-density minimum, not at a corner where the driving force for normal growth is instead a local maximum. Moreover, the standard theory relies upon step clumping on a facetted surface.

We agree with the reviewer's ideas about protrusion initiation. But they were already in the manuscript—in the 2$^{nd}$ paragraph of §3.22. This section is now moved to section 5.1. The passage, from line 20

A possible answer to (1) is a large vapor-density gradient. Consider again the sketch in Fig. 2b. If the vapor-density contours closely parallel the surface, but "skim over" the inside corner **c**, then the vapor density would sharply decrease from **e** to **c** provided that this distance exceeded the vapor mean-free path. In such a case, the AST flux may build up nearer to **e** and not reach **c**, initiating the protrusion.

Thus, the reviewer simply misread what we had written. Hopefully, it is now clear. Finally, whether or not one refers to lateral-type growth as a "traditional concept" is immaterial (we never say it is not); the fact we emphasize is that it has not been considered in ice-crystal growth models, which is our main point.
* * *
In addition, the authors emphasize the importance of the diffusion of admolecules on the crystal surface (the surface diffusion of admolecules). Then the authors named this process "adjoining surface transport (AST)". I fully agree with the importance of the surface diffusion for the formation of the overhang and the subsequent protruding growth. However, the concept of the surface diffusion of admolecules is very traditional (firstly proposed by Frank and coworkers in the 1960s, and then experimentally proved by the growth of various crystals). Therefore, the authors should clearly show what is the authors' novel concept and what is not.
* * *
Our reply:

One paragraph in the original introduction was devoted to prior work on surface transport over crystal edges (page 8, paragraph starting on line 12). There are 8

references to prior work on the topic. So, we made no new additions to this discussion, as we feel the 8 citations are sufficient. Moreover, the overhanging aspect of protruding growth appears in none of these previous papers. The reviewer appears to be confusing surface diffusion with surface transport over the edge to the lateral-growth front. These are distinct processes that we have tried to further clarify in the new Background section.

Concerning the point that we should clarify what is new and what is not: We have followed the long-standing scientific practice of giving references to all relevant prior work except when long-established (e.g., kinetic theory of gases), with all else being presumably new. If we were to start saying "this is novel" every time we express a new result or idea, the reader would soon tire of the repetition and deem us arrogant. Even though some authors break with this practice and announce their result as "novel", most do not, and for good reason—that is the job of others to declare. Relevant changes to the manuscript are in our next reply.
* * *
3. Throughout the manuscript, the authors should clearly explain what is the authors' new finding and what is not, with respect to phenomena and formulas as well. Followings are minor comments.
* * *
Our reply:

Whatever does not have a citation is thought to be new, as per standard scientific writing practices. However, several additions help clarify our contributions: In the introduction to the role of AST on secondary habits (Appendix B)

Most of these features and habits appear inexplicable with normal-type growth processes only, and only a few of them have even seen attempts at explanation.

And in the introduction to the main results (§4)

The following subsections survey, and partly explain, observations made in CC2, including previously unreported "corner pockets", "planar pockets", and "elongated edge pockets".
* * *
4. The term "droxtal": since many readers (including me) are not familiar with this term, the authors need to explain it properly at the beginning.
* * *
Our reply:

In the revised manuscript, we define the term in the new background section, page 3 as

Atmospheric ice crystals generally begin with the simplest of shapes, a solid ice sphere, also called a droxtal.

The term is no longer used in the abstract, as we now refer to the droxtal center as

small circular centers in dendrites

The term is also defined in the figure with droxtal images. Before, this was Fig. 2, now it is Fig. 1:

Figure 1: Crystals at different stages between large droxtals (just-frozen droplets) and prisms
* * *
5. The section 1.2 gives the impression that the authors do not fully understand the fundamental growth mechanism of crystal growth. The concept of the surface diffusion of admolecules (AST) is widely accepted in the crystal growth of wide variety of materials: not only for the metal whiskers, but also for semiconductor crystals, molecular crystals and ice crystals as well (as studied by Hallett, Mason et al, Kobayashi, and Asakawa et al.) Hence, for me, the application of the surface diffusion to the lateral and protruding growth by Yamashita (2015) does not look a significant revision, since the lateral and protruding growth can be explained easily, as shown in my above-mentioned comment 2.
* * *
Our reply:

We do not see the relation here between the first sentence and the rest of the paragraph. We agree that surface diffusion is widely accepted, and cite these and other authors for work on AST in the Background section. But surface diffusion is not AST: AST is defined as adjoining surface transport to the lateral "growth front" as stated on lines 11-14 of the abstract. AST was defined elsewhere as well. To help reduce this confusion, we briefly list the relevant definitions now at the end of the background section. The previous section 1.2 has been expanded to the new Background section, so this "impression" should no longer be present.

Concerning comment 2, we address that confusion above. However, whether or not the reviewer considers lateral and protruding growth a "significant revision", the fact remains that it had not been considered as an important aspect of ice growth from the vapor. Lateral-type growth does not appear in the ice-growth literature, and AST is generally ignored. For example, at the end of the new section 5.2.1, after briefly summarizing how AST-driven lateral growth can explain seven secondary habits and features, we add

A recent review of ice growth from the vapor suggested that AST may be unnecessary for understanding ice growth forms (Libbrecht, 2005). The above examples suggest otherwise, instead arguing that many oft-observed secondary features may be inexplicable without the AST mechanism.

Getting back to the reviewer's first sentence, do we "fully understand the fundamental growth mechanism of crystal growth"? Does anyone?
* * *
By the way, it is impossible to obtain the reference Yamashita 2015. The page numbers of the references Yamashita 2013 and 2016 should be 165-176 and 393-400, respectively: the page numbers of 23-33 and 15-22 are those in the issues 60 (3) and 63 (5).
* * *
Our reply:

Yamashita 2015 is a conference proceeding paper. Indeed, such papers are often hard to obtain, but not impossible (we found it). The reference serves the purpose of giving credit to the originator of the idea. Given that this is a common practice in scientific papers, we keep the reference. We have a few other cases like this, but the vast majority of our references are easy to access.

Thank you for the corrected page numbers for Yamashita 2013, 2016. The Tenki journal has two page numbering systems on each page, and, being unfamiliar with this journal's system, it seems we picked the wrong one. This has been corrected.
* * *
6. The authors should show the schematic illustration of the new crystal-growth apparatus (CC2) in this study, since the reference Swanson and Nelson 2019 is still in preparation.
* * *
Our reply:

The manuscript is under review and available for viewing, so we did not include a drawing here. We updated the reference so readers can view the apparatus. Please see the link in the new references:

Swanson, B., and Nelson, J.: Low-Temperature Triple-Capillary Cryostat for Ice Crystal Growth Studies. Atmos. Measurement Techniques, https://doi.org/10.5194/amt-2019-137, 2019.
* * *
7. In the section 3.4, from Fig. 4, I cannot understand the difference between the expanding boundary of the basal face and growing macro-steps. The authors also should clearly explain the kinetic models I, II and III in the main

text (of a separated paper), since the quantitative discussion has no meaning without obtaining the complete understanding of the models.
* * *
Our reply:

A macrostep exists as a large step within a facet and it arises from the clustering of smaller steps. Neither apply here. The case here, as we try to clarify in the main text and the appendix, is a facet expanding over a rough, round surface. The rough, round surface is not yet a facet, and thus our case is not a macrostep; moreover, the facet edge did not arise from step clustering. To help clarify, we added the following text where we discuss the plot:

…, but it is a reasonable fit to the initial cross-section profile. ==This profile is that of a flat facet out to a radius $r < a$, and a curved profile between $r$ and $a$ where the crystal had rounded during sublimation. (Refer to Fig. A1 for further details.)==

We now discuss macrosteps in a few places in the new Background section, clarifying their difference with a spreading facet.

Finally, we keep the appendix here to explain the calculations for the plot, but as suggested will have further details of the model in a separate paper.
* * *
By changing the value of h/xs arbitrarily, one can easily fit the experimental data. Hence, here the authors need to explain the causes of the change in the value of h/xs (I believe that the cause of the change is the evolution of macro-steps) and also whether the change is appropriate or not. The authors also need to discuss the values of h and xs.
* * *
Our reply:

Yes, changing h/xs allows us to fit the data. We made no claim to the contrary and write that accurate measurements with a different apparatus are needed. Nevertheless, the qualitative trend in the fitted curve is consistent with the cross-sectional profile of the crystal when the facet began expanding. It has nothing to do with macrosteps because the expanding facet is not a macrostep. The cause of the change in $h$ is shown in Fig. A1, and explained in the caption:

At a later time $t'$, the value of $h$ is larger (light shading) due to the advancement to $r(t')$, making a larger distance between the rough surface and basal surface at $c$.

 Of course, it is likely that some normal growth on the basal facet occurred that contributed to $h$, but this rate of growth in this case is negligible compared to the

rate of lateral growth, also making the contribution to $h$ negligible. (Other cases may differ, so we do not claim this contribution can always be ignored.)

Without an accurate measure of the crystal profile and an established value for $xs$, providing separate numbers for $h$ and $xs$ here is largely pointless; the trend is clearly consistent with the profile, but accurate measurements are needed. The whole point of the comparison is to show that the AST is the only process capable of explaining the data. Nevertheless, we added the following discussion of $h$ and $x_s$:

A reasonable estimate of height $h$ upon reaching the edge is 1–5 μm. With this range, the fit in Fig. 6 (inset) predicts $h/x_s$ = 0.3, giving $x_s$ = 3–17 μm at this temperature, which is comparable to the value of about 2 μm found by Mason et al. (1963).

But to emphasize the main point here, the last paragraph was modified slightly as

Nevertheless, the observed behavior clearly shows that mechanisms I and III cannot explain the observed lateral growth. Only growth driven by a flux of surface mobile molecules, the AST mechanism, from the facet to the lateral-growth front is capable of fitting the observations.

Also, to help reduce confusion on this issue, we removed the two intermediate curves with high supersaturations from the plot. Now there are just three curves for the three models, all at the same supersaturation.
* * *
8. In the section 3.9, the authors should explain the impossibility and instability much more in detail.
* * *
Our reply:

Detailed examination of impossibility and instability theories are covered well in the literature (though the term 'impossibility' is not used elsewhere). However, we agree that a little more explanation and rewording will help. In this section, after the 2$^{nd}$ sentence, we have the following rewrite that expresses the key differences between these two types of behavior:

In the standard treatment, however, the hollow occurs when the gradient in supersaturation needed for uniform growth can no longer be compensated for by the step density (e.g., Kuroda et al., 1977; Frank, 1982). In other words, normal growth of the entire facet becomes impossible, which is different from being unstable. In this "impossibility" case, one expects the hollow initiation and shape to be nearly identical on identical faces in a nearly uniform environment as well as being highly reproducible when other crystals grow under the same conditions. If merely an unstable phenomenon, then a sufficiently uniform, constant condition may be expected to circumvent the hollowing. Conversely, if hollows do form, then their initiation and shape should differ between identical faces due to minute differences in

conditions. We suggest here that inclusion of lateral-type growth processes predicts qualities of unstable growth at low supersaturations, leading to hollow close-off and terracing features.
* * *
9. The sections 3.10-3.22: If these sections have scientific significance, the authors should explain them in separated papers. If their scientific significance is not so large, the authors should move them into the supporting information.
* * *
Our reply:

These sections provide explanations for commonly observed ice-crystal forms and secondary features, making them significant to anyone interested in the causes of crystal shapes. Please compare to any paper (*of thousands*) that have been written about dendrite branching or some of the cited ones dealing with trigonal growth.

 As these crystal forms and features had not been satisfactorily explained by other mechanisms, the explanations here involving lateral-type growth provide support for the importance of lateral-type growth, the main point of our paper.

 We understand that the volume of information makes reading in one push difficult, so we moved these subsections to the appendix, instead summarizing them in the new subsection 5.2.1. We also added text and two figures showing images of two such crystal forms to more clearly motivate the need for explanation.

Perhaps what we failed to express here is that the extreme complexity of ice-crystal growth in air warrants a variety of approaches. At this stage, we need to at least know which processes to include in a crystal growth model. No model has yet included lateral-type growth. By including evidence that such a growth process can explain numerous micron-scale features in vapor grown ice (the secondary features), this paper shows just how prevalent the lateral-type growth is.

If getting the relevant growth processes right is not significant, then we do not know what is.
* * *
Anonymous Referee #2

The paper "Air pockets and secondary habits in ice from lateral-type growth" presents extensive collection of micrographs of growing ice. I am very pleased by its esthetical beauty. I think it should be published for its experimental value, regardless the theoretical explanations. The explanation of air pockets formation is elegant. I cannot judge on its correctness. Nevertheless, I have some suggestions stemming mostly from the fact that I am not an expert in the field of ice grow from the vapor phase and thus I would welcome some introduction and generalizations. That may be a case for most of the readers, though. I suggest the paper be published after considerations the comments.
* * *
Our reply:

Thank you very much for recognizing the extent and significance of the work in this paper and making these suggestions. We have added some more general background in the beginning, breaking the introduction into a shorter introduction section that gives our motivation, similar to before, and a new background section that covers the more general topic (or "generalizations").
* * *
I would think that such extensive work deserves broader introductions and connections to what "general ice knowledge" may cover. I started form Hobbs: Physics of ice Ch. 8: I found the description of the growth in the direction of c and a-axis easier to understand then to consider the basal face and prismatic face in presented manuscript. Can both description be shown in the pictures?
* * *
Our reply:

Yes. We revised the sketches in Fig. 1 (now Fig. 2) to include different views and the principle axes as expressed in Hobbs. As mentioned above, we have broadened the introductory section, adding a section 3 "Background". This section explains the normal growth directions.
* * *
In Hobbs (ch. 8.3) the linear growth (here named normal) is defined as normal to crystallographic face. Is the here discussed lateral growth perpendicular to both c-axis and a-axis? Would not that be more exact definition than that given on the line 5 of page 2?
* * *
Our reply:

Lateral growth of a basal face is growth perpendicular to the c-axis, but the same cannot be said for the prism face. However, the maximum dimensions normal to the c-axis tends to be in the a-axes directions as shown in the revised Fig. 1, now Fig. 2.

We also improved the sketches in this new Fig. 2 to help explain what we mean by lateral growth. Also, in the Introduction, we help to clarify lateral growth:

The rate is often called the linear growth rate (e.g., Lamb and Scott, 1972), but to help distinguish this face-normal growth from face-lateral (or areal) growth, we refer to it as the normal growth rate.

Also, we try to clarify the types of lateral growth with a bulleted list of definitions at the end of the new Background section.
* * *
I think, that schema in Figure 1a suggests that the droxtal has 8 prismatic phases –should not there be only 6 of them?
* * *
Our reply:

There are just two prism faces in back. The figure has been revised to help address your previous concern, and now makes the six prisms clear by showing the top view at right in the new Fig. 2.
* * *
I had some previous knowledge of "snow morphology diagram", where temperature and humidity is decisive for the shape of snowflakes. Thus I was surprised that the current paper does not describe the humidity in details or does not attempt systematic study of the influence of temperature and humidity. Is reasonable to suppose the dependence? Is AST necessarily needed for the observations or would the vapor deposition normal to a-axis be sufficient?
* * *
Our reply:

Lateral-type growth should depend on temperature and humidity. We plan to investigate this experimentally, and also hope that this paper spurs others to investigate as well. We now mention some expected dependences. For example, in the Background section, pg. 3, lines 3-5, we mention the temperature dependence of $x_s$:

Experiments reported in the 1960s indicated that $x_s$ on the basal face varied dramatically with temperature, changing by a factor of 5–7 between about −7 and −12 °C (Mason et al., 1963; Kobayashi, 1967). Although the exact values of $x_s$ may be disputed, both studies independently found the values to be largest in the tabular regime, smallest in the columnar.

In other parts of the manuscript, we refer to this temperature dependence for the basal as being potentially important for various features, such as the two-level structure of planar crystals, capped columns, and trigonals. But the values for the basal have not been verified by other experiments and we do not have measurements for the prism. For protruding growth, another length scale may be important as well, the migration distance on rough faces, which may depend on temperature and supersaturation, but we have no theory or experiments to guide us. We mention this in the new Background section. Thus, we do not attempt anything like the snow crystal habit diagram for lateral-type growth features. However, we now mention the snow-crystal habit diagram in the introduction:

The primary and secondary habits depend on temperature and humidity as often portrayed on the habit diagram. This diagram has generally remained the same since Ukichiro Nakaya first proposed it (Nakaya, 1954), though some extensions and modifications have come from subsequent studies (e.g., Hallett and Mason, 1958; Takahashi et al., 1991; Bailey and Hallett, 2004; Takahashi, 2014).
* * *
 It would be nice to shortly connect current observations to those of "classical" snowflakes formations. Is there AST mechanism needed there? I would appreciate if some discussed term are more explained and/or shown in the pictures (droxtal, adjoining facets, basal and prismatic facets in Figure 2).
* * *
Our reply:

We make several connections to the classical stellar crystal, but do not suggest that the main features such as the branches and sidebranches are due to lateral growth. In the main body, we make the connection to corner pockets in classical snow crystals in Fig. 8 and section 4.5, as well as discussing the two-level structure in section 5.2. Several other aspects of the classic snow crystal are addressed in Appendix B. We now list the common lateral-growth terms at the end of the Background section.
* * *
I think the abstract should be modified according the final content. Currently, I find some disagreement between it and the content of the manuscript. Also the name of prof. Yamashita in the abstract does not seem appropriate to me.
* * *
Our reply:

We are not sure what "some disagreement" refers to here, but we have assumed the reviewer means that some findings are not explicitly mentioned in the abstract. In response, we added our model fit to the abstract and mention other results from the experiments:

Further experiments revealed other types of pockets that are difficult to explain without invoking AST and protruding growth. We develop a simple model for lateral growth on a tabular crystal in air, finding that AST is also required to explain observations of facet spreading.

and added some words to help clarify our applications to observed secondary features:

Applying the AST concept to observed ice and snow crystals, we argue that AST promotes facet spreading, causes protruding growth, and increases layer nucleation rates. In particular, depending on the crystal shape and conditions, combinations of these lateral-type processes with normal growth can help explain presently inexplicable features and secondary habits such as air pockets, small circular centers in dendrites, hollow terracing and banding, multiple-capped columns, scrolls, trigonals, and sheath clusters. For dendrites and sheaths, AST may increase their maximum dimensions and round their tips. Although these applications presently lack quantitative detail, the overall body of evidence here demonstrates that any complete model of ice growth from the vapor must include these lateral-type growth processes.

About mentioning Prof. Yamashita in the abstract, we realize that the practice is not common. But it is done (see e.g., some abstracts from the physicist J. A. Wheeler, and the one by Frenkel that we cite), and in this case we prefer to have his name. He has promoted the idea of AST and protruding growth for years, often communicating with one of us, but has had difficulty writing his results up for an English journal. We reference all of his relevant work in the main text, but some readers only read the abstract and we feel that he deserves recognition for the concepts on the front page lest readers mistakenly think we originated the concepts of AST and protruding growth. But to help address your concern, we shortened the mention of his name in the abstract.
* * ** * *
Anonymous Referee #3

This paper reports an experimental study of ice growth from vapor in air, with a focus on the formation of air pockets and secondary habits. It ultimately looks to explain a wide variety of experimental observations on lateral-type growth and looks to relate the observed behaviour to a surface flux of water molecules, which the authors call "adjoining surface transport" (AST). As a non-expert this specific field (experimental studies of ice growth from vapor), I found this paper rather difficult to read, and I felt I learned very little in reading it. I found it to be not well written, and a bit of a jumble of data and ideas with no clear narrative. Hence, I think it could be considerably improved by shortening (i.e. less would be more here) and organizing the material better.
* * *
Our reply:

We are sorry to read that you learned very little from it. We agree that the original was hard to read completely through due to the length of the main body. To improve the narrative, we have reduced by half the main body, largely reorganized the paper, rewriting many parts, adding more motivation for the work, removed most theory sections, and added to the introduction a background section to put the work in context of standard crystal growth theory.

Applications to secondary features, which occupied much of the main body of the original, no longer appear before the summary, instead appearing in Appendix B, beginning with a list of the content. The main point is now emphasized more, which should also improve the narrative.
* * *
Also, the paper often seems to read more like a review, where it was often unclear where the authors' work started and ended. Perhaps after considerable revision with work might be appropriate for publication.
* * *
Our reply:

We understand how one might read the paper in this way because we have examined many observed crystal types. And in a sense, it is a review of our own work going back six years. But whether one considers it a review or not, it is all tied together by the clear evidence that lateral-type growth is needed to understand many ice-crystal forms. That this point should be emphasized is given by the fact

that a previously published review considered it unnecessary to include AST in growth models. We now explicitly state this at the end of section 5.2.1:

A recent review of ice growth from the vapor suggested that AST may be unnecessary for understanding ice growth forms (Libbrecht, 2005). The above examples suggest otherwise, instead arguing that many oft-observed secondary features may be inexplicable without the AST mechanism. Additional cases, including aspects of primary habit, are briefly examined in the appendix as well. The arguments are mostly qualitative; nevertheless, they may help stimulate new measurements of $x_s$, further observations, and more detailed modeling of these interesting crystal forms.

However that review examines cases of normal growth for the primary habit and "morphological instability", but does not consider some of the secondary features like we do.

Concerning where our work started and others' ended, we reply to this point in a previous reply: we have followed standard procedure and given references for any work or results that are not ours.

Finally, we have made considerable revisions, as suggested.
* * *
I was also unable to make sense of their physical model.
* * *
Our reply:

For the examination of facet spreading in Fig. 5 (now Fig. 6), the model we used is the BCF model as justified in the Background section. Starting in the first paragraph:

The most widely used model for the growth of crystal faces from the vapor is the BCF model (Burton et al., 1951; Woodruff, 2015). This model supposes that a given molecule in the vapor above a faceted surface strikes the crystal surface and become temporarily trapped in a mobile state until either desorbing back to the vapor or migrating along the surface and reaching a more strongly bound state at a step edge….

Then at the end of the third paragraph

…Hence, at least as a first approximation, it is still useful to compare observed behavior of ice to the BCF model and make use of measured $x_s$ values.

The basic physical model involves AST, which is not new. From the old Introduction, now near the end of the new Background:

==The types of lateral growth here are driven by AST.== Evidence for AST on ice is indirect, partly coming from early studies of spreading ice layers on covellite (Hallett, 1961; Mason et al., 1963, Kobayashi, 1967). In these studies, the rates of approaching micron-scale layers, also known as macrosteps (arising from clustering of smaller steps or contact between crystals of differing height) changed in a way consistent with a flux of molecules over the top edge of the layer. The concept has long been applied to the growth rates of metal whiskers (e.g., Sears, 1955; Avramov, 2007), but rarely applied to ice.

Most theory sub-sections have been removed (put in a separate manuscript). The details of the model calculation for Fig. 6 are in the first appendix. In the rest of the manuscript, we are only arguing qualitatively for AST effects. The reason is two-fold: 1) calculations of an AST for most secondary features would require much more space in a separate paper (e.g., for the needle crystals) and 2) the key parameter $x_s$ is poorly constrained, particularly for the prism face. Our hope is that this study stimulates work on the topic, allowing such quantitative treatment in the future.
* * *
It has now be very well established that the surface of ice features a layer water molecules that exhibit mobility (this is sometimes referred to as the quasi-liquid-layer – QLL; see Rosenberg, Phys. Today 2005, 58, 50; Li and Somorjai, J. Phys. Chem. C 2007, 111, 9631; Björneholm et al., Chem. Rev. 2016, 116, 7698 ). While different experiments report different thickness for the QLL, they generally agree on its presence, and that its thickness (and other properties) are strongly temperature dependent. One would expect any surface flux to then depend on the thickness of the QLL and the mobility within the layer (which will also be strongly temperature dependent). Thus, AST should exhibit strong temperature dependence, and one would expect to observed "protruding growth" only when the effect of AST is large compared with the rate of vapor deposition (which will depend on the level of supersaturation).
* * *
Our reply:

Thank you for pointing this out. We agree that the surface of ice has significant mobility and that many researchers have found evidence for a QLL. In the background section, we now briefly discuss the apparently disordered nature of the ice surface, its role in growth, and our reasoning for using BCF with the parameter $x_s$:

==The BCF model of surface diffusion assumes that the mobile surface molecules are sparse and non-interacting. For ice, this assumption is widely believed to be violated over much of the atmospheric temperature range where the surface is thought to contain significant disorder (e.g., Rosenberg, 2005). Yet the BCF model is nevertheless often used to interpret experimental results (e.g., Sei and Gonda, 1989; Asakawa et al., 2014). A key parameter in the model is the mean migration distance $x_s$ of a mobile molecule on the surface before desorbing, a distance that should differ between the basal (**b**) and prism==

(*p*) faces as well as depend on temperature. With interactions between these surface-mobile molecules (e.g., Myers-Beaghton and Vvedensky, 1990), $x_s$ may also depend on supersaturation. In addition, the migration of surface vacancies may also affect $x_s$ (Frank, 1993). Experiments reported in the 1960s indicated that $x_s$ on the basal face varied dramatically with temperature, changing by a factor of 5–7 between about −7 and −12 °C (Mason et al., 1963; Kobayashi, 1967). Although the exact values of $x_s$ may be disputed, both studies independently showed that the values are largest in the tabular regime, smallest in the columnar. Corresponding values for the prism have not been determined. Later, Nelson and Knight (1998) found a similarly sharp behavior in basal-face critical supersaturation between these temperatures. A possible link between these two parameters is clustering of the mobile species responsible for growth: when the temperature is such that clustering is strong, the critical supersaturation is low and surface-mobile molecules would become temporarily trapped in sub-critical nuclei, giving them very low mobility. Thus, the critical supersaturation would be low when $x_s$ is low and vice-versa as found by experiments. The values of the measured critical supersaturations led Nelson and Knight to conclude that the surface was indeed disordered but " ...the [common] view of the ice surface as a liquid layer is not a useful idealization for crystal growth processes." Hence, at least as a first approximation, it is still useful to compare observed behavior of ice to the BCF model and make use of measured $x_s$ values.

That passage also includes the review article by Rosenberg that you cited. Also, in the discussion of hollows and center-pocket formation, we mention a newer study of step dynamics in a disordered region (now in Appendix B.1):

(Neshyba et al. (2016) proposed a more detailed model of step dynamics for ice with a thick surface-disordered region, but it is not yet clear how a hollow would develop in that model.)

Finally, we consider that with or without a thick disordered layer, the AST process may continue. Even as isolated mobile molecules, the simple calculation using BCF gives a significant contribution to growth, the amount depending on the ratio of the surface-migration distance $x_s$ to the edge thickness $t$. In the new section 5.1, at the start, we added

The microscale mechanism of facet spreading involves AST, which is the migration of molecules, first over the edge of the facet, and second with their finding a high-density of growth sites on the other side. The first may occur via isolated molecules or as a more cooperative phenomena in a thicker disordered region, but either case may be consistent with the observations here.

Concerning the temperature dependence of AST, this will largely depend on $x_s$, which is presently poorly constrained, and the thickness of the lateral-edge front. Other factors will be addressed in a follow-up paper.

In the section on the two-level formation on droxtals, now appendix B.3, we describe the temperatures that these are found and make the connection to the high $x_s$ values measured by Mason et al. (1963) at these temperatures. In this

section and elsewhere, we refer to their finding a temperature-dependence of $x_s$ on the basal face.
* * *
Yet, the authors do a rather poor job of characterizing the conditions, temperatures and supersaturations, in the reported experiments. Moreover, in a carefully designed set of experiments (when one varies one of these, for example), it should be possible to see the effect become manifest. I would find such a set of data much more convincing. As a minimum, the authors should do a much better job of describing the conditions for each experiment, and then comparing and contrast the behaviour on the basis of these conditions.
* * *
Our reply:

We have included the conditions whenever possible. When they are only partly known, the examples are nevertheless useful to help support the roles of lateral-type growth. For example, in Fig. 1 (old Fig. 2), the main points of showing them are the spreading facets and the pockets. In this case, we give references to other cases with more precisely known conditions. For the case in Fig. 3, we give the temperature, supersaturations, and crystal sizes. For the case in the old Fig. 4 (now Fig. 5), we had the conditions in Appendix A, but now added them in the main text. Conditions were given for the crystal in Fig. 6 (now Fig. 7). We added the conditions in the caption of Fig. 12, but we empasize at the beginning that the behavior being reported occur over a wide range of conditions. This range will be explored quantitatively in a future study. In general, the effects of AST described here will be larger when the mean migration distance is larger than the edge-front or protrusion thickness. These quantities are not known very well, so testing the predictions will require their measurements. Our task here has been to establish that AST occurs and is worthy of further study.

---

## Author Response (AR3)

**Reply to the reviewers, round 2**

We greatly thank the reviewers and editor for their time evaluating our manuscript in this second round. In this round, we followed all of the suggestions. To help show exactly how we followed the suggestions, see the details below. We list the suggestions, and follow with our changes. These changes are marked with a "→" and the new text is highlighted yellow (deletions are not shown).

   In addition, we carefully reread the entire manuscript, finding small changes such as an added word or two here and there, or a change in wording to improve consistentency, to help clarify the text.

**Anonymous Referee #1**

1) The length: Although the original manuscript was too long for the publication as an original article, now the revised manuscript becomes significantly shorter. Some people may think still lengthy, the present length would be within the acceptable range. I understand the authors' thought that they want to include other secondary habits in this paper. Then I leave the decision about the length to the editor.
→Thank you. We understand the difficulty of reading long papers. On the other hand, if organized well, the long paper can be easier than trying to keep track of two or three separate papers that overlap. Our aim is to reduce the overall length (from having several medium-length papers) and have all the connected material together. So, we tried to make the organization much clearer. It seems we have at least partly succeeded.

2) The authors' main point and the formation mechanism of the air pockets: Now I understand the authors' main point: the lateral-type growth has not been considered in ice-crystal growth models. Since the main point was not clearly explained in the original manuscript (it was fully impossible for me to find the point), I misunderstood that the authors' main point is the novelty of the lateral-type growth: the authors claim that the lateral-type growth cannot be explained by the traditional concept. Then I could not agree with this in my last review.
The addition of the section "2 Background" is also helpful in avoiding the wrong impression that the authors do not fully understand the fundamental growth mechanism of crystal growth.
My misreading of the original manuscript was also due to the usage of the words "lateral growth" and "lateral-type growth" without the careful definitions. I am a crystal growth physicist working outside the field of atmospheric science. In my field, "lateral growth" explicitly means the "lateral growth of elementary steps (and also that of bunched steps)" and does not mean "the areal growth on fully facetted faces". Without the definitions newly added at the end of the section 2, I could not understand the authors' meanings. The words "lateral growth" and "lateral-type growth" still remain throughout the manuscript. I believe that many crystal growth physicists will have the same

misreading. Then I strongly recommend that the authors should replace these words with a different word newly prepared, such as "lateral-areal growth" or "lateral-protrusion growth".

→We changed the term in the title, abstract, introduction, and a few other spots to "lateral facet growth". This should be sufficient to prevent confusion with lateral growth of steps (which is similar, particulary with macrosteps). In some places, we have simply "lateral growth", but the reader has already been told it is about the spread of facets so the distinction should be clear throughout the text.

3) The "disorder of the surface" appeared at the end of the third paragraph of the section 2 Background: Here, the usage of the term "quasi-liquid layers" makes many readers easier to catch the point.

The following is not my review comment but my personal comment unconnected with this review: a recent review article (Y. Nagata, et al., "The surface of ice under equilibrium and non-equilibrium conditions", Accounts Chem. Res., 52, (2019) 1006-1015) will be also helpful to clarify the situation of ice surfaces; this article demonstrates that top-most bi-layers of ice basal faces are disordered above -90°C (and second-top-most bi-layers are also disordered above -16°C), however such disordered layers grow fully in the "layer-by-layer manner" like elementary steps.

→This is a good study to reference, and we agree that it is better to use the common term QLL. As a result, we rewrote the section as

"The BCF model of surface diffusion assumes that the mobile surface molecules are sparse and non-interacting. For ice, this assumption is suspect over much of the atmospheric temperature range. Specifically, the ice–vapor interface is widely thought to contain significant disorder, a phenomenon also called the quasi-liquid layer QLL (e.g., Rosenberg, 2005). A recent study finds that this "layer" is limited to two ice bilayers (~0.74 nm) below −2 °C, and less than half that below −16 °C (Nagata et al., 2019). Despite this layer's thinness, such a surface still deviates greatly from the BCF assumption. Nevertheless, the BCF model is often …"

**Anonymous Referee #2**

Summary and Main Comments:

This paper reports on measurements of the growth of single faceted ice crystals grown on a capillary in a new cryogenic chamber. The chamber allows for crystals to undergo growth and sublimation cycles with imaging that has enough detail so that approximate rates of growth can be derived from the data. The data show clear indications of the formation of entrapped air pockets in crystal corners and edges among other interesting features. These pockets appear after periods of sublimation. While the appearance of these pockets has been noted in prior measurements and observations, the authors of this article provide an explanation for the existence with of these pockets with a theory of lateral facet growth by protrusions driven by the flux of molecules across an adjoining facet. The authors

show that a physically plausible model of protrusion growth driven by this adjoining flux of mobile surface molecules can explain the rate of lateral facet spreading derived from the data. The authors then make use of theories of lateral growth, and protrusion growth in particular, to provide qualitative explanations (or perhaps hypotheses) for the development of various secondary habits of ice crystals. This article is the first attempt (that I am aware of) to quantify and explain the lateral growth of crystal facets. At present, there are no quantitative or qualitative models of lateral growth used in the atmospheric sciences. Ice crystal growth models currently used in the atmospheric sciences fit into only two categories: (1) Models that are designed for growth that is normal to the facet. In other words, steps formed by nucleation or by outcrops of dislocations propagate parallel to the facet, causing the facet to grow outward, normally (what the authors refer to as normal growth). These models are used to study the development of single crystalline habits, and they are sometimes used to understand laboratory growth data. (2) Models that treat the growth of ice as if the surface doesn't matter. These models use the capacitance framework, which is not strictly appropriate for faceted growth. These models have been used to interpret laboratory growth data and are used ubiquitously in atmospheric cloud models. Hence, and as the authors themselves point out, there has been no focus in the atmospheric science community on facet growth that is lateral, however it is clear that this sort of growth must be important. For instance, facets must develop over time anytime an ice crystal nucleates either from a frozen droplet or from a solid aerosol nucleus. This process must include the lateral spreading of facets, a process for which we have very little data and no quantitative models. One can easily imagine where lateral spreading may be important: There are numerous laboratory measurements of the growth rates of ice crystals that begin from a nucleated ice particle. These crystals all probably undergo a period of facet development where lateral spreading, and perhaps protrusions, are likely important. However, lateral growth is never considered when laboratory growth data are interpreted (because, of course, no model of this sort of growth exists at the present time). One can also imagine that lateral growth is important in the modeling of atmospheric clouds: The overall mass growth rates of crystals that are growing laterally are likely quite different than normal growth of crystal facets, and probably very different than the capacitance model growth rates. Substantial differences in the crystal growth rates would naturally lead to impacts on model simulated cloud properties including numbers of nucleated ice crystals, and the mass and thermal energy budget of a cloud layer (through latent heating and crystal sedimentation out of the cloud layer). At the present time, the community lacks measurements, ideas, and theories (even simple ones) to advance the way we think about growing ice crystals and the impacts they may have on clouds. This paper is a nice first step in examining lateral growth and its potential impacts on a variety of complex crystal forms, and I think the paper will stimulate the thinking of those interested in advancing our methods of modeling ice in the laboratory and the atmosphere. I am therefore eager to see this paper published in some form, and I would suggest minor revisions: The paper is quite clearly written and is well argued within the constraints of the available data. While the paper is shorter and clearer than the original discussion paper (I perused this paper as well during my review), and the science appears quite sound, I do have a number of suggestions and questions (see below)

related to the presentation of the material, and this is the reason for my recommendation of minor revisions. The above summary of the paper is, of course, my current understanding of the material and I hope that I have not misunderstood the authors' ideas and intent.

→Thank you for this nice summary. It provided useful ideas for revisions made to the introduction sections as mentioned in (1) below.

Enumerated Comments:

(1) General comment on the introduction/background: While the introduction is quite clear and well written, I do think it may be hard for those who are not quite familiar with the theories of ice crystal growth to place the results here into an atmospheric context. Ice crystal growth theorists and laboratory scientists who measure the growth of crystals will probably be able to grasp the concepts presented in the present paper, but those outside of these areas may have more difficulty even though the material is of general interest (in my view). Perhaps adding a few sentences that place these results into a broader atmospheric context would be worthwhile. I do not think that adding this is critical to the paper, it is a suggestion that may help interested readers see the possible implications of these results.

→We added a more general introduction to snow and ice as a new first paragraph, with emphasis on factors that may (eventually) involve applications of the present research. Thus, the entire first paragraph is new, with the old first paragraph shifted to being the 2$^{nd}$ paragraph etc.

(2) Line 18, pg2: I may have missed it, but I do not think that I saw the definition of the initialism "BCF" given earlier in the text.

→We reworded it to show the source of the initials: "The most widely used model for the growth of crystal faces from the vapor is the "BCF" model (Burton, Cabrera, and Frank, 1951; see Woodruff, 2015, for updates and history)."

(3) Line 33, pg2: "all thick surface regions leading growth" is a little awkward. I would suggest rewording.

→We reworded and clarified as ", but for vapor growth, the leading fronts (i.e., outermost faces that define the maximum diameter and have the fastest normal growth) are usually facetted. Individual steps, and steps clumped into macrosteps, instead tend to have a rough edge as indicated by their curved perimeter (generally circular or spiral). Also, when the leading front is very thin, it may appear rounded."

(4) Line 34, pg2: "tend to have a rough edge" perhaps add "indicated by rounding", as you later point out. I think here you are trying to point out that vapor grown crystals are faceted, meaning that the surfaces are not "rough" but that individual steps can be rough as indicated by their rounded in appearance.

→We revised as "Individual steps, and steps clumped into macrosteps, instead tend to have a rough edge as indicated by their curved perimeter (generally circular or spiral)." We specified the perimeter to

help clarify that the rounding is as viewed normal to the face.

(5) Line 17, pg 3: "Instead, atmospheric ice models usually…" This statement is definitely true for models like Wood et al. (2001), but ice models used for cloud simulations usually do not include any information about the crystal surface. The usual assumption is that the surface is at equilibrium and that no steps exist at all, since they use the capacitance model.

→Good point. We revised the passage, adding more explanation:

"Instead, atmospheric crystal-growth models usually assume a locally uniform vapor density near the step source and allow the vapor density to monotonically decrease or increase across the surface. (Most cloud models use the more extreme simplifications of the "capacitance model", which includes no detail of surface structure and assumes local equilibrium over the entire surface. But the recent work by Harrington et al., 2019 is a welcome exception.) As the crystal shape…"

(6) Line 18, pg 3: Should "density" be inserted in "vapor near the step source"?

→Yes, now added.

(7) Caption of Figure 1: The word "sizes" always seems ambiguous to me. Perhaps "diameters"?

→Yes, and the change has been made.

(8) Figures in general: A number of the figures show crystals grown in various chambers. I think it might be good to provide some more information on the environmental conditions: Temperature is sometimes given, but what about pressure and supersaturation?

→We now provide conditions as far as they are known for all captions.

(9) Line 9, pg4: I would insert "diameter" in the parenthetical "(~20 micrometers)" since Gonda's measurements were of the width of the frozen droplet.

→Yes, we added "diameter".

(10) Line 11, pg4: "…show these edges as rough.." Is this indicated by the fact that they are rounded? It might be worth it to point that out.

→Changed as " Figure 1a–d shows these edge fronts as rounded, indicating rough edge and hence an efficient collector of molecules.

(11) Line 13, pg4: I have real difficulties seeing the pyramidal facets on Fig 1a,b since the image is a little fuzzy. Perhaps an arrow could be used to indicate the location?

→We added arrows and modified the caption accordingly.

(12) Line 19, pg4: "nucleation of new growth layers". Is it possible that the facets are growing by dislocations instead of layer nucleation? Gonda and Yamazaki's (1984) paper shows the growth velocities of the a and c axes of their crystals, and the growth rates are quite close to each other (their Fig. 3). Given that the supersaturation in that case was between 1 and 2% it would seem that the growth would have to have been dominated by dislocations. Otherwise the axis growth rates

would have been different I would think (since the critical supersaturations for the basal and prism faces are around 0.5% and 2% respectively at Gonda's growth temperature of -15C).

→Yes, though still unknown, it seems likely that dislocations would be present on some faces. Our argument remains essentially the same, but to clarify we changed to " Thus, the propagation rate (and nucleation of new layers in the absence of a permanent step source) of surface steps will be reduced until the facet radius **m–e** exceeds the surface migration distance $x_s$."

(13) Figure 2: I very much like this figure. However, later on in the paper you discuss the vapor gradient near the protrusion. If it wouldn't make the figure too messy, it might be good to add in isolines of vapor density. I was able to follow your description of the structure of the contours, but an image would certainly help. Especially for those who are not familiar with the way vapor gradients may change near the surface of a crystal.

→Nice idea. We added an additional sketch to the figure to show qualitative features of the contours. To make better use of the revised figure, we also revised the relevant discussion in section 5.1 where we already referred to Fig. 2.

(14) Line 3-4, pg 6: I personally find it hard to see much in some of the images that Gonda and Yamazaki present in their papers. Would it even be possible to discern small air pockets given the image quality?

→We added "Given their small droxtal sizes and darkness of their images, one cannot rule out the existence of very small pockets, but their results show no indication of pockets of the scale seen in Fig. 1." We also try to clarify a closely related point in section 4.1.1 where we modified the text at the end as "Thus, although the phenomenon can appear on a range of crystal shapes, the corner radius may need to exceed a certain value for the corner pockets to either exist or become resolvable with standard microscopy. "

(15) Line 34, pg 6: Perhaps you could add "in a later study, both of us (JN and BS) began…" This would provide an implicit reference to BS as an author, since the initials BS are not otherwise defined. And this would be consistent with line 31 of the same page.

→Nice idea. We made the change.

(16) Line 2, pg 7: The initialism CC2 should be defined. Are the crystals grown at ambient atmospheric pressure in the new chamber? Can you provide a very brief explanation of how the supersaturation is estimated, since that is required for the growth calculation shown later.

→We revised as "For this work, we used a new crystal-growth apparatus, hereafter CC2, that improves upon the first "capillary–chamber" method in Nelson and Knight (1996)." The added part may help the reader remember the reason for the name CC2.

A few lines down, we added "…valve stopper. Briefly, the vapor source (ice, pure melt, or solution) has a surface area vastly greater than that of the observed crystal on a capillary. Thus, except very near the observed crystal (when air is present), the vapor density throughout the system is the equilibrium value of the vapor source from which we calculate far-field supersaturation. With this system,…"

In the next paragraph, we added "…thus complementing our CC2 results. In both the cloud chamber and CC2 experiments, the crystals grew in an atmosphere of air. Other…"

(17) Line 11, pg 8: "after the sublimation". It seems like the word "period" should be inserted here after sublimation.

→Yes. We added it.

(18) Line 12, pg 10: "unusually thin" Are they usually thicker? If so, how much?

→Unfortunately, we do not have clear sets of images of the more common "thick" cases here (we are preparing a more detailed study of hollows now), so we cannot give a quantitative or specific example. But this is a good point, so we added the sentence "That is, hollows often start by widening with a nearly circular rim shape (e.g., in hollow columns), whereas the hollows that preceded these planar pockets must have instead had a rim shape similar to a thick line segment before closing into pockets."

(19) Line 23, pg 10: "unusual for a crystal grown at such low supersaturation" Can a reference be provided here?

→We added "More typical cases for low supersaturation are shown in Figs. 7,9 and the literature (e.g., Nelson and Knight, 1996; Gonda, Sei, and Gomi, 1984)."

(20) Lines 5-8, pg 11: Using rings to determine the facet spreading is a nice idea. How is the location of each ring determined? It also might be good to provide an error estimate, which can then be used to provide an error estimate for the rate of spreading shown in Fig. 6.

→The estimated ring position was made by eye. We made a few small changes to this paragraph, also explaining as "The positions of these rings, simply estimated by eye, are marked in (f), with the time interval (units of 5 min) between marked positions in the upper right." The error estimates are addressed in the next reply.

(21) Concerning Fig 6: On a first glance, I thought that the partial grid behind the data points were actually error bars! However, it would be good to provide some estimate of the error. Since the supersaturation is used in the theoretical calculations, an error estimate on the supersaturation and the calculated growth would be good as well. Finally, the title of the figure led me to believe, initially, that the basal face radius was plotted, but it appears that this is a plot of the ring radius divided by the actual crystal radius. If this is correct, then you may want to clarify the title and, perhaps, use this as the y-axis label instead.

→We made most of these changes. We added error bars on the measurements. For the calculations, the rates for I and III are so small in comparison to observations that the uncertainty due to supersaturation uncertainty is within the thickness of the curves. For II, as this is a fit, we did not add the supersaturation uncertainty. We also clarified the title and added a sketch to clarify the variables. Also, we put "r/a" onto the y-axis label.

(22) Line 1, pg 15: This sentence was a little confusing to me. Should it read, "….was essential or if it was the greater amount of normal growth…"

→Agreed. We changed this to " As this is the only case we observed, it is hard to strongly argue a particular cause. One potentially important distinction from previous crystals with corner pockets is that the crystal in this case had a significantly greater rate of normal growth. We account for such normal growth by including S-type lateral growth along with the P-type in a possible mechanism argued in the next section."

(23) Line 20, pg 16: "…form near an edge or corner instead of one." I was a little confused by this sentence. Do you mean that there can be a single air pocket at a corner, but there can also be a pair of pockets near a corner but along the edge (as in Fig. 10h)?

→Changed to " Near the edge, the advancing fronts may generate pockets before converging, generating a pair of pockets instead of one. Figure 10e–h shows such a process." A few other small changes were made in the paragraph to help clarify the point.

(24) Line 19, pg 17: ""…the rim is narrower than that just inside the rim") This is a bit confusing and should probably be reworded.

→Now "(i.e., the rim radius is narrower than that just inside the hollow)".

(25) Figures 11 and 12: One way the discussion of these figures could be made a little clearer is if indications of various feature were made on the images themselves. For instance, on Fig. 12 one could indicate the "fan" like hollow in (a) and the flat terrace in (b).

→Done. We marked examples of center pockets, terraces, fanning, and corner pockets, referring to the markings in the text and captions.

(26) Line 19, pg 20: "analogous facet" is a bit ambiguous.

→Changed to " As with F-growth above, why would the thin front of the protrusion have a high density of growth sites that can efficiently collect all the AST flux and continue protruding?"

(27) Line 24, pg 20: "grown and sublimated in a pure vapor." It is my understanding that here you mean the situation where gas-phase diffusion becomes unimportant, which happens at very low pressures. However, at high temperatures aren't the vapor pressures high enough so that diffusion is still an issue? If so then perhaps one should add "near vacuum conditions."

→Even near melting, diffusion-driven vapor-gradient features such as hollows are not known to occur in a pure vapor. At least, we know of no such evidence. To help clarify this point, we added " in a pure vapor where such gradients are likely insignificant."

(28) Line 36, pg 20: "It would be less likely at the much lower-temperature protruding-growth effects found here." I found this sentence to be a bit awkward and suggest rewording.

→Changed to ", it would be less likely at much lower temperatures, such as for the corner pockets observed here near −30.0 °C."

(29) Line 30, pg 21: You may want to add a sentence or two here about why a small basal face is all that is needed.

→Yes. Changed as " Here, the appearance of a small basal face is all that is needed: AST from that face drives a small protrusion, and as the face grows, the rate increases due to the larger collection area and the protrusion extending into a region of higher vapor density. In this way, a large plate can develop, even when starting from the side of a small rime droxtal."

(30) Lines 36-37, pg 21: It may be worth pointing out that scroll forms are also found at lower temperatures (below -20C).

→Thanks. Added " Scrolls also form below −20 °C, also being part of some polycrystalline types."

(31) Appendix A: I very much like the model of protrusion growth. It is relatively simple but appears to capture the main physical features of the lateral spreading of a facet. On Line 1, pg 25, I assume this is an infinitesimally thin disk?

→Yes, clarified as " an infinitesimally thin disc"

(32) Line 2, pg 25: "Shifting and normalizing" What do you mean by "shifting"? Is this just the subtraction of N_infinity? And you may want to be a little more specific about the normalization.

→Rewritten as " It is convenient to shift $N$ and making the variables dimensionless as"

(33) Line 5, pg 25: "can be shown". Was this shown in Nelson (1994), otherwise it might be good to give a reference to the solution.

→Yes. We rewrote the part as

"… vapor density. To determine the normalized flux $F_v'$, we first assume it is known and solve for $\Delta N'$. In Nelson (1994), it is shown that

$$\Delta N'(\rho', z') = -F_v' \cdot h_{\mathrm{td}}(\rho', z') \ ,$$

(A3)

where the thin-disc basis function $h_{\mathrm{td}}$ is an integral of Bessel functions."

(34) Equation A5, pg 25: My understanding of this function is that it provides the appropriate basis

function for the area (r-x_s to a) over which the flux of vapor is non-zero. If this is correct, then it might be good to introduce this basis function in this way. Perhaps pointing out that the idea here is to define a basis function for the ring region over which the vapor flux is non zero.

→ Correct. We revised as " However, in the facet spreading case, the flux is non-zero only in the thin ring $r-x_s \leq \rho \leq a$, not the entire thin disc. So we consider now the "thin-ring" basis function $h_{tr}$ defined as…"

(35) Line 16, pg 25: "derivative of this function" Since there are two functions above this line, the function being referred to should probably be specified. I assume it is Eq. A5.

→ Correct. Replaced "this function" with the function.

(36) Line 15, pg 26: "From Eq A3 and A7…" I presume that h_td is replaced with h_tr though? If so then this should be specified.

→ Correct. To clarify, we added 1) " As the facet spreading situation is most similar to this thin-ring case, we use only $h_{tr}$ from here." to line 18 on page 25 and 2) clarified this definition " where $h_{tr}(r')$ is shorthand for the full expression plotted in Fig. A2 (i.e., $h_{tr}(r', 0, r', x_s')$)." Immediately after the equation.

(37) Line 5, pg 27: The refinements described here sound like they would produce a more precise model, but given that we lack detailed measurements it's probably not warranted. It seems to me that the present "simplified" model is well-suited for the measurements that were taken from the growth chamber.

→In response, we added " and should be suitable for the present measurements."